# Thousands of human non-AUG extended proteoforms lack evidence of evolutionary selection among mammals

Alla D. Fedorova [1,2] ✉, Stephen J. Kiniry[1], Dmitry E. Andreev [3,4], Jonathan M. Mudge [5] & Pavel V. Baranov [1] ✉

The synthesis of most proteins begins at AUG codons, yet a small number of non-AUG initiated proteoforms are also known. Here we analyse a large number of publicly available Ribo-seq datasets to identify novel, previously uncharacterised non-AUG proteoforms using Trips-Viz implementation of a novel algorithm for detecting translated ORFs. In parallel we analyse genomic alignment of 120 mammals to identify evidence of protein coding evolution in sequences encoding potential extensions. Unexpectedly we find that the number of non-AUG proteoforms identified with ribosome profiling data greatly exceeds those with strong phylogenetic support suggesting their recent evolution. Our study argues that the protein coding potential of human genome greatly exceeds that detectable through comparative genomics and exposes the existence of multiple proteins encoded by the same genomic loci.

The current paradigm of translation initiation in eukaryotes follows the scanning mechanism wherein the preinitiation complex (PIC), assembled on the small ribosomal subunit (40S) and containing initiator Met-tRNAi (methionyl tRNAi), scans the mRNA 5′ leader for an AUG codon in a suitable context using complementarity to the anticodon of Met-tRNAi. The first AUG codon entered the peptidyl-tRNA (P) site of the 40S subunit is usually employed as the start codon, but it can be missed in unfavourable surrounding contexts. When optimised, this 'Kozak' context has a purine at position −3 and guanine at +4 position relative to the AUG (+1 position). When the first AUG codon is skipped due to weak Kozak context, the next AUG codon can be used. This phenomenon is known as leaky scanning[1,2].

Although it was long believed that the synthesis of eukaryotic proteins initiates at an AUG start codon, translation initiation at codons differing by 1nt from AUG (near-cognate) have been documented in the early 80 s, albeit with much lower efficiency[3–6]. It occurs in spite of the near-cognate codon mispairing with the anticodon of Met-tRNA. This mispairing can be tolerated only during initiation because it is the only stage when an incoming Met-tRNAi is bound directly in the ribosomal P-site[7,8]. Unlike the A-site, where mRNA:tRNA interactions are thoroughly monitored by the decoding centre[9], the P-site is more promiscuous to mismatches in the codon:anticodon duplex[10–13]. Near-cognate triplets such as CUG, GUG, UUG, AUA, AUU, AUC and ACG have been shown to be recognised as starts at frequencies of -1–10% of AUG in the optimal context depending on the gene and study, while AAG and AGG are essentially not recognised[5,14]. However, there is an astonishing example of a highly efficient CUG initiation conserved in mammals. The CUG codon is located in the 5′ leader of the *POLG* gene which encodes the catalytic subunit of the mitochondrial DNA polymerase. The efficiency of initiation at this CUG is comparable (-60–70%) to that at an AUG in the optimal context[15]. It results in the translation of a 260-triplet-long overlapping open reading frame[16] called POLGARF, its functional role is suggested to be involved in the extracellular signalling[15]. Another example of a very conserved near-cognate initiation is the *EIF4G2* gene, whose translation is initiated at a GUG. It encodes a paralog of eIF4F complex subunit eIF4G which lacks the binding site for the cap-binding subunit eIF4E. The GUG-initiation rate for *EIF4G2* is unusually high, ~30% compared

[1]School of Biochemistry and Cell Biology, University College Cork, Cork, Ireland. [2]SFI Centre for Research Training in Genomics Data Science, University College Cork, Cork, Ireland. [3]Shemyakin-Ovchinnikov Institute of Bioorganic Chemistry, RAS, Moscow, Russia. [4]Belozersky Institute of Physico-Chemical Biology, Lomonosov Moscow State University, Moscow, Russia. [5]European Molecular Biology Laboratory, European Bioinformatics Institute, Wellcome Genome Campus, Hinxton, Cambridge, UK. ✉e-mail: 120220049@umail.ucc.ie; p.baranov@ucc.ie

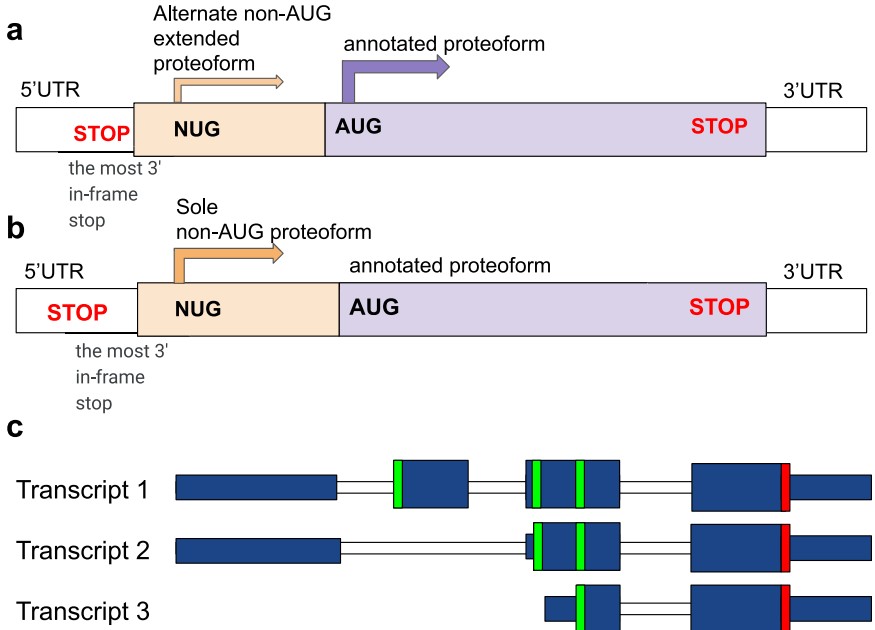

**Fig. 1 | Sources of alternative N-termini. a** Transcript with both major annotated proteoform and N-terminally extended non-AUG (NUG) initiated proteoform. **b** Transcript with sole non-AUG-initiated (NUG) proteoform. **c** Transcript 1 has an extra CDS exon relative to transcript 2, while transcript 3 has a different transcription starts. As a result, non-coding parts of the first CDS exons in transcripts 2 and 3 exhibits protein-coding evolution. Green bars represent start codons, red bars show stop codons. Thick blue bars represent coding exons, thin blue bars are non-coding exons.

to an AUG-mutant version of the same *EIF4G2* expression plasmid[17,18]. Initiation efficiency from a non-AUG start may be enhanced by secondary structure elements starting approximately 15 nt downstream of the non-AUG codon[19]. The Kozak context for non-AUG codons is similar to AUG and also important for initiation efficiency[20]. Overall, the mechanisms involved in non-AUG initiation are reviewed e.g. in refs. 14,21.

Due to leaky scanning, translation initiation from non-cognate start codons may result in extended proteoforms in addition to a proteoform resulting from initiation at the "main" downstream AUG (Fig. 1a). Genes with multiple non-AUG initiated proteoforms are known, e.g. human tumour suppressor *PTEN* with firstly identified CUG-initiated N-terminally extended proteform[22,23], then more abundant AUU-initiated proteoform and two additional CUG-initiated proteoforms have been discovered[24]. All of them retain the ability to downregulate the PI3K pathway[25]. Another possibility is translation initiation from non-cognate start codon downstream of AUG which leads to a truncated proteoform e.g. in *MRPL18*[26] and *ASCT2*[27]. Cases of exclusive translation initiation (Fig. 1b) at non-AUG codons have also been reported, e.g. already mentioned mammalian *EIF4G2*[28], human *TRPV6* gene (and its mouse *Trpv6* homologue) generates a single ACG-initiated TRPV6 protein, translation of human STIM2 occurs exclusively at a UUG start codon[29], human *TEAD1* exemplifies translation initiation at AUU start codon[30]. In total, more than 60 instances of non-AUG-initiated proteoforms have been reported[22,31].

Detection and annotation of non-AUG-initiated proteoforms is clinically important since their misregulation may lead to multiple human diseases including neurodegenerative disorders and cancer progression. For instance, *FGF2* controls cell proliferation, differentiation and angiogenesis and it has a canonical AUG-initiated proteoform which is mostly cytoplasmic or secreted. At least four upstream CUG-codons can be used to generate longer isoforms that localise to the nucleus, therefore, causing cell immortalisation[32,33]. The *MYC* proto-oncogene regulates cell proliferation and transformation. Two proteoforms are generated from *MYC* using a canonical AUG codon and an upstream in-frame CUG codon. The CUG-encoded

proteoform becomes more prevalent during the limited availability of amino acids when the density of cells increases and its overexpression was shown to inhibit the growth of cultured cells. Inactivation of this proteoform is observed in Burkitt's lymphomas suggesting that the inability to generate this CUG-initiated proteoform may provide a selective growth advantage[4,34].

Detection of extended non-AUG proteoforms can be carried out by various approaches. Bioinformatics methods include comparative genomic analysis where multiple nucleotide sequence alignments analysed for the presence of substitution patterns typical for protein-coding evolution e.g. reduced rate of non-synonymous substitutions relative to synonymous measured as their ratio, $d_N/d_S$ or Ka/Ks[22]. Western blots, ribosome profiling and proteomics can serve as experimental support for predicted extensions. Ribosome profiling (Ribo-seq) is a method based on deep sequencing of ribosome-protected mRNA fragments which was introduced by Nicholas Ingolia and Jonathan Weissman[35]. Modifications of the Ribo-seq such as QTI-seq have been developed to capture initiation ribosomes and can be used to infer the start of translation[36]. The identification of non-AUG proteoforms can also be assisted with the application of machine learning methods[37].

Here we analysed multiple publicly available ribosome profiling datasets and genomic alignments of 120 mammals to detect translation and protein-coding evolution in sequences encoding potential non-AUG extensions. Our analysis suggests that thousands of such extensions in human proteoforms do not exhibit signatures of protein-coding evolution typical for proteins in mammals.

## Results

### Detecting purifying selection upstream of annotated coding regions

Purifying selection is a typical evolutionary signature of protein-coding sequences. One would expect that if translation initiates upstream of the annotated start codon, this upstream region should evolve as a protein-coding sequence. We used this indicator for the identification of N-terminally extended proteoforms in the human genome utilising

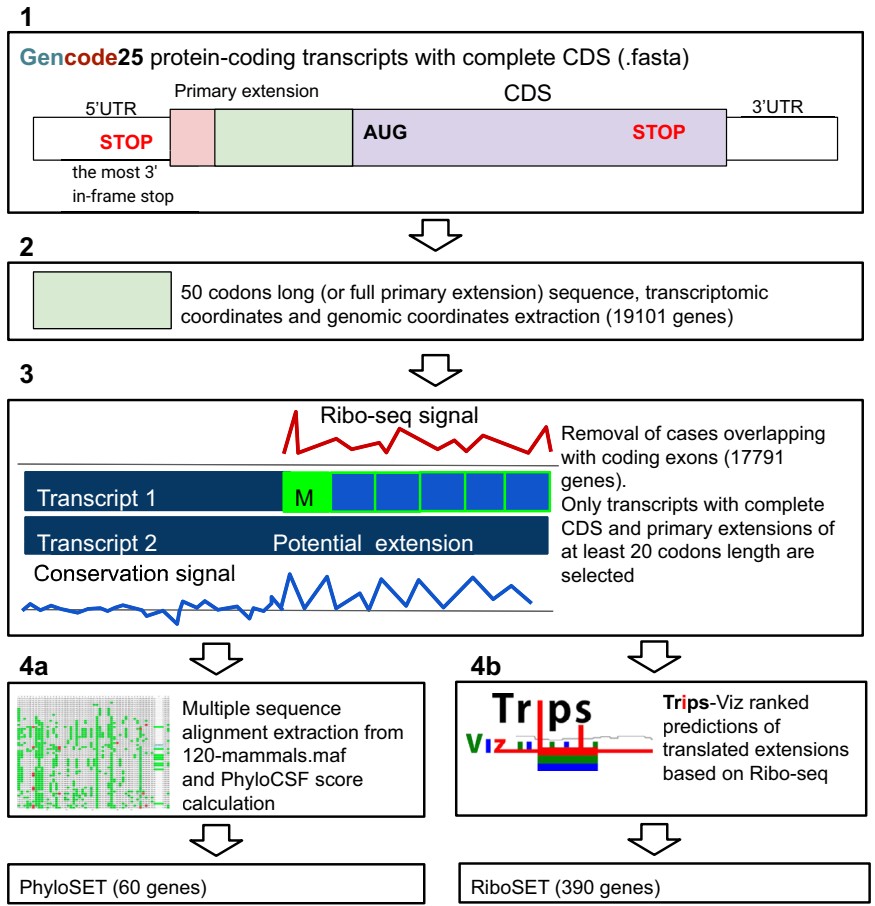

**Fig. 2 | Scheme of the pipeline.** (Step 1) GENCODE v25 protein-coding transcripts with complete CDS were used to extract primary extension; (step 2) 50 codons-long extension (or the whole primary extension if it is shorter than 50 codons) was obtained; (step 3) transcripts for which extensions overlap with coding exons in any reading frame were excluded; (step 4a) PhyloCSF score was calculated for the 50 codons-long set of extensions using 120-mammals multiple sequence alignment shown with CodeAlignView; strictly positive threshold led to a set of 60 genes which we called PhyloSET; (step 4b) the retrieval of the ranked extensions predicted based on ribosome profiling data in Trips-Viz browser that composed RiboSET (390 genes).

PhyloCSF score. PhyloCSF is a state-of-the-art method for assessing the evolutionary protein-coding potential of a genomic region based on multiple sequence alignment[38]. However, the protein-coding evolution of an upstream region does not necessarily mean that the underlying mechanism is the non-AUG initiation or alternative transcription (Fig. 1c).

Determining the exact start of non-AUG extensions in practice can be quite challenging because there might be several non-AUG starts upstream of AUG like in the case of *PTEN*. So we focused on predicting the genes which are most likely to have alternative extended non-AUG proteoforms irrespective of our ability to identify specific locations of start codons. The final set of candidates (PhyloSET, see Supplementary Data 1A) was obtained via the following steps (see Methods for further detail) shown in Fig. 2, steps 1–4a. PhyloCSF score is ranging from 0.1452 (*NRXN1*) to 2693.8893 (*CCDC8*) with a median value of 155.9191 (*TRPC1*). PhyloCSF scores per codon were calculated to observe how the selection changes over the selected upstream region. Ideally, we would expect that the start of extension is clearly separated from the non-coding sequence, in other words, PhyloCSF score becomes positive at the border between the extension and preceding non-extension part as it happens for the *CCDC8* gene (Fig. 3a). However, for the majority of genes, no such clear change in scores can be spotted perhaps because evolutionary selection on N-terminus is relaxed in comparison with internal parts.

## Detecting translation upstream of annotated coding regions with Ribo-seq and proteomics data

Another set of candidates (RiboSET) was selected solely based on ranked translated extensions predicted using ribosome profiling data (both elongating and initiating) with Trips-Viz, (Fig. 2, steps 1–3, 4b and Supplementary Data 1B). Trips-viz is a computational data environment for analysing Ribo-seq data on a transcriptome level[39,40]. It contains thousands of uniformly processed public Ribo-seq data and provides tools for analysis and visualisation of translation. Our recent addition to the Trips-viz platform is Ribo-seq ORF predictor which outputs a list of ranked translated ORFs. The biggest advantage of the tool is that a large number of processed public Ribo-seq data is already available to users and they can apply ORF predictor and visualise results immediately. The tool is tailored to detect different types of ORFs including uORFs, N-terminal extensions (NTEs), nORFs, CDSs and dORFs. The algorithm for NTEs is based on triplet periodicity present across the entire length of the extension and utilises patterns of ORF translation such as consistency of ribosome footprint triplet periodicity within the reading frame, the increase of footprint density at the potential start, non-zero coverage and average read density (see Supplementary Methods). Of note, it automatically filters out regions that overlap with coding exons. We employed this algorithm to detect non-AUG N-terminally extended proteoforms using aggregated elongating and initiating ribosome profiling data with high triplet periodicity scores (Supplementary Data 8).

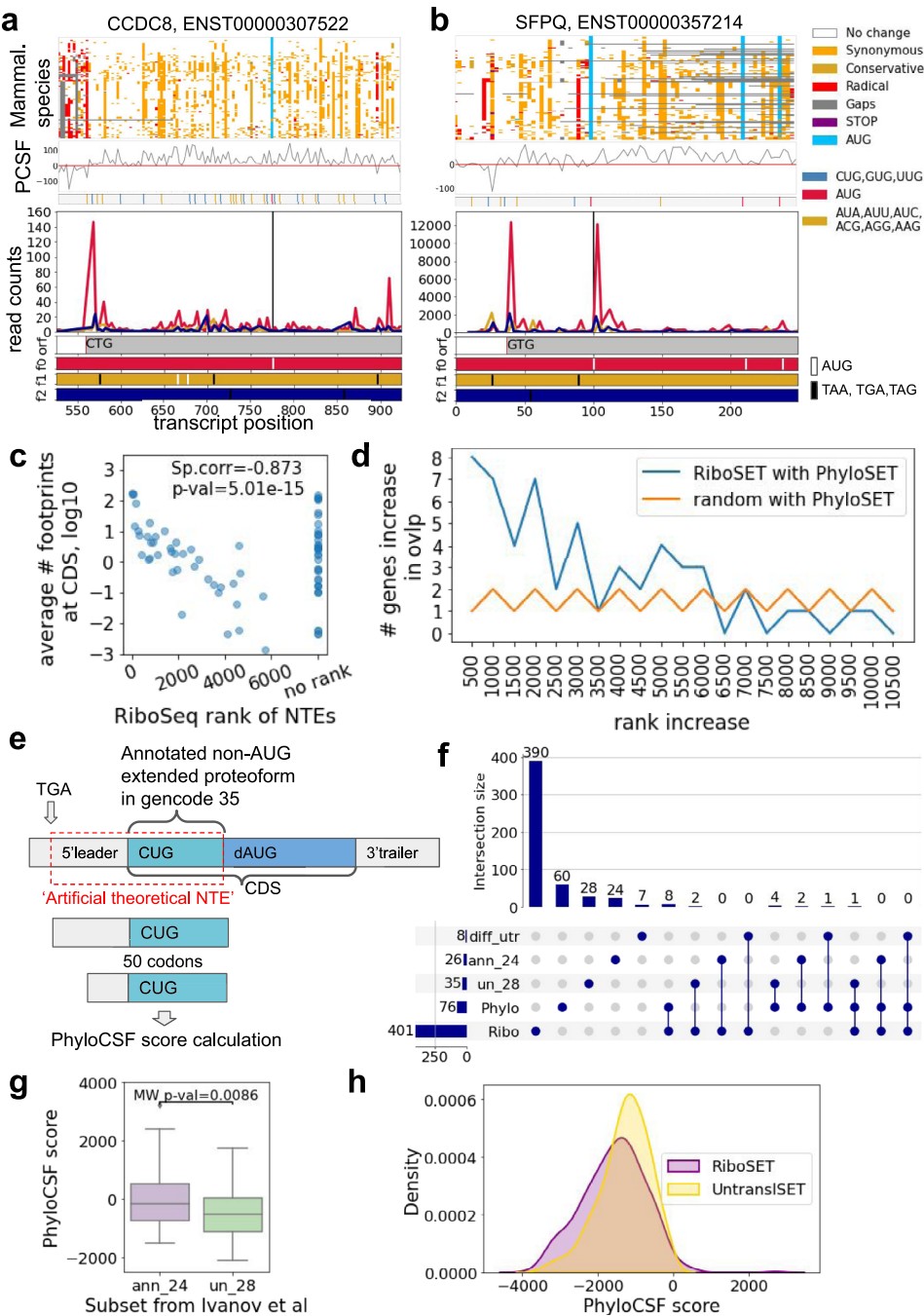

**Fig. 3 | Characterisation of genes with non-AUG initiation.** Genes from both PhyloSET and RiboSET (**a**–*CCDC8*, **b**–*SFPQ*), primary extension and the first 50 codons of CDS are shown. The top panel is a colour-coded codon alignment. The middle panel represents the PhyloCSF score per codon (the bottom bar shows the positions of potential start codons). The bottom panel shows Trips-Viz subcodon Ribo-seq profiles with densities of ribosome footprints differentially coloured based on the supported reading frame. The colours are matched to the reading frames in the ORF plot at the bottom. Black vertical lines indicate the start of the annotated CDS. Grey bars are extended CDS initiated at the proposed non-AUG starts. **c** Average footprint density at CDS of PhyloSET genes (log10) compared to the Ribo-Seq rank of N-extensions in them; lowest rank is 10470. Spearman correlation (two-sided) for genes with known rank (corr = −0.873, *p* value = 5.01e-15). **d** The size of the increase in the number of genes between PhyloSET and RiboSET overlap depending on Ribo-Seq NTE threshold. The distributions (from 500 to 6500 rank, where they overlap) are compared with Mann–Whitney *U* two-sided test

(*p* value = 0.0016, statistic = 28.5). **e** Re-identification of non-AUG N-terminal extensions in 24 genes from study Ivanov et al[22]. 'Artificial theoretical NTE' starts from the most 3′ in-frame stop codon and stretches till the first downstream ATG right after non-AUG. PhyloCSF score is calculated for the first upstream 50 codons of theoretical extension. **f** Comparison of RiboSET and PhyloSET with genes from ref. [22]. 'Phylo'–PhyloSET, 'Ribo'–RiboSET, 'ann_24'–genes with annotated non-AUG extensions in GENCODE v35, 'un_28' and 'diff_utr' are genes which non-AUG extensions are not annotated in GENCODE v35 ('diff_utr' genes have different 5′leaders from RefSeq). **g** PhyloCSF score of upstream regions of 'ann_24' (with starts moved downstream) and 'un_28'[22]. Box plots: the central line indicates the median, the box limits indicate the interquartile area and whiskers indicate 1.5 × interquartile range. The Mann–Whitney *U* one-sided test (*N* = 24 and 28 genes, *p* value = 0.0086). **h** PhyloCSF score of upstream regions of RiboSET and UntranslSET genes. Source data is provided as a Source Data file.

In the absence of an objective threshold for discriminating genuine translation from biological and technical noise, we decided to incorporate 500 top scoring extensions into RiboSET which upon further filtering (Methods) was reduced to 390 genes. Although important to mention that the actual number of translated non-AUG extensions is much higher (multiple extensions ranked below 5000 are reliably translated). Proteomics data available in Trips-Viz[40] supported extensions in 90 genes in this RiboSET (18 in PhyloSET). Only eight genes (*CCDC8, CYTH2, FXR2, H1FX, HNRNPAO, MARCKS, RPTOR* and *SFPQ*) are common between PhyloSET and RiboSET (Fig. 3a, b and Supplementary Fig. 1). Overlap between RiboSET and PhyloSET is statistically significant (hypergeom. $p_{adj}$ = 3.59e-05).

The small overlap between PhyloSET and RiboSET requires an explanation (more detailed in Discussion). Genes may occur in PhyloSET exclusively either because they are not expressed in the cells for which Ribo-seq data are available or because the 500 top-ranking threshold is too conservative. To explore the first possibility we studied a relationship between the rank of predicted extension and CDS coverage—the average number of footprints at CDS. We observed that the lower the average number of footprints at CDS, the lower extension is ranked (Fig. 3c). To show whether a threshold of 500 top-ranking candidates is too conservative, we explored the size of the overlap increase (we use step = 500) between PhyloSET and RiboSET depending on the ranking threshold (Fig. 3d). It appears that until the rank reaches 6500, there is a statistically significant ($p$ = 0.0016, Mann–Whitney $U$-test) increase in the overlap size in comparison with what would be expected by chance suggesting that Ribo-seq signal above our selected threshold has a clear positive value at predicting genuine N-terminal extensions. Genes with high CDS coverage but with no rank may still have extensions translated under certain conditions (examples can be found in Supplementary Fig. 2). Thus the top-500 cutoff is very conservative. While this threshold avoids false positives, it also generates many false negatives. Hence, we also generated an extended dataset with a more relaxed threshold equal to 5000 which after filtration resulted in 3451 genes (Supplementary Data 2).

We also addressed the occurrence of in-frame and out-of-frame AUGs within extensions (Supplementary Notes and Supplementary Figs. 3–6).

### Comparison with previously identified non-AUG proteoforms
We also wanted to know how well translation detected with ribosome profiling concords with phylogenetic approaches. We performed comparisons of a set of predicted proteoforms from the previous study[22] with PhyloSET and RiboSET. In brief, the study utilised alignments of human and mouse RefSeq transcript sequences which resulted in the prediction of 59 genes with evolutionary conserved extensions. PhyloSET and RiboSET are meant to have only new non-AUG proteoforms which have not been described in GENCODE v35 and the latest RefSeq annotation (due to the exclusion of overlapping coding exons). In GENCODE v35, 24 non-AUG proteoforms from study[22] have been annotated (we called this set 'ann_24'); 28 genes have not been annotated with non-AUG proteoforms and retained intact extension sequences detected in the previous study ('un_28') and 7 genes remained without annotated near-cognate initiated proteoforms or intact extensions ('diff_utr', *HELZ2, ANKRD42, WDR26, ZFP62, C1QL1, PTEN, TIAL1*, where *WT1* was shown to be annotated and intact extension in *TIAL1* still corresponds to only nonsense-mediated decay transcript, (Supplementary Data 3). Among 28 genes there were four genes found in PhyloSET and two genes in RiboSET. Among seven genes one gene and zero genes were shown in PhyloSET and RiboSET correspondingly (Fig. 3f). As could be expected overlap between 'un_28' and PhyloSET is statistically significant in comparison to all protein-coding genes (hypergeom $p_{adj}$ = 2.838e-06) while the overlap between 'un_28' and RiboSET is not significant (hypergeom $p_{adj}$ = 0.122).

We also decided to test whether a PhyloCSF-based approach is able to re-identify those 24 genes ('ann_24') which have been already annotated in GENCODE v35. First, we created a set of transcripts with non-AUG proteoforms for these genes where the start of CDS is moved to downstream AUG (Fig. 3e). It turned out that the PhyloCSF score is positive for less than half of genes (11/24, Supplementary Data 3). The remaining 13 genes have not shown a positive PhyloCSF score. It might be explained by the extended part being much shorter than 50 codons which may have led to an excessive codons' impact on negative scores (*MYC, YPEL1, HCK* and *TRPV6*). We also compared the distribution of PhyloCSF scores for upstream regions of 'ann_24' genes with the moved start of CDS and 'un_28' genes (Fig. 3g). The already annotated genes ('ann_24') have significantly higher PhyloCSF score (Mann–Whitney, $p$ value = 0.0086) than not yet annotated ones ('un_28').

When compared to the previous study (ref. 22), we observed a statistically significant but nevertheless small overlap (four genes) with PhyloSET. One would expect that due to both studies using similar approaches (phylogenetic analysis in mammalian species), the overlap should be higher. This can be explained by multiple reasons including that 24 genes out of a total of 60 have already been annotated in GENCODE v35 and thus were excluded from our analysis and seven genes have unmatching sequences of 5'UTRs in comparison to RefSeq which was used in a previous study. Only 28 genes were left available for discovery. We used multiple sequence alignments of 120 mammals in contrast to ref. 22 where only paired alignments of human and mouse were studied. It could explain why such a small overlap was observed - the probability of upstream regions to be conserved in a wider range of species is generally lower than in just two species. Also, the length of the region we took for evaluation is 50 codons-long which simply might be longer than the actual extension thus leading to a negative PhyloCSF score derived from excessive upstream non-conserved triplets not included in the actual extension. The same reason may explain why we could re-identify only half of the genes already annotated in GENCODE v35 using our phylogenetic approach. Nevertheless, PhyloCSF score of extensions in already annotated genes from ref. 22 study is significantly higher than in upstream regions of not yet annotated genes (Fig. 3g).

We revisited Ribo-seq profiles for 'ann_24' genes (Supplementary Figs. 7, 8). Lack of Ribo-Seq data coverage was shown for the entire mRNA of 7 genes (*FNDC5, NR1I2, PRPS1L1, TRPV6, HCK, YPEL1* and *OAZ3*); extensions are clearly supported by Ribo-seq data in 16 genes, although for *KCTD1* it is not clear whether extension starts where it is annotated.

In addition, we also compared our gene sets with Van Damme et al. study[31] which identified 17 human genes with non-AUG N-terminal extension using ribosome profiling and N-terminal proteomics (Supplementary Data 3). Among 17 candidates, two genes from PhyloSET (*FXR2* and *HNRNPAO*) and seven genes from RiboSET (*NARS, HDGF, HNRNPAO, FXR2, SYAP1, KAT7* and *BAG6*) were present. Both overlaps between Van Damme et al. study and PhyloSET or RiboSET are significant (hypergeom. $p_{adj}$ = 0.0016 and 1.002e-07 correspondingly). We also compared RiboSET and PhyloSET gene lists with extensions detected with the N-terminal-peptide-enrichment method from the study Yeom L. et al[41]. Only 1 gene from PhyloSET and 17 genes from RiboSET were present among 171 genes with N-terminal extensions (Supplemental Data 3). Overlap between Yeom L. et al study and RiboSET is significant (hypergeom. $p_{adj}$ = 1.322e-07) in contrast to PhyloSET (hypergeom. $p_{adj}$ = 0.403).

Of note, there was a significant discrepancy between RefSeq and GENCODE gene annotations in *PTEN*. In the latest RefSeq mRNA the CUG-initiated proteoform is annotated correctly, while in GENCODE v35 this proteoform has not been annotated yet and 5' leader of the only one available transcript ENST00000371953. This can be explained by the incorrect sequence of the reference genome

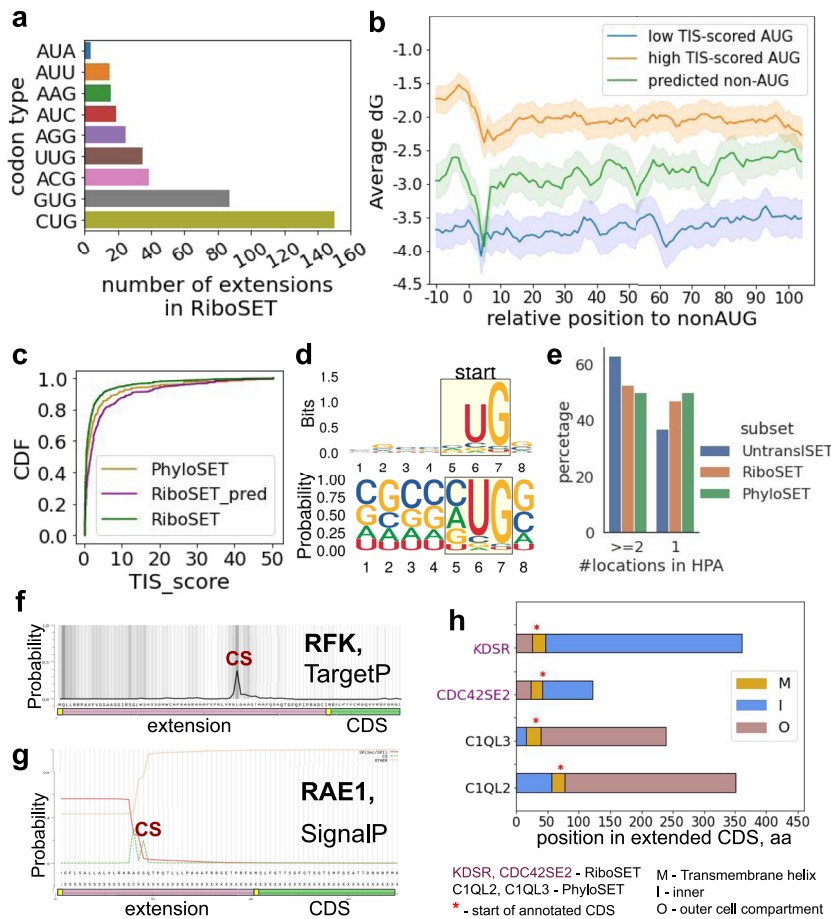

**Fig. 4 | Characterisation of predicted non-AUG initiation codons. a** Distribution of start codon types predicted by Trips-viz (RiboSET). **b** The stability (dG, Gibbs free energy) of mRNA secondary structure downstream of start codons within 22nt window '0' corresponds to the start codon. 390 starts from RiboSET (green line), a sample of 400 AUG starts with high-scored TIS (blue line), a sample of 400 AUG starts with low-scored TIS (orange line). Lines are mean values across genes with 95% confidence intervals. **c** Cumulative distribution functions of TIS scores for all non-AUG codons in theoretical NTE, RiboSET (green), all non-AUG codons in theoretical NTE, PhyloSET (yellow), Trips-viz predicted non-AUG starts, RiboSET

(purple). **d** TIS sequence logo and frequency plot of non-AUG starts predicted by Trips-viz. **e** Fraction of genes in RiboSET, PhyloSET and UntranslSET with one or at least two alternative localisations in the Human Protein atlas. **f** Probability of mitochondrial presequence calculated by TargetP 2.0 for RFK. **g** Probability of signal peptide predicted by SignalP 5.0, for RAE1. **h** Domain organisation of proteoforms from RiboSET (purple labels) and PhyloSET (black labels) with TM regions found by DeepTMHMM. Yellow region (M) is the TM helix, blue (I) is inner and pink (O) is the outer cell compartment. Red asterisks represent the start of CDS in a proteoform. Source data is provided as a Source Data file.

(assembly GRCh38)—it has the variant which is known as NC_000010.11:g.87864104delT and its global minor allele frequency in 1000 Genomes 0.00000 (T). This variant introduces frameshift into the 5′ leader of a transcript thus disrupting the sequence of CUG-extension. In RefSeq gene annotation this variant is cut from the transcript sequence and shown as 1nt-intron in Genome Browser (Supplementary Fig. 9).

## Characterisation of genes with predicted non-AUG initiation

Firstly, we studied the distribution of start codon type across starts predicted by Trips-viz in RiboSET. As expected, the most frequent non-AUG start in extensions was CUG, followed by GUG and ACG (Fig. 4a and Supplementary Fig. 10e). In RiboSET only one non-AUG initiation site per transcript is predicted based on internal probability ranking relying on features associated with the intensity of Ribo-seq signal. Therefore, one would expect that the initiation efficiency of such starts may be facilitated by certain features including the optimality of Kozak context and downstream secondary mRNA structures. It has been shown that certain non-AUG start codons with the appropriate sequence context can initiate translation comparable to that of AUG start codons[42]. The efficiencies of TIS (Translation Initiation Site) including the start codon and four positions upstream and

downstream were previously measured with FACS-seq. For this purpose, a library of fluorescent reporters under control of all possible contexts surrounding near-cognate initiation codons was transfected into cells and then cells were sorted based on fluorescence. The efficiency of a specific TIS was measured based on its enrichment within a specific fraction and scaled relative to the optimal TIS (CACC<u>AUG</u>G) efficiency score set to 100[42] (the scores of non-AUG starts were found in the range from 0.2 to 50.4). We compared TIS scores of predicted non-AUG starts from RiboSET (Supplementary Data 4) to all other non-AUG codons in theoretical extensions in RiboSET and PhyloSET (Fig. 4c and Supplementary Fig. 10d). It is clearly seen that predicted non-AUG starts in RiboSET have more favourable initiation contexts in comparison to all theoretical non-AUG codons in primary extension sequences thus endorsing Trips-viz-based prediction method (Kolmogorov–Smirnov two-sample test, *p* value 1.19e-18, statistic = 0.245). Similarly to AUG, the optimal sequence context of predicted non-AUG starts tends towards having guanosine in −4 and +4 positions (Fig. 4d and Supplementary Fig. 10f).

The next step was to assess the stability and presence of mRNA secondary structures located downstream of predicted start codons. It has been shown that a strong RNA secondary structure located downstream of the initiation site significantly increases the efficiency

of initiation at non-AUG codons[19,43]. We selected 400 genes with high-scored AUG-containing TISs and 400 genes with low-scored AUG-containing TISs as well as all predicted non-AUG TISs from RiboSET (390 genes). We then used RNAfold[44] to calculate the free energy of predicted RNA secondary) within a sliding window of 22nt with the step of 1nt in the region surrounding the potential start codon (10nt upstream and 100nt downstream), see Fig. 4b and Supplementary Fig. 10c. As expected, more stable mRNA secondary structures were present on transcripts with less optimal TIS codons.

Next, we compared PhyloCSF scores of upstream regions of RiboSET genes and genes with no translation upstream. In brief, we created a set called UntranslSET containing 384 genes with no translation upstream in theoretical N-terminal extension but with well-translated CDS and a theoretical extension is at least 20 codons (Methods). Interestingly, it turned out that both PhyloCSF score distribution of RiboSET and UntranslSET are skewed towards negative values (Fig. 3h), however, we can clearly see (Kolmogorov–Smirnov, $p$ value = 2.23e-06) that PhyloCSF score of RiboSET genes is generally lower than of UntranslSET. More importantly, the PhyloCSF score of upstream regions of RiboSET_ext is also skewed towards large negative values providing us with evidence of a lack of evolutionary selection among mammals for thousands of translated non-AUG extensions (Supplementary Fig. 10a). Such a significant difference in PhyloCSF scores between UntranslSET and RiboSETs might be explained by the length of the theoretical extensions (Supplementary Fig. 10g, h)−upstream regions in UntransSET are shorter than in RiboSET (RiboSET_ext) therefore PhyloCSF score might be higher for them.

One would expect that proteoforms with different alternative N-termini (PANTs) may possess different functional properties. For instance, longer proteoforms may contain a signal of subcellular localisation in their extended part for alternative compartmentalisation[33,45,46]. Functional gene enrichment analysis was performed using the Gene Ontology resource[47,48] using simplified GO terms[49]. Significant enrichment for genes from RiboSET was shown in 9 terms in 'cellular component' and 4 terms in 'molecular function'. We also observed overlaps between terms, e.g. 58 genes associated both with term 'nucleoplasm' and 'cytoplasm', 29 common genes between 'cytoplasm' and 'membrane' and 16 genes between 'nucleoplasm' and 'membrane' terms (Supplementary Data 5, Fisher's exact test, $p$ values are adjusted with Benjamini–Hochberg correction). No enrichment was shown for PhyloSET genes. We also repeated the analysis for RiboSET_ext and found similar patterns (Supplementary Data 5, Fisher's exact test, $p$ values are adjusted with Benjamini–Hochberg correction).

In eukaryotes, N-terminal targeting signals include mitochondrial targeting signal and the signal sequence for the secretory pathway (signal peptides)[50]. Membrane proteins may also contain a signal peptide, but most often the N-terminal transmembrane (TM) region functions as the signal sequence[51]. First, we extracted the main subcellular location of proteins based on immunofluorescently stained cells from The Human Protein Atlas (HPA)[52,53]. We split genes into two groups based on the number of alternative cell compartments: 1 or at least 2 (Fig. 4e and Supplementary Fig. 10b). We found no significant enrichment in multiple localisation for genes in RiboSET or PhyloSET (and RiboSET_ext) in comparison to UntranslSET (Supplementary Data 6).

Next, we utilised algorithms for the prediction of localisation signals in the extended proteoforms from RiboSET. SignalP 5.0 is a deep neural network approach that detects the presence of signal peptides and the location of their cleavage sites in proteins[54]. We utilised the web-server interface for extended proteins from RiboSET, PhyloSET, UntranslSET and RiboSET_ext. Given that signal peptide length varies from 16 aa to 30 aa[55], 10 genes from RiboSET (53 genes from RiboSET_ext) turned out to have signal peptide at least partially within extension; no genes in PhyloSET with signal peptide within

theoretical extensions were found (Supplementary Data 6). Among genes with predicted signal peptides, in RiboSET there were 7 genes (44 in RiboSET_ext) with detected signal peptides residing entirely within the N-terminal extension including *RAE1* (Fig. 4g).

We also explored the possibility that extended parts of proteoforms can target them to mitochondria. We applied TargetP 2.0[56] and given the length of mitochondrial presequences is 20–60 amino, we found 18 genes in RiboSET (83 in RiboSET_ext) and no genes in PhyloSET with mitochondrial signal in their extensions (Supplementary Data 6). For 12 genes (57 in RiboSET_ext) including RFK (Fig. 4f) the cleavage site of mitochondrial transfer peptide was located within the extension part.

Next we explored the existence of N-terminal transmembrane helices within predicted extensions using the transmembrane domain prediction ability of DeepTMHMM[57]. We found two genes from RiboSET and two genes from PhyloSET (12 genes in RiboSET_ext, Supplementary Data 6) with the first TM helix located at least partially within the extension (Fig. 4h).

Next, we tested whether SignalP, TargetP, and DeepTMHMM detection of compartmentalisation signals is more common in translated or conserved upstream extensions than in equivalent genes with no upstream translation (UntranslSET). The occurrence of mitochondrial presequence predicted with TargetP in RiboSET was shown to have significant enrichment of presequences in comparison to vs UntranslSET ($p$ value = 0.022) although after multiple testing correction the $p$ value did not retain significance. Also, no TM helices were found in UntranslSET unlike 2 and 2 genes in RiboSET and PhyloSET (12 genes in RiboSET_ext). For other predictors, there was no significant enrichment of upstream regions of RiboSET (RiboSET_ext) and PhyloSET genes in predicted localisation signals in comparison to genes with no detected translation (Fisher exact test, alternative = 'greater', $p_{adj}$ > 0.05, Supplementary Data 6). Taken together, no strong association between alternative localisation signals and non-AUG extensions were found.

## Exclusive non-AUG initiation

One intriguing aspect of non-AUG initiation is that it can be exclusive, which means that unlike in case of PANTs non-AUG initiated proteoform is the only one synthesised from a transcript. This suggests that the non-AUG initiation function might be different from the production of alternative proteoforms. There are very few known examples of sole non-AUG initiation e.g. *EIF4G2*, *TRPV6*, *TEAD1* and *STIM2* and the reason why non-AUG is preferred evolutionary over AUG has not been elucidated yet. Here we reported several examples of most likely exclusive non-AUG initiation according to their Ribo-seq profiles (Fig. 5, Supplementary Figs. 11, 12, Supplementary Data 7 and Supplementary Notes).

## Reannotation of non_AUG proteoforms

One of the goals of this study is updating human gene annotation with newly identified non-AUG proteoforms as well as reannotation of incomplete or incorrect transcript isoforms discovered along the way. GENCODE also maintains annotation of mouse genes, so human orthologs in mouse have also been incorporated. Of note, annotation requires not only the information about extension being present for that gene, but also an exact position of translation initiation which is challenging to infer precisely due to multiple reasons.

Therefore, the initial phase of annotation includes several immediate cases. Thirty genes have maintained canonical AUG start is their models in human, while additional models with non-AUG start have been introduced: *SFPQ* (CUG), *VANGL2* (AUA), *CCDC8* (CUG), *PELI2* (CUG), *CYTH2* (CUG), *FXR2* (GUG), *H1F* or *H1-10* (CUG), *RPTOR* (CUG), *USP19* (AUA), *SLC6A1* (CUG), *NPLOC4* (GUG), SLC25A32 (UUG), ADO (CUG), JUN (CUG), HDGF (AUU), POGZ (ACG), BRD7 (CUG), PIM2 (CUG), PTMS (CUG), CARM1 (CUG), TARBP2 (ACG), CHTOP (GUG),

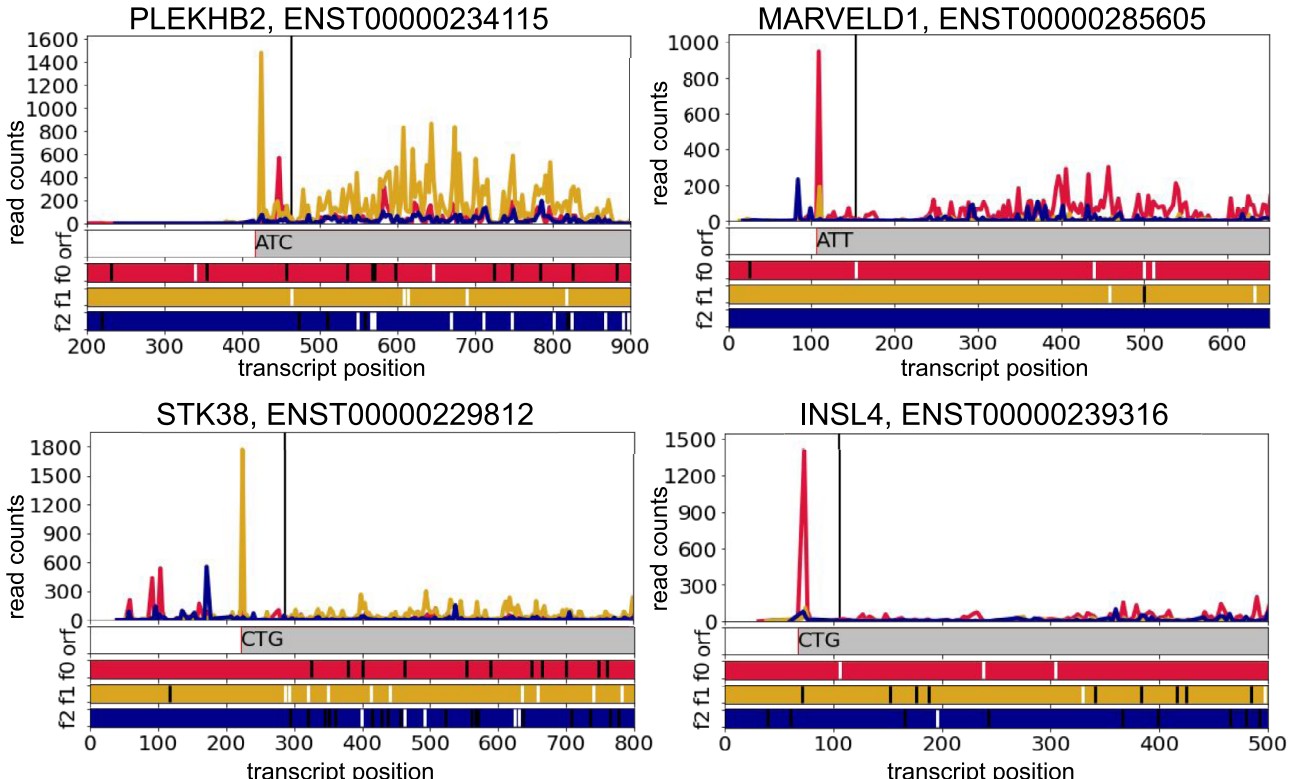

**Fig. 5 | Subcodon Ribo-seq profiles with the densities of ribosome footprints differentially coloured based on the supported reading frame for examples of predicted genes with exclusive non-AUG initiation (*STK38*, *MARVELD1*, *PLEKHB2* and *INSL4*).** The colours are matched to the reading frames in the ORF plot at the bottom where AUG codons are depicted as white and stop codons as black dashes. Black vertical lines indicate the starts of the annotated CDS. Grey bars correspond to extended CDS initiated at the proposed non-AUG starts. The genomic intervals including primary extension and at least 150 first codons of CDS are shown. Source data is provided as a Source Data file.

TNKS2 (UUG), KAT7 (ACG), FHIP2A (CUG), SYNCRIP (CUG), KCTD9 (GUG), PPP4R2 (CUG), ZNF384 (CUG) and SNRNP25 (CUG). Similarly, non-AUG extensions will be annotated in the next GENCODE release for mice orthologs (*Sfpq, Vangl2, Ccdc8, Peli2, Cyth2, Fxr2, H1fx* or *H1f10, Rptor, Usp19, Slc6a1, Nploc4, Slc25a32, Ado, Jun, Hdgf, Pogz, Brd7, Pim2, Ptms, Carm1, Tarbp2, Chtop, Tnks2, Kat7, Fhip2a, Syncrip, Kctd9, Ppp4r2, Zfp384* and *Snrnp25*). One more gene (*XRRA1*) which was not included in RiboSET due to its rank (rank 711, below 500 threshold) has also gotten new transcript models both for mouse and human (GUG). Additionally, AUG-extended proteoform in human (*PTPRJ*) has been introduced.

Having a comprehensive and accurate gene annotation is crucially important for a wide research community, especially for clinicians who heavily rely on gene annotation with regard to variant interpretation. Since we discovered novel protein-coding regions, we overlapped ClinVar variants with predicted non-AUG extensions from RiboSET_ext and found 124 genes (201 variants) with either pathogenic, likely pathogenic or conflicting interpretations of pathogenicity (Supplementary Data 9). This is likely to be an underestimation of variation simply because only annotated coding regions are generally used for variant calling. Similarly, for primary extensions in PhyloSET we found gene *GDF5OS* carrying 4 variants (pathogenic and likely pathogenic, Supplementary Data 9). Therefore this set can be used for assessing variation occurring in 5′UTRs.

## Discussion

In this study, we developed two approaches for the detection of non-AUG-initiated proteoforms in the human genome. The comparative genomics approach is based on utilisation of the PhyloCSF score which is able to capture signatures of protein-coding evolution in the upstream region of CDS. It resulted in 60 candidate genes (PhyloSET).

The second approach employs Ribo-seq data to predict translated extended proteoforms. The analysis of 500 top-ranked extensions based on Ribo-seq support led to 390 gene candidates after additional filtering (RiboSET) with only 8 common genes between RiboSET and PhyloSET.

For the prediction of translated non-AUG NTEs we utilised a Ribo-seq ORF predictor—a new addition to the Trips-viz environment—platform for analysing and visualising ribosome profiling data. The key advantage of the ORF predictor is its convenience: users have a large collection of Ribo-seq samples already embedded into the platform and can start using the predictor and visualise results straight away. Its output is in the form of a ranked list of translated ORFs that can be treated as both advantage and disadvantage since it provides flexibility for setting out your own threshold although it might not be always straightforward to select one. In general, the Trips-viz ORF predictor algorithm is adapted individually for all classes of ORFs including uORFs, CDSs, NTEs, nORFs and dORFs. It can use ribosome profiling data obtained with inhibitors of both, elongation and initiation. The latter is achieved with inhibitors such as lactimidomycin that preferentially arrest ribosomes at the sites of initiation[58]. Variations of this approach are known under different names such as Quantitative Initiation Sequencing QTI-seq[36] and are very useful in the identification of translation initiation sites. However, the data obtained with this approach should be used with caution because it is intrinsically noisy as reads come from a single location per initiation site. Translation initiation inhibitors also generate artefacts such as the arrest of ribosomes at AAG/AGG codons during elongation and may distort translation initiation[59–61]. Thus, we believe that the combination of data obtained with different approaches is more reliable.

We did not aim to exhaustively detect all existing non-AUG-initiated proteoforms but rather compare two approaches and the

relationship between translated N-terminal extensions detected with ribosome profiling and evidence of protein-coding evolution in the corresponding genes.

The large overlap between the sets would not have been expected. For example, some genes from PhyloSET may not be expressed in the cells for which Ribo-seq data are available. Indeed the lower CDS coverage, the lower the extension ranking based on Ribo-seq data, see Fig. 3c. Interestingly, however, we saw several genes with no rank assigned (no Ribo-seq evidence) having significantly high CDS coverage. Given strong evolutionary support for these extensions, there could be two potential explanations for this. The extended proteoforms are required in extremely small quantities or they are being synthesised only in specific cells or under specific conditions for which Ribo-seq data were not available. It would be interesting to explore the latter possibility using a larger set of Ribo-seq data of good quality. The correlation between CDS coverage in PhyloSET genes and their Ribo-seq ranking below the top 500 suggests that lower Ribo-seq rankings still correspond to genuine translation. We further tested this possibility by exploring the predictive value of Ribo-seq ranking in finding genes from PhyloSET. Indeed, even the sets of the lowest ranking genes contain more genes in PhyloSET than what is expected by chance (Fig. 3d). Furthermore, around a quarter of extensions in RiboSET and PhyloSET were confirmed with proteomics data available in Trips-Viz (Supplementary Data 1). Thus, while we cannot estimate the exact number of N-terminal extensions it is likely that it is much greater than 390. Thus we supplemented our study with extended RiboSET by choosing a less conservative threshold of 5000. The lack of phylogenetic support for the majority of non-AUG initiation events have been reported earlier[62]. However, that study did not discriminate between initiation resulting in extended proteoforms or translation of short ORFs. One may argue that unlike N-terminally extended proteoforms, products of short ORF translation are less likely to have phenotypic effects.

The lack of phylogenetic support for some of the extensions could be explained simply by the technical limitations of our phylogenetic approach which is based on the genomic alignment that is subjected to sequencing errors and misalignments. In addition, the PhyloCSF score could be reduced by genuine Loss of Function events in some of the lineages. Nonetheless, the number of such cases is unlikely to be very high, while the number of genes in RiboSETs greatly exceeds those in PhyloSET. So what is the reason for the lack of phylogenetics support for most N-terminal extensions detectable with Ribo-seq?

First, N-termini are often processed[63,64], thus it may not matter where exactly initiation occurs, at the annotated AUG codon or at the upstream non-AUG since the mature protein product is the same. The evolution of non-AUG starts upstream of AUG codons is expected to be neutral in this case and would not be detected with the PhyloCSF approach. A similar situation would be expected when N-terminal extensions do not alter the functional properties of the proteins. Nonetheless, it does not mean that all extensions occurring exclusively in RiboSET are inconsequential. First, they may shape the immune response by providing potential antigens. Second, mutations in the regions corresponding to N-terminal extensions may have different effects from those occurring in non-coding parts of 5′ leaders. For example, an introduction of an in-frame stop codon would result in an uORF that would inhibit the translation of CDS. It is also conceivable that in a number of cases the evolutionary selection was not detected with our approach. Some extensions may be very recent, so there is an insufficient number of substitutions in the narrow phylogenetic group in which they exist, some may even be uniquely human. The analysis of the data in aggregated databases of personal genomes, such as gnomAD may eventually reveal evolutionary selection in such cases. Also, we may have failed to detect evolutionary selection in those cases where it acts only at a very short region, below 50 codons upstream that we used in this study.

Of note, a positive PhyloCSF score upstream of annotated starts may have a different reason than a non-AUG extension. There is a possibility that the positive PhyloCSF signal might be explained by the remnant of CDS which was truncated in humans by nonsense mutation. For instance, in *LRP5L*, nonsense mutation happened in the ancestor of human and chimpanzee thus possibly disrupting a CDS present in other mammals (Supplementary Fig. 13). Therefore the next in-frame AUG which was internal became a new start in humans. Nevertheless, it does not exclude the possibility that human-specific non-AUG extensions can exist and be translated under certain conditions. There is also a need for acknowledgement of a limitation of genome annotations. For example, 5′ends of transcript may be incomplete. For the *HES3* gene, in GENCODE v38 there is an upstream AUG which has a clearly Ribo-seq stalling signal according to GWIPs-viz and it is well-conserved in mammals. However, in GENCODE v25 (that we used) 5′ends of *HES3* transcripts are shorter and do not contain such AUG thus making it impossible to detect such AUG.

Interestingly, one of the possible functions of non-AUG extension could be alternative compartmentalisation. GO analysis has shown that there are plenty of common genes between subcellular localisation terms. It can be interpreted as there are multiple subcellular localisations for gene products which might be facilitated by extensions. Also, data from HPA where 167 genes can be found in multiple compartments also leads to an idea that such distinct locations may correspond to proteoforms with alternative N-termini. This led us to applying tools for predicting signal peptides, mitochondrial presequences and transmembrane domains which revealed that there are genes with localisation signals residing within their non-AUG extensions. When comparing to UntranslSET—genes with no translation upstream in theoretical NTEs, we found no enrichment in predicted localisation signals in general (though we found no TM helices for genes in UntranslSET unlike RiboSET and PhyloSET) which suggests that there is no strong association between alternative localisation signals and non-AUG extensions. Nevertheless, it does not mean that at least some of the predicted signals might be genuine and some extensions indeed carry alternative compartmentalisation signals. Of note, it was reported that a significant fraction of random sequences can encode secretion signals as is shown for yeast invertase[65].

As a result of these study, we reannotated 30 human genes and also provided information on the likely functions of some of these extensions in the differential compartmentalisation of short and long proteoforms. The number of annotated human and mouse non-AUG-initiated proteoforms will likely increase as manual GENCODE curation progresses. The substantial number of pathogenic and likely pathogenic variants from ClinVar overlapping with predicted non-AUG extensions and the fact that novel annotations are found within genes of clear medical relevance, such as *JUN*, it is reasonable to assume that such extensions will hold biological information that will prove to be of profound importance in genomic science and medicine.

## Methods

### A pipeline for detection of non-AUG proteoforms using evolutionary signatures

We obtained 94,359 human protein-coding mRNA sequences from GENCODE v25 (GRCh38.p7), this particular version was chosen because of the available processed Ribo-seq and proteomics data available in the Trips-Viz browser[39]. For each transcript starting from an annotated AUG codon we moved along the transcript in the 5′ direction by three nucleotides until an in-frame STOP codon was reached. If there was no in-frame STOP codon, we took the first in-frame position on a transcript. We termed this sequence primary (or theoretical) extension. For assessing PhyloCSF score[38] using 120 mammals alignment in.maf format[66] we took 50 codons upstream of the annotated AUG (or less if the length of the primary extension is shorter, genes where theoretical extension is less than 20 codons are

discarded). MafExtract from CESAR2.0[67] coupled with PHAST v1.5[68] and custom Python scripts were used to extract multiple sequence alignments for 50 codons of N-termini. Transcripts with a positive PhyloCSF score for 50 codons-long or shorter N-termini (3058 genes and 5417 transcripts) were selected as candidates for further analysis. PhyloCSF v1.0.1 was used for the calculation of score[38].

We excluded transcripts for which theoretical N-terminal extensions have any overlaps with coding exons in the same strand from GENCODE v25 release (GRCh38.p7), GENCODE v35 release (GRCh38.p13), RefSeq (July 1, 2020; GRCh38.p13, 109.20200815). Briefly, for GENCODE annotations we extracted transcriptomic coordinates of CDS from protein-coding fasta files (gencode.v25.pc_transcripts.fa, gencode.v35.pc_transcripts.fa) and transformed to genomic coordinates using pmapFromTranscripts from the GenomicFeatures package in R (v3.6.1). For RefSeq annotations coding exons were extracted from GRCh38_latest_genomic.gff. Intersections were identified using bedtools intersect[69] (v2.29.2).

### Translating ORF predictor in Trips-Viz
We developed and first described an algorithm allowing the prediction of translated ORFs using Ribo-seq data (translating ORF predictor). More details are in the Supplementary Methods file.

### Detection of translated regions in Trips-Viz
We used the translating ORF predictor in Trips-Viz using Ribo-seq data (elongating and initiating ribosome profiling) from 13 studies (152 samples) GSE62247[70], GSE114794[71], GSE79664[72], GSE51584[73], GSE94460[74], GSE73136[75], GSE87328[76], GSE64962[77], GSE65885[78], GSE56887[79], GSE70211 and GSE79392[80], GSE77401[81], GSE58207[82] and four studies of proteomics data PXD004452[83], PXD002395[84], PXD002082[85] and PXD002815[86]. These data were selected from thousands of the datasets processed in Trips-viz based on high (>0.5) triplet periodicity score[39] (shown in Supplementary Data 8).

In the publicly available version of Trips-viz only AUG, CUG and GUG-initiated extensions are available to retrieve, so additional non-cognate starts including UUG, AUA, AUU, AUC, ACG, AGG and AAG were supplied into the in-house Trips-viz version. The top-500 ranked extensions were used to construct RiboSET. The top-5000 ranked extensions were used to construct RiboSET_ext.

### Set with untranslated upstream regions (UntranslSET)
We build UntranslSET—set of genes with no translation upstream following several rules: (1) transcripts have non-truncated CDS; (2) theoretical extension do not overlap with coding exons of other transcripts; (3) theoretical extension length is at least 20 codons; (4) neither transcript nor gene is present in the entire Trips-viz predicted set (all ~10,000 genes); (5) fraction of covered positions in theoretical extension is ≤0.0005; (6) CDS coverage (the number of in-frame reads divided by number of codons in CDS) ≥1; (7) fraction of coverage positions in CDS ≥0.3. We ended up with 384 genes (384 transcripts: 1 transcript per gene with the highest CDS coverage).

### Characterisation of predicted non-AUG starts in RiboSET
TIS scores were extracted from ref. 42. For RNA secondary structure analysis 400 genes initiating with AUG codon with the highest TIS scores and the lowest TIS scores were selected for comparison from RiboSET. Free Gibbs energy (dG) was calculated using RNAfold 2.4.17[87] within a sliding window of 22nt at 1nt step in the region starting from 10nt upstream and up to 100nt downstream of the start codon (averaged by genes with a 95% confidence interval using Python 3.7.3 package statsmodels v 0.10.1). Multiple sequence alignment was retrieved by MAFFT v7.310[88] and plotted by ggmsa_1.0.0, R v3.6.1. The sequence logo was built by ggseqlogo_0.1, R v3.6.1. Data and all other plots were analysed by pandas 1.2.1 and drawn using matplotlib 3.2.1, Python 3.7.3. GO enrichment analysis on RiboSET and PhyloSET genes

was performed using GOATOOLS, v1.1.6[89] with GO-basic version of database[90] and GENCODE v25 protein-coding genes as reference set, significant terms were selected based on adjusted *p* value <0.05 (after using fdr multiple testing correction).

As a source of experimentally supported protein localisation The Human Protein Atlas was used. In order to predict signal peptides we used SignalP 5.0[54], for identification of mitochondrial presequences we applied TargetP 2.0[56] and for extraction of TM domains DeepTMHMM[91] were employed.

We utilised variants from ClinVar (clinvar_20221015.vcf.gz) and overlapped them with predicted extensions (RiboSET_ext, PhyloSET) using 'bedtools intersect'.

### Statistical tests
All statistical tests (Kolmogorov–Smirnov test, Mann–Whitney *U*-rank test, Spearman correlation test, hypergeometric test, Fisher exact test and Benjamini–Hochberg multiple testing correction) were performed with python and R.

### Reporting summary
Further information on research design is available in the Nature Portfolio Reporting Summary linked to this article.

## Data availability
All data generated during this study are included in this published article and its Supplementary Information files. The datasets analysed during the current study are available in GEO under accession numbers: GSE62247[70], GSE114794[71], GSE79664[72], GSE51584[73], GSE94460[74], GSE73136[75], GSE87328[76], GSE64962[77], GSE65885[78], GSE56887[79], GSE70211, GSE79392[80], GSE77401[81], GSE58207[82], and four studies of proteomics data the ProteomeXchange PXD004452[83], PXD002395[84], PXD002082[85] and PXD002815[86]. We used GENCODE v25 (GRCh38.p7, https://www.gencodegenes.org/human/release_25.html) fasta and gtf files and GENCODE v35 (GRCh38.p13, https://www.gencodegenes.org/human/release_35.html) fastq and gtf files; RefSeq fasta and gtf files (July 1, 2020; GRCh38.p13, 109.20200815, https://ftp.ncbi.nlm.nih.gov/refseq/H_sapiens/). Source data are provided with this paper.

## Code availability
Trips-viz ORF predictor (v.1.0) can be found here: https://github.com/skiniry/Trips-Viz orfquery_routes.py (https://doi.org/10.5281/zenodo.7390032). Custom code can be found at https://github.com/triasteran/nonAUG_manuscript/tree/main/jupyter_notebooks (https://doi.org/10.5281/zenodo.7390032).

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

## Acknowledgements

We are grateful to Irwin Jungreis and Manolis Kellis at CSAIL (MIT) for supplying PhyloCSF parameters for 120-way mammalian genomic alignment and making CodAlignView available to us. This work was supported by Science Foundation Ireland grant (20/FFP-A/8929) and SFI-HRB-Wellcome Trust Biomedical Research Partnership (Investigator Award in Science) [210692/Z/18/Z] to P.V.B; Science Foundation Ireland Centre for Research Training in Genomics Data Science (18/CRT/6214 to A.D.F.) and Russian Science Foundation (19-14-00152) to D.E.A. S.J.K. wishes to acknowledge support from the Irish Research Council. J.M.M. is supported by the National Human Genome Research Institute of the National Institutes of Health under award number 2U41HG007234 and the European Molecular Biology Laboratory. The content is solely the responsibility of the authors and does not necessarily represent the official views of the National Institutes of Health. Ensembl is a registered trademark of EMBL.

## Author contributions

P.V.B. conceived the work, acquired funding and supervised the study. A.D.F. performed most of the computational analyses, prepared the first draft of the manuscript and all data figures. S.J.K. implemented the ranking algorithm for the identification of translated non-AUGs in Trips-

Viz. J.M.M. evaluated the suitability of N-terminal extension for relevant GENCODE reannotations. D.E.A. contributed to the review of the manuscript draft and the conceptualisation of the study. All authors participated in the interpretation of the data and editing of the final version of the manuscript.

## Competing interests

P.V.B. is a co-founder and a shareholder of Eirna Bio. The remaining authors declare no competing interests.
