## [Peer Review File · Nature Communications]

Thousands of human non-AUG extended proteoforms lack evidence of evolutionary selection among mammalsEditorial Note: Parts of this Peer Review File have been redacted as indicated to remove third-party material where no permission to publish could be obtained.

REVIEWER COMMENTS

Reviewer #1 (Remarks to the Author):

In this manuscript, Fedorova and the co-authors used two approaches to identify non-AUG initiated proteoforms. The first is phyloCSF, in which 60 candidate genes were identified based on genomic data; while the second, Trips-Viz based on Ribo-seq and proteomics data, found 392 candidate genes. However, only 8 genes are common between the two methods. By comparing their dataset with GENCODE v35, less than half annotated non-AUG proteoforms were validated. They further studied the properties of predicted non-AUG initiation codons, including the distribution of start codon types, TIS sequence logo and frequency, mRNA secondary structure, and domains. Finally, they updated 11 genes about the non-AUG initiated proteoforms.

Although the authors claim they “develop” the approaches for identifying of non-AUG translation initiation, the methods/tools used are not developed in the study. The authors did not generate any high-throughput data, they thus did not produce useful resources yet. Although they analyzed 11 datasets, the data is still small relative to tons of existed Ribo-seq data and proteomics data, which cannot provide an impressive landscape of non-AUG initiated proteoforms. More importantly, an interesting story, such as a genomic/genetical pattern or mechanism, should be proposed, but it seems to be lack. Taking together, this research needs to be improved on the novelty and workload, and does not fulfil the publication standard of Nature Communications at its current stage.

Some concerns:

The first problem is the title “Thousands of human non-AUG extended proteoforms lack evidence of evolutionary selection among mammals”. In fact, the main work or goal of this study is the identification of some non-AUG initiated proteoforms, which is far from the content in the title. The authors need to apply as many ways as possible to validate the title’s content (especially for the “lack evidence”), but they did not. The authors seem to be deliberately overstating their research to draw attentions to readers.

In addition, non-AUG extended proteoforms are evolutionarily un-conserved is already known, because most of them are translational molecular error that has been reported before (Mol Biol Evol. 2020;37:2015–28). Even the authors tried their best to validate this focus, it lacks novelty.

There are some important datasets this study did not collect and use. For translation initiation, the high throughput sequencing method, GTI/QTI-seq (Nature methods, 12(2), 147-153), can directly catch the initiation codon, level and position. Thus, it is more direct and useful than Ribo-seq. I did not see how this study include or analyzes this kind of data.

The authors did comparisons, including between the two methods and between this study and previous study, but the overlap part is always small, which makes people suspect the reliability and rationality of the data or methods used. As the main aim in this study is not on the comparison, the authors should not take much effort on the comparison.

Reviewer #2 (Remarks to the Author):

The manuscript by Fedorova et al. deals with the interesting question of how many and which genes might give rise to additional proteoforms due to translation initiation at non-AUG start codons. It is a purely bioinformatic investigation of pre-existing data. The introduction is concise, knowledgeable, cites the relevant literature and leads to main objective of the study. It is a little unusual to have a figure in the introduction and there could have been the conventional final short paragraph to state: ‘Here we show...’ The results section is quite clearly laid out, although at times it contains quite a lot of ‘introduction-type material’, e.g., the explanations of stop codon readthrough and frameshifting. The work appears to have been competently done and the conclusions are supported by evidence in the

figures and other (supplementary) material. It is altogether a valiant exploration of the prevalence of yet undiscovered N-terminally extended CDS. The work provides a better understanding of the underlying mechanisms, it outlines some of the functional consequences of N-terminal extensions in general and for individual examples, and there is the clear expectation that it will contribute to database updates with useful information on new cases.

The figures are quite well laid out and informative. I saw several panels lacking a y-axis label, e.g., bottom panels in Figure 3 A and B, Figure 4 E and H (truncated), Figure 5, and most of the supplemental figures. In most cases, the legend will have the relevant information but I still think it would be useful to label the graphs directly.

Reviewer #3 (Remarks to the Author):

This manuscript compares unannotated N-terminal extensions in human transcripts detected by two complementary approaches: experimental evidence of translation from published ribosome profiling, and protein-level conservation by evolutionary analysis of mammalian genomes. Experimental evidence of upstream translation is far more common than conserved protein composition; overlap between these classes is limited, but contains characterized examples of N-terminal extensions. The possibility that alternative upstream isoforms lead to different localization is explored, although it was unclear whether there were patterns of enrichment. In many cases, upstream translation is driven by initiation at non-AUG codons, and the initiation sites seen in ribosome profiling data correspond with mRNA sequence features reported to enhance translation.

This manuscript provides a comprehensive analysis of protein-level consequences of upstream translation. This provides an advance relative to other studies in the field, and fills an important role in genome annotation. However, many results are presented in a disorganized fashion—it is sometimes hard to determine which data set is being discussed for a particular analysis, and hard to know what fraction of genes are affected by a phenomenon or whether this represents a significant enrichment. This contrasts surprisingly with the clear presentation of the basic phylogenetic and translational data sets. More specific concerns include:

1. Is there a difference in the PhyloCSF score distributions between transcripts with upstream translation ("RiboSET") and those without upstream translation (perhaps matched for CDS expression, which affects detectability of translation, in light of Fig 3C)?
2. The analysis in Fig 3D seems to argue that the top-500 cutoff used for the RiboSET may be too conservative?
3. Are the levels of overlap between Ivanov et al., 2011 (ref. 21) and PhyloSET, RiboSET statistically significant? Is there a reason that the current PhyloSET and the Ivanov analysis share so little overlap, given that both rely on evolutionary comparison within mammals?
4. In analyzing re-identification of non-AUG extensions (Fig. 3F and Table S3D), how does the PhyloCSF score distribution of the 24 annotated non-AUG extensions compare with the PhyloCSF score distributions of other comparable transcripts?
5. Are the overlaps between Van Damme et al. and Yeom et al. data sets with PhyloSET and RiboSET significant?
6. It is reported that, "PhyloCSF score distribution for RiboSET genes is clearly skewed towards large negative values arguing for massive non-conserved non-AUG translation in 5' leaders (Fig.4E)." This analysis compares RiboSET with PhyloSET, chosen based on PhyloCSF score. A more meaningful comparison would be the analysis of PhyloCSF between transcripts with and without upstream

translation, as described in (1).

7. The localization of proteins as reported in Fig. 4F and 4G seem largely irrelevant to the question of whether alternative isoforms drive multiple localizations. Do PhyloSET or RiboSET proteins have multiple localizations more often than proteins with no evidence for alternative translation?

8. Likewise, the key question for upstream extensions seems to be whether SignalP, TargetP, or TMHMM detection is more common in translated/conserved upstream extensions than in equivalent, non-coding 5' UTR sequence. Notably a significant fraction of random sequences encode secretion signals ([doi://10.1126/science.3541205](https://doi.org/10.1126/science.3541205)).

9. In the discussion, it is written about PhyloSET and RiboSET that "The large overlap between the sets is not expected." This seems at odds with the general conclusions of the paper and the rest of the paragraph—there is not a large overlap between these sets. Perhaps an error in editing? Is it meant that, "A large overlap between the sets would not have been expected."?

REVIEWER COMMENTS

Reviewer #1 (Remarks to the Author):

In this manuscript, Fedorova and the co-authors used two approaches to identify non-AUG initiated proteoforms. The first is phyloCSF, in which 60 candidate genes were identified based on genomic data; while the second, Trips-Viz based on Ribo-seq and proteomics data, found 392 candidate genes. However, only 8 genes are common between the two methods. By comparing their dataset with GENCODE v35, less than half annotated non-AUG proteoforms were validated. They further studied the properties of predicted non-AUG initiation codons, including the distribution of start codon types, TIS sequence logo and frequency, mRNA secondary structure, and domains. Finally, they updated 11 genes about the non-AUG initiated proteoforms.

We kindly thank the reviewer for their feedback.

We would like to note that we did not compare our datasets with GENCODE v35. We excluded genes that have already been annotated in GENCODE v35 from RiboSET and PhyloSET. Instead, we tested whether we were able to re-identify genes with already annotated non-AUG extensions in GENCODE v35 from the Ivanov et al 2011 study using the PhyloCSF score if we moved the start to the originally annotated AUG. Indeed, only 11 genes in the original study have a positive PhyloCSF score and we discuss the possible explanations of such an overlap in the Discussion. Briefly, in Ivanov et al only human-mouse paired alignments were used unlike the 120-mammals alignment that we used for PhyloCSF score calculation. Also we calculated the score for 50 upstream codons while conserved extension could be way shorter thus extra upstream codons could add a large negative value to the final score.

We updated annotations for 11 genes, however we expect many more to be annotated in the future, for example SLC25A32 was annotated as such during the revision of this paper.

Although the authors claim they “develop” the approaches for identifying of non-AUG translation initiation, the methods/tools used are not developed in the study. The authors did not generate any high-throughput data, they thus did not produce useful resources yet.

We thank the reviewer for bringing our attention to the fact that we did not sufficiently highlight the new methods developed in our work.

First of all, we described the algorithm which allows for prediction of translated ORFs using aggregated Ribo-seq data. While we used it in this manuscript only for identification of non-AUG N-terminal extensions, its implementation in the Trips-Viz browser can be used for detecting any kind of unannotated translation including upstream, downstream, nested and overlapping ORFs. The detailed description of the algorithm is in Supplementary Methods. We also provided a link to the code at github - https://github.com/skiniry/Trips-Viz/blob/master/orfquery_routes.py.

We apologise for not explicitly stating this in the original version of manuscript and thank the Reviewer for pointing this out. We have now added the following sentence to the corresponding Results:

We have developed a translating ORF predictor, an algorithm for detecting translated ORFs using aggregated Ribo-seq data (**Supplementary Methods**). The implementation of this algorithm is now available in the Trips-Viz browser.

And also modified the abstract:

Here we analysed a large number of publicly available ribo-seq datasets to identify novel, previously uncharacterised non-AUG proteoforms using Trips-Viz implementation of a novel algorithm for detecting translated ORFs.

In addition to the general purpose of translated ORF detection, specifically for the purpose of this work, we designed and described a pipeline that can detect non-AUG N-terminal extensions using 2 orthogonal approaches. This pipeline includes extracting upstream regions of canonically annotated transcript models, deriving genomic coordinates of theoretical N-terminal extensions (NTEs), filtering overlaps with known coding regions, stitching together exons of NTEs and calculating PhyloCSF scores. Another branch includes applying the aforementioned ORF-predictor on transcript models. To our knowledge, this pipeline has not been used for detection of non-AUG initiated proteoforms anywhere else.

Although they analyzed 11 datasets, the data is still small relative to tons of existed Ribo-seq data and proteomics data, which cannot provide an impressive landscape of non-AUG initiated proteoforms.

Again, we thank the reviewer for drawing our attention to this matter as the text in the previous version somewhat undermined the scale of our effort.

It is important to point out that for this analysis we started with over 1000 ribosome profiling datasets from 48 published papers which have been processed and are currently available in Trips-Viz. However, for accurate detection of N-extensions it is crucial to detect the correct translated frame. Thus, we analysed all these datasets to determine the strength of triplet periodicity. We chose only 152 datasets from 13 research publications that produced ribo-seq data with high periodicity score (more than 0.5). Perhaps misleadingly we also used the number of research papers used as the number of datasets. This is now corrected, we also added an explanatory statement in the “Detection of translated regions in Trips-Viz” section of Methods explaining this.

We provided periodicity scores for each study in Supplementary table 8.

In relation to the entire landscape of N-terminal extensions in human genes, this would require a generation of ribosome profiling data from virtually all human cells, which is currently not feasible for technical and ethical reasons.

We emphasised that the aim of our study was not to exhaustively detect all existing non-AUG initiated proteoforms, but to develop methodology for this and explore the relationship between the two approaches. Unexpectedly we found that the number of N-extensions detectable with ribosome profiling obtained in a limited number of human cells by far exceeds that detectable with current phylogenetic approaches. This points to the existence of thousands of translated non-AUG initiated proteoforms with no evidence of purifying selection across mammals. We made the following change in the discussion to clarify this:

We did not aim to exhaustively detect all existing non-AUG initiated proteoforms but rather compare two approaches and the relationship between translated N-extensions detected with ribosome profiling and evidence of protein coding evolution in the corresponding genes.

More importantly, an interesting story, such as a genomic/genetical pattern or mechanism, should be proposed, but it seems to be lack. Taking together, this research needs to be improved on the novelty and workload, and does not fulfil the publication standard of Nature Communications at its current stage.

The mechanisms involved in non-AUG initiation in general are reviewed e.g. in <http://www.genesdev.org/cgi/doi/10.1101/gad.305250.117>, <https://doi.org/10.1186/s13059-022-02674-2> and briefly described in our introduction. It is not clear to us what the reviewer meant by genomic/genetical pattern.

Some concerns:

The first problem is the title “Thousands of human non-AUG extended proteoforms lack evidence of evolutionary selection among mammals”. In fact, the main work or goal of this study is the identification of some non-AUG initiated proteoforms, which is far from the content in the title. The authors need to apply as many ways as possible to validate the title’s content (especially for the “lack evidence”), but they did not. The authors seem to be deliberately overstating their research to draw attentions to readers.

The title indeed focuses on an important outcome of our study that has not been emphasised enough in the manuscript.

Describing features of extensions, we focused on only hundreds of detected genes taking a very conservative threshold (**Fig.3d**) initially and we argue that it is better to use a high-confident set which is more likely devoid of false positives.

The relaxed thresholds lead to detection of thousands of translated non-AUG extensions enriched with those from PhyloSET up to thresholds (6000-7000 rank) see **Fig. 3d**. Most lack phylogenetic support (considering that PhyloSET has only 60 genes). We have now generated RiboSET_ext - a set of 3451 genes with predicted translated extensions based on a relaxed threshold (rank=5000) which is now provided in **Supplementary table 2**. We also replicated the analysis of non-AUG initiation features for RiboSET_ext (**Supplementary Fig.10**), it is very similar to what we observed in RiboSET, so it does not contradict with findings that we discovered using the strict threshold.

We claim that a negative PhyloCSF score based on 120-mammals alignment calculated for extensions in both RiboSET (**Fig.3h**) and RiboSET_ext (**Supplementary Fig.10a**) provides us with evidence of a lack of evolutionary selection among mammals. We added following sentences in ‘Characterisation of genes with predicted non-AUG initiation’ section:

More importantly, the PhyloCSF score of upstream regions of RiboSET_ext is also skewed towards large negative values providing us with evidence of no evolutionary selection among mammals for thousands of translated non-AUG extensions (**Supplementary Fig.10a**).

Perhaps, the reviewer misinterpreted our claim as “evidence of a lack of selection” in general. This is not the case. We cannot exclude a possibility of evolutionary selection acting on these genes within a narrow phylogenetic group such as primates, hominids or even humans only and we discuss such a possibility in the Discussion.

In addition, non-AUG extended proteoforms are evolutionarily un-conserved is already known, because most of them are translational molecular error that has been reported before (Mol Biol Evol. 2020;37:2015–28). Even the authors tried their best to validate this focus, it lacks novelty.

1. We disagree with placing an equal sign between the lack of selection within a specific phylogenetic group to “a molecular error”. First the selection may occur in a narrower phylogenetic group as argued above. Second, most of the evolution occurring in species with a small effective population size such as mammals is neutral. This was established through seminal works by Motoo Kimura in the 70s and 80s. Just as an example, consider olfactory receptors in humans. While most are not selected as evident from the large number of corresponding pseudogenes in our genome, we wouldn’t term expression of an active olfactory receptor as a “molecular error”. If we accept this point of view we would need to refer to whole classes of non-conserved biological molecules such as lncRNAs as “molecular errors”. On the other hand, all evolution may be described as a result of “molecular errors” as errors are a major driver of evolution.

2. Furthermore Mol Biol Evol. 2020;37:2015–28, focuses on non-AUG initiation irrespective of its functional consequences, e.g. whether they result in uORFs or N-extensions. Most importantly their study was limited to QTI-seq data. While QTI-seq data are useful and we used it in our own study when trying to predict locations of potential non-AUG codons, these data are very noisy and subject to technical errors.

First QTI-seq has been shown to enrich ribosomes not only at initiating codons, but also at two specific codons during elongation see <http://www.rnajournal.org/cgi/doi/10.1261/rna.049908.115> and Dmitriev SE, Akulich KA, Andreev DE, Terenin IM, Shastky IN. The peculiar mode of translation elongation inhibition by antitumor drug harringtonin. FEBS J. 2013;280:51.

Furthermore QTI-seq is very narrow, reads supporting initiation at a particular codon map to a corresponding single location. Assessing translation based on a single translation event is highly unreliable due to sequencing biases, mappability and other issues that we discussed in detail in the first section of our recent review on the topic (doi: 10.1002/wrna.1577).

The approach that we used here is by far more reliable and robust because it relies on reads aligned from a large region of mRNA and multiple studies, thus reducing the effect of artefacts and biases of individual locations and studies.

Of note, the MBE study did not rely on phylogenetic approaches to infer non-adaptiveness of extensions encoded by the non-AUG starts; their claims are based mostly on comparisons between translational amount of genes and TIS diversity.

There are some important datasets this study did not collect and use. For translation initiation, the high throughput sequencing method, GTI/QTI-seq (Nature methods, 12(2), 147-153), can directly catch the initiation codon, level and position. Thus, it is more direct and useful than Ribo-seq. I did not see how this study include or analyzes this kind of data.

As we discussed above, while useful, GTI/QTI-seq is very noisy and predictions made based on it are unlikely to be robust since they would rely on very narrow mapping. We did use it however in manual assessments of exact potential locations of non-AUG starts during manual evaluation of the selected genes.

The authors did comparisons, including between the two methods and between this study and previous study, but the overlap part is always small, which makes people suspect the reliability and rationality of the data or methods used. As the main aim in this study is not on the comparison, the authors should not take much effort on the comparison.

We extensively discussed why overlap between phylogenetic and ribo-seq approaches is smaller. It is one of the most important outcomes of our study.

The small overlap with the previous study is due to various reasons that we thoroughly explained in Results. E.g. in comparison with Ivanov et.al there are multiple reasons including differences in RefSeq and GENCODE; a considerable fraction of originally discovered non-AUG genes have been already annotated in the recent GENCODE version. Also, we tried to rediscover genes that have not yet been annotated.

Back then, in 2011, massive multiple sequence codon alignments such as 100 vertebrates or 120 mammals were not available and the analysis was based solely on human-mouse paired alignment which is likely to have less information in comparison to 120-mammals. Also, there was only a single Ribo-seq dataset available for humans and we used it purely for illustrative purposes. It was not used for detection of N-extensions.

Reviewer #2 (Remarks to the Author):

The manuscript by Fedorova et al. deals with the interesting question of how many and which genes might give rise to additional proteoforms due to translation initiation at non-AUG start codons. It is a purely bioinformatic investigation of pre-existing data. The introduction is concise, knowledgeable, cites the relevant literature and leads to main objective of the study. It is a little unusual to have a figure in the introduction and there could have been the conventional final short paragraph to state: 'Here we show...'

We thank the reviewer for their feedback. It is indeed not very common to have a figure in the Introduction, but we understand that it is not against the journal policy and we hope it helps for clarity.

As suggested by the Reviewer we added a final short paragraph to the end of Introduction:

Here we analysed multiple publicly available ribosome profiling datasets and genomic alignments of 120 mammals to detect translation and protein coding evolution in sequences encoding potential non-AUG extensions. Our analysis suggests thousands of such

extensions in human proteoforms do not exhibit signatures of protein coding evolution typical for proteins in mammals.

The results section is quite clearly laid out, although at times it contains quite a lot of 'introduction-type material', e.g., the explanations of stop codon readthrough and frameshifting.

Indeed, it makes the manuscript unnecessarily lengthy, so we removed stop codon readthrough and frameshifting stories from the Results section.

The work appears to have been competently done and the conclusions are supported by evidence in the figures and other (supplementary) material. It is altogether a valiant exploration of the prevalence of yet undiscovered N-terminally extended CDS. The work provides a better understanding of the underlying mechanisms, it outlines some of the functional consequences of N-terminal extensions in general and for individual examples, and there is the clear expectation that it will contribute to database updates with useful information on new cases.

We thank the reviewer for their positive feedback.

The figures are quite well laid out and informative. I saw several panels lacking a y-axis label, e.g., bottom panels in Figure 3 A and B, Figure 4 E and H (truncated), Figure 5, and most of the supplemental figures. In most cases, the legend will have the relevant information but I still think it would be useful to label the graphs directly.

Thank you for noticing this. We carefully went through plots and added y-axis labels for better clarity.

Reviewer #3 (Remarks to the Author):

This manuscript compares unannotated N-terminal extensions in human transcripts detected by two complementary approaches: experimental evidence of translation from published ribosome profiling, and protein-level conservation by evolutionary analysis of mammalian genomes. Experimental evidence of upstream translation is far more common than conserved protein composition; overlap between these classes is limited, but contains characterized examples of N-terminal extensions. The possibility that alternative upstream isoforms lead to different localization is explored, although it was unclear whether there were patterns of enrichment. In many cases, upstream translation is driven by initiation at non-AUG codons, and the initiation sites seen in ribosome profiling data correspond with mRNA sequence features reported to enhance translation.

We kindly thank the reviewer for their feedback. Patterns of enrichment for different localisations are addressed in the next comments.

This manuscript provides a comprehensive analysis of protein-level consequences of upstream translation. This provides an advance relative to other studies in the field, and fills an important role in genome annotation. However, many results are

presented in a disorganized fashion—it is sometimes hard to determine which data set is being discussed for a particular analysis, and hard to know what fraction of genes are affected by a phenomenon or whether this represents a significant enrichment. This contrasts surprisingly with the clear presentation of the basic phylogenetic and translomic data sets.

In terms of the sets obtained in this study we had only two sets PhyloSET and RiboSET to which we added two for this revision RiboSET_ext (RiboSET under rank 5000) and UntranslSET (see below). We presume that the Reviewer here refers to the comparison with previously published datasets which is described in “Comparison with previously identified non-AUG proteoforms”. The situation is indeed complicated as we found many different reasons for the discrepancies between these published datasets and sets produced in our study and we grouped the genes accordingly. To help readability we now gave explicit names to these sets of genes e.g. genes from Ivanov et al study (2011) now explicitly divided into 3 groups: ‘ann_24’, ‘un_28’ and ‘diff_utr’, where ‘ann_24’ represent 24 genes which have been already annotated in GENCODE v35 and ‘un_28’ is set of 28 genes which have not yet been annotated in GENCODE v35. ‘Diff_utr’ are the remaining 7 genes that have differing 5'UTR sequences in GENCODE in comparison to RefSeq. Those names are used in **Fig.3g,f**.

More specific concerns include:

1. Is there a difference in the PhyloCSF score distributions between transcripts with upstream translation ("RiboSET") and those without upstream translation (perhaps matched for CDS expression, which affects detectability of translation, in light of Fig 3C)?

This is indeed a very interesting question, thank you for suggesting it.

To answer this question, we derived a high-confident set of genes with no translation in theoretical extensions (regions upstream of annotated start codons up to the first encountered in-frame stop). We called this set ‘UntranslSET’, and it has 384 genes. We wanted to make sure that there is no translation upstream and there is a significant translation of CDS as well as a theoretical extension length of at least 20 codons - the same as for RiboSET and PhyloSET genes. All filters used for creating this set are described in Methods. Therefore, we ended up with a relatively small set.

We added the following to the Results:

Next, we compared PhyloCSF scores of upstream regions of RiboSET genes and genes with no translation upstream. In brief, we created a set called UntranslSET containing 384 genes with no translation upstream in theoretical N-terminal extension but with well translated CDS and a theoretical extension length of at least 20 codons (Methods). Interestingly, it turned out that both PhyloCSF score distribution of RiboSET and UntranslSET are skewed towards negative values (**Fig.3h**), however we can clearly see (Komogorov-Smirnov, p-value=2.23e-06) that PhyloCSF score of RiboSET genes is generally lower than that of UntranslSET. More importantly, the PhyloCSF score of upstream regions of RiboSET_ext is also skewed towards large negative values providing us with evidence of a lack of evolutionary selection among mammals for thousands of translated non-AUG extensions (**Supplementary Fig.10a**). Such a significant difference in PhyloCSF scores between UntranslSET and RiboSETs might be explained by the length of the theoretical extensions (**Supplementary Fig.10g,h**) - upstream regions in UntransSET are

shorter than in RiboSET (RiboSET_ext) therefore PhyloCSF score might be higher for them.

2. The analysis in Fig 3D seems to argue that the top-500 cutoff used for the RiboSET may be too conservative?

This is indeed a very conservative cutoff. We chose arbitrary rank 500 because this threshold is more likely to avoid false positives thus making RiboSET enriched in genuine non-AUG initiation. Therefore we carried all downstream analysis based on this small but very high quality dataset. Nevertheless, **Fig.3d** analysis shows that in fact there are likely to be many more genes with upstream non-AUG initiation: after reaching rank above 6000 the overlap is fully saturated. Based on this saturation mark, we decided to take the top-5000 genes, performed the same filtration as for the top-500 RiboSET and it resulted in 3451 genes which we provided in **Supplementary table 2**. This set is called RiboSET_ext. We also repeated the analysis of non-AUG associated features for RiboSET_ext (**Supplementary Fig.10**); it resulted in very similar patterns.

3. Are the levels of overlap between Ivanov et al., 2011 (ref. 21) and PhyloSET, RiboSET statistically significant? Is there a reason that the current PhyloSET and the Ivanov analysis share so little overlap, given that both rely on evolutionary comparison within mammals?

We have now added tests of overlap significance for comparisons between RiboSET, PhyloSET and Ivanov et al ('un_28' set), Van Damme et al., Yeom et al. data sets. We also performed multiple testing corrections (Benjamini-Hochberg) for derived p-values. We added the following:

Overlap between 'un_28' and PhyloSET is statistically significant in comparison to all protein-coding genes (hypergeom pv-adj=2.838e-06). Overlap between 'un_28' and RiboSET is not significant (hypergeom pv-adj=0.122).

The main reason for the small overlap is that over 30% of non-AUG extensions predicted in Ivanov et al work are already annotated as non-AUG initiated and thus were excluded from our analysis. We have now added a short explanation for the small overlap in the Results section as follows:

When compared to the previous study (Ivanov et al 2011, ref 21), we observed a statistically significant but nevertheless a small overlap (4 genes) with PhyloSET. One would expect that due to both studies using similar approaches (phylogenetic analysis in mammalian species), the overlap should be higher. This can be explained by multiple reasons including that 24 genes out of a total of 60 have already been annotated in GENCODE v35 thus were excluded from our analysis and 7 genes have unmatching sequences of 5'UTRs in comparison to RefSeq which was used in a previous study. Only 28 genes were left available for discovery. We used multiple sequence alignments of 120 mammals in contrast to Ivanov et al where only paired alignments of human and mouse were studied. It could explain why such small overlap was observed - the probability of upstream regions to be conserved in a wider range of species is generally lower than in just two species. Also, the length of the region we took for evaluation is 50-codons long which simply might be longer than the actual extension thus leading to a negative PhyloCSF score derived from excessive upstream non-conserved triplets not included in the actual extension. The same reason may explain why we could re-identify only half of the genes already annotated in GENCODE v35 using our phylogenetic approach. Nevertheless, PhyloCSF score of extensions in already

annotated genes from Ivanov et al study is significantly higher than in upstream regions of not yet annotated genes (**Fig. 3g**).

Besides, some of the non-AUG extensions described in Ivanov et al may be false positives. We are carefully revising such cases for a potential removal from the future GENCODE releases.

4. In analyzing re-identification of non-AUG extensions (Fig. 3F and Table S3D), how does the PhyloCSF score distribution of the 24 annotated non-AUG extensions compare with the PhyloCSF score distributions of other comparable transcripts?

We now added the PhyloCSF score distribution of upstream regions for 24 annotated genes (with starts moved downstream to AUGs) and compared it to the PhyloCSF scores of not yet annotated 28 genes (**Fig.3g**). As expected, the already annotated genes ('ann_24') have significantly higher PhyloCSF score (Mann-Whitney, p-value=0.0086) than not yet annotated ones ('un_28') which might explain why they were not prioritised for annotation.

5. Are the overlaps between Van Damme et al. and Yeom et al. data sets with PhyloSET and RiboSET significant?

We added the following lines:

Both overlaps between Van Damme et al study and PhyloSET or RiboSET are significant (hypergeom. pv-adj=0.0016 and 1.002e-07 correspondingly).

Overlap between Yeom L. et al study and RiboSET is significant (hypergeom. pv-adj=1.322e-07) in contrast to PhyloSET (hypergeom. pv-adj=0.403).

6. It is reported that, "PhyloCSF score distribution for RiboSET genes is clearly skewed towards large negative values arguing for massive non-conserved non-AUG translation in 5' leaders (Fig.4E)." This analysis compares RiboSET with PhyloSET, chosen based on PhyloCSF score. A more meaningful comparison would be the analysis of PhyloCSF between transcripts with and without upstream translation, as described in (1).

We addressed that while replying to comment 1 (**Fig.3h, Supplementary Fig.10a**).

7. The localization of proteins as reported in Fig. 4F and 4G seem largely irrelevant to the question of whether alternative isoforms drive multiple localizations. Do PhyloSET or RiboSET proteins have multiple localizations more often than proteins with no evidence for alternative translation?

This is a valid argument, thank you. We now removed Fig.4F and 4G plots. Instead, we replaced it with a bar chart comparing fractions of genes in PhyloSET, RiboSET and UntranslSET (genes having no evidence for upstream translation) having 1 or >=2 locations in the Human Protein Atlas. To supplement the visualisation, we also performed statistical enrichment tests (**Supplementary table 6**).

We added the following line in the Result section:

We found no significant enrichment in multiple localisation for genes in RiboSET or PhyloSET (and RiboSET_ext) in comparison to UntranslSET.

8. Likewise, the key question for upstream extensions seems to be whether SignalP, TargetP, or TMHMM detection is more common in translated/conserved upstream extensions than in equivalent, non-coding 5' UTR sequence. Notably a significant fraction of random sequences encode secretion signals (doi://10.1126/science.3541205).

This is also a valid argument. We added the following lines in the manuscript (section 'Characterisation of genes with predicted non-AUG initiation'; of note we substituted TMHMM with DeepTMHMM, a more recent version):

We tested whether SignalP, TargetP, and DeepTMHMM detection of compartmentalisation signals is more common in translated or conserved upstream extensions than in equivalent genes with no upstream translation (UntranslSET). Occurrence of mitochondrial presequence predicted with TargetP in RiboSET was shown to have a significant enrichment of presequences in comparison to vs UntranslSET (p-value=0.022) although after multiple testing correction the p-value did not retain significance. Also, no TM helices were found in UntranslSET unlike 2 and 2 genes in RiboSET and PhyloSET (12 genes in RiboSET_ext). For other predictors, there was no significant enrichment of upstream regions of RiboSET (RiboSET_ext) and PhyloSET genes in predicted localisation signals in comparison to genes with no detected translation (Fisher exact test, alternative='greater', p-adj > 0.05, **Supplementary table 6**).

'Discussion':

When comparing to UntranslSET - genes with no translation upstream in theoretical NTEs, we found no enrichment in predicted localisation signals in general (though we found no TM helices for genes in UntranslSET unlike RiboSET and PhyloSET) which suggests that there is no strong association between alternative localisation signals and non-AUG extensions. Nevertheless, it does not mean that at least some of the predicted signals might be genuine and some extensions indeed carry alternative compartmentalisation signals. Of note, it was reported that a significant fraction of random sequences can encode secretion signals as it is shown for yeast invertase⁵⁸.

Reference 58 is doi://10.1126/science.3541205.

We also removed that statement from the abstract (as there is no strong association): Despite the lack of detectable evolutionary selection, the biological significance of non-AUG proteoforms is supported by their association with multiple compartmentalizations of corresponding gene products as well by the presence of localisation signals within N-terminal extensions.

9. In the discussion, it is written about PhyloSET and RiboSET that "The large overlap between the sets is not expected." This seems at odds with the general conclusions of the paper and the rest of the paragraph—there is not a large overlap between these sets. Perhaps an error in editing? Is it meant that, "A large overlap between the sets would not have been expected."?

Thank you, we have changed the sentence to:

The large overlap between the sets would not have been expected.

REVIEWER COMMENTS

Reviewer #1 (Remarks to the Author):

In this manuscript, Fedorova and the co-authors used two approaches to identify non-AUG initiated proteoforms. The first is phyloCSF, in which 60 candidate genes were identified based on genomic data; while the second, Trips-Viz based on Ribo-seq and proteomics data, found 392 candidate genes. However, only 8 genes are common between the two methods. By comparing their dataset with GENCODE v35, less than half annotated non-AUG proteoforms were validated. They further studied the properties of predicted non-AUG initiation codons, including the distribution of start codon types, TIS sequence logo and frequency, mRNA secondary structure, and domains. Finally, they updated 11 genes about the non-AUG initiated proteoforms.

We kindly thank the reviewer for their feedback.

We would like to note that we did not compare our datasets with GENCODE v35. We excluded genes that have already been annotated in GENCODE v35 from RiboSET and PhyloSET. Instead, we tested whether we were able to re-identify genes with already annotated non-AUG extensions in GENCODE v35 from the Ivanov et al 2011 study using the PhyloCSF score if we moved the start to the originally annotated AUG. Indeed, only 11 genes in the original study have a positive PhyloCSF score and we discuss the possible explanations of such an overlap in the Discussion. Briefly, in Ivanov et al only human-mouse paired alignments were used unlike the 120-mammals alignment that we used for PhyloCSF score calculation. Also we calculated the score for 50 upstream codons while conserved extension could be way shorter thus extra upstream codons could add a large negative value to the final score.

We updated annotations for 11 genes, however we expect many more to be annotated in the future, for example SLC25A32 was annotated as such during the revision of this paper.

I was not arguing about their comparison, instead I was talking about the work's significance to the field as only 11 genes were updated.

Although the authors claim they “develop” the approaches for identifying of non-AUG translation initiation, the methods/tools used are not developed in the study. The authors did not generate any high-throughput data, they thus did not produce useful resources yet.

We thank the reviewer for bringing our attention to the fact that we did not sufficiently highlight the new methods developed in our work.

First of all, we described the algorithm which allows for prediction of translated ORFs using aggregated Ribo-seq data. While we used it in this manuscript only for identification of non-AUG N-terminal extensions, its implementation in the Trips-Viz browser can be used for detecting any kind of unannotated translation including upstream, downstream, nested and overlapping ORFs. The detailed description of the algorithm is in Supplementary Methods. We also provided a link to the code at github - https://github.com/skiniry/Trips-Viz/blob/master/orfquery_routes.py.

We apologise for not explicitly stating this in the original version of manuscript and thank the Reviewer for pointing this out. We have now added the following sentence to the corresponding Results:

We have developed a translating ORF predictor, an algorithm for detecting translated ORFs using aggregated Ribo-seq data (Supplementary Methods). The implementation of this algorithm is now available in the Trips-Viz browser.

And also modified the abstract:

Here we analysed a large number of publicly available ribo-seq datasets to identify novel, previously uncharacterised non-AUG proteoforms using Trips-Viz implementation of a novel algorithm for detecting translated ORFs.

In addition to the general purpose of translated ORF detection, specifically for the purpose of this work, we designed and described a pipeline that can detect non-AUG N-terminal extensions using 2 orthogonal approaches. This pipeline includes extracting upstream regions of canonically annotated transcript models, deriving genomic coordinates of theoretical N-terminal extensions (NTEs), filtering overlaps with known coding regions, stitching together exons of NTEs and calculating PhyloCSF scores. Another branch includes applying the aforementioned ORF-predictor on transcript models. To our knowledge, this pipeline has not been used for detection of non-AUG initiated proteoforms anywhere else.

They did not describe their new approach in the original manuscript, so that I had to point out the problem. In the revised version, they provided some details, but did not show its novelty relative to others. In addition, the reliability and advancement of their method is unknown.

Again, I was arguing about the work's significance to the field. They did not provide us method with novelty and did not generate original high-throughput useful data yet even in current revised version.

Although they analyzed 11 datasets, the data is still small relative to tons of existed Ribo-seq data and proteomics data, which cannot provide an impressive landscape of non-AUG initiated proteoforms.

Again, we thank the reviewer for drawing our attention to this matter as the text in the previous version somewhat undermined the scale of our effort.

It is important to point out that for this analysis we started with over 1000 ribosome profiling datasets from 48 published papers which have been processed and are currently available in Trips-Viz. However, for accurate detection of N-extensions it is crucial to detect the correct translated frame. Thus, we analysed all these datasets to determine the strength of triplet periodicity. We chose only 152 datasets from 13 research publications that produced ribo-seq data with high periodicity score (more than 0.5). Perhaps misleadingly we also used the number of research papers used as the number of datasets. This is now corrected, we also added an explanatory statement in the "Detection of translated regions in Trips-Viz" section of Methods explaining this.

We provided periodicity scores for each study in Supplementary table 8.

In relation to the entire landscape of N-terminal extensions in human genes, this would require a generation of ribosome profiling data from virtually all human cells, which is currently not feasible for technical and ethical reasons.

We emphasised that the aim of our study was not to exhaustively detect all existing non-AUG initiated proteoforms, but to develop methodology for this and explore the relationship between the two

approaches. Unexpectedly we found that the number of N-extensions detectable with ribosome profiling obtained in a limited number of human cells by far exceeds that detectable with current phylogenetic approaches. This points to the existence of thousands of translated non-AUG initiated proteoforms with no evidence of purifying selection across mammals. We made the following change in the discussion to clarify this:

We did not aim to exhaustively detect all existing non-AUG initiated proteoforms but rather compare two approaches and the relationship between translated N-extensions detected with ribosome profiling and evidence of protein coding evolution in the corresponding genes.

As the authors emphasized their aim of this study is to develop methodology, they need to compare with other methodologies to show the novelty and reliability, but they did not in both versions.

More importantly, an interesting story, such as a genomic/genetical pattern or mechanism, should be proposed, but it seems to be lack. Taking together, this research needs to be improved on the novelty and workload, and does not fulfil the publication standard of Nature Communications at its current stage.

The mechanisms involved in non-AUG initiation in general are reviewed e.g. in <http://www.genesdev.org/cgi/doi/10.1101/gad.305250.117>, <https://doi.org/10.1186/s13059-022-02674-2> and briefly described in our introduction. It is not clear to us what the reviewer meant by genomic/genetical pattern.

Again, I was arguing about their work's significance. One of the reasons is that their work failed to provide any new stories about the genomic or genetical pattern/principles in both previous and current versions.

Some concerns:

The first problem is the title "Thousands of human non-AUG extended proteoforms lack evidence of evolutionary selection among mammals". In fact, the main work or goal of this study is the identification of some non-AUG initiated proteoforms, which is far from the content in the title. The authors need to apply as many ways as possible to validate the title's content (especially for the "lack evidence"), but they did not. The authors seem to be deliberately overstating their research to draw attentions to readers.

The title indeed focuses on an important outcome of our study that has not been emphasised enough in the manuscript.

Describing features of extensions, we focused on only hundreds of detected genes taking a very conservative threshold (Fig.3d) initially and we argue that it is better to use a high-confident set which is more likely devoid of false positives.

The relaxed thresholds lead to detection of thousands of translated non-AUG extensions enriched with those from PhyloSET up to thresholds (6000-7000 rank) see Fig. 3d. Most lack phylogenetic support (considering that PhyloSET has only 60 genes). We have now generated RiboSET_ext - a set of 3451 genes with predicted translated extensions based on a relaxed threshold (rank=5000) which is now provided in Supplementary table 2. We also replicated the analysis of non-AUG initiation features for

RiboSET_ext (Supplementary Fig.10), it is very similar to what we observed in RiboSET, so it does not contradict with findings that we discovered using the strict threshold.

We claim that a negative PhyloCSF score based on 120-mammals alignment calculated for extensions in both RiboSET (Fig.3h) and RiboSET_ext (Supplementary Fig.10a) provides us with evidence of a lack of evolutionary selection among mammals. We added following sentences in 'Characterisation of genes with predicted non-AUG initiation' section:

More importantly, the PhyloCSF score of upstream regions of RiboSET_ext is also skewed towards large negative values providing us with evidence of no evolutionary selection among mammals for thousands of translated non-AUG extensions (Supplementary Fig.10a).

Perhaps, the reviewer misinterpreted our claim as "evidence of a lack of selection" in general. This is not the case. We cannot exclude a possibility of evolutionary selection acting on these genes within a narrow phylogenetic group such as primates, hominids or even humans only and we discuss such a possibility in the Discussion.

As the authors emphasized above their aim of this study is to develop methodology, but their ABSTRACT and title does not reflect the aim in both versions. It is hard to accept that the authors did not talk about the aim of this study at all in the ABSTRACT but only described the result in one or two panels in the manuscript.

If the authors want to draw a conclusion that "Thousands of human non-AUG extended proteoforms lack evidence of evolutionary selection among mammals", they need to provide more evidence to validate this point. Only one or two panels with evidence are not enough.

In addition, non-AUG extended proteoforms are evolutionarily un-conserved is already known, because most of them are translational molecular error that has been reported before (Mol Biol Evol. 2020;37:2015–28). Even the authors tried their best to validate this focus, it lacks novelty.

1. We disagree with placing an equal sign between the lack of selection within a specific phylogenetic group to "a molecular error". First the selection may occur in a narrower phylogenetic group as argued above. Second, most of the evolution occurring in species with a small effective population size such as mammals is neutral. This was established through seminal works by Motoo Kimura in the 70s and 80s. Just as an example, consider olfactory receptors in humans. While most are not selected as evident from the large number of corresponding pseudogenes in our genome, we wouldn't term expression of an active olfactory receptor as a "molecular error". If we accept this point of view we would need to refer to whole classes of non-conserved biological molecules such as lncRNAs as "molecular errors". On the other hand, all evolution may be described as a result of "molecular errors" as errors are a major driver of evolution.

I was arguing about the significance of this study not the molecular error itself, as that non-AUG extended proteoforms are evolutionarily un-conserved has been reflected in Mol Biol Evol. 2020;37:2015–28.

"Slightly deleterious or nearly neutral" is better than "neutral" when talking about molecular evolution.

Under the "molecular error hypothesis" in Mol Biol Evol. 2020;37:2015–28, the authors should not refer to all lncRNAs as molecular error, but could refer to most of them as molecular error.

The authors can analogically treat "molecular error" as a kind of mutation. Most mutations are deleterious and un-conserved, but they are one major driver of evolution.

2. Furthermore *Mol Biol Evol.* 2020;37:2015–28, focuses on non-AUG initiation irrespective of its functional consequences, e.g. whether they result in uORFs or N-extensions. Most importantly their study was limited to QTI-seq data. While QTI-seq data are useful and we used it in our own study when trying to predict locations of potential non-AUG codons, these data are very noisy and subject to technical errors.

First QTI-seq has been shown to enrich ribosomes not only at initiating codons, but also at two specific codons during elongation see <http://www.rnajournal.org/cgi/doi/10.1261/rna.049908.115> and Dmitriev SE, Akulich KA, Andreev DE, Terenin IM, Shastky IN. The peculiar mode of translation elongation inhibition by antitumor drug harringtonin. *FEBS J.* 2013;280:51.

QTI-seq paper was published in 2015, but the cited references here, the latter was published in 2013, although the former was published in 2015 but it does not cite QTI-seq. The authors did not show the analyses based QTI-seq and did not provide any evidence (not the WORDS in the response) about why did not use QTI-seq data in both versions.

Furthermore QTI-seq is very narrow, reads supporting initiation at a particular codon map to a corresponding single location. Assessing translation based on a single translation event is highly unreliable due to sequencing biases, mappability and other issues that we discussed in detail in the first section of our recent review on the topic (doi: 10.1002/wrna.1577).

I did not find the citation of QTI-seq in the first section of the review (doi: 10.1002/wrna.1577); the discussion about QIT-seq is only one small paragraph in the fifth section, but did not talk about the problem you mentioned here either. In addition, if the authors think it is not reliable to base on a single translation event, they can combine the Ribo-seq and QTI-seq, but they did nothing at genome scale in both versions.

The approach that we used here is by far more reliable and robust because it relies on reads aligned from a large region of mRNA and multiple studies, thus reducing the effect of artefacts and biases of individual locations and studies.

The authors did not provide results using other methods and datasets, how to say "more reliable"? For example, comparing with the combination of QTI-seq and Ribo-seq, which will be better?

Of note, the MBE study did not rely on phylogenetic approaches to infer non-adaptiveness of extensions encoded by the non-AUG starts; their claims are based mostly on comparisons between translational amount of genes and TIS diversity.

MBE study did, but the authors ignored at reading. Please see it's Fig. 4 about the phastCons scores.

There are some important datasets this study did not collect and use. For translation initiation, the high throughput sequencing method, GTI/QTI-seq (Nature methods, 12(2), 147-153), can directly catch the initiation codon, level and position. Thus, it is more direct and useful than Ribo-seq. I did not see how this study include or analyzes this kind of data.

As we discussed above, while useful, GTI/QTI-seq is very noisy and predictions made based on it are unlikely to be robust since they would rely on very narrow mapping. We did use it however in manual assessments of exact potential locations of non-AUG starts during manual evaluation of the selected genes.

See the above about the QTI-seq. The authors did not show the results based QTI-seq.

The authors did comparisons, including between the two methods and between this study and previous study, but the overlap part is always small, which makes people suspect the reliability and rationality of the data or methods used. As the main aim in this study is not on the comparison, the authors should not take much effort on the comparison.

We extensively discussed why overlap between phylogenetic and ribo-seq approaches is smaller. It is one of the most important outcomes of our study.

The small overlap with the previous study is due to various reasons that we thoroughly explained in Results. E.g. in comparison with Ivanov et.al there are multiple reasons including differences in RefSeq and GENCODE; a considerable fraction of originally discovered non-AUG genes have been already annotated in the recent GENCODE version. Also, we tried to rediscover genes that have not yet been annotated.

Back then, in 2011, massive multiple sequence codon alignments such as 100 vertebrates or 120 mammals were not available and the analysis was based solely on human-mouse paired alignment which is likely to have less information in comparison to 120-mammals. Also, there was only a single Ribo-seq dataset available for humans and we used it purely for illustrative purposes. It was not used for detection of N-extensions.

I was arguing about the reliability and rationality of the data or methods they used. Although they explain the overlap part is small, but lack the discussion about the reliability of their method.

Reviewer #3 (Remarks to the Author):

The revisions have addressed my concerns with the previous version.

I did notice a few minor points in the current manuscript:

In Fig. 3h, the x- and y-axis labels appear to be switched

On p. 7, it is written, "To show whether a threshold of 500 top ranking candidates is too conserved..." and I think this should be "conservATIVE"

On p. 8, there is a citation to "(Ivanov et al 2011, ref 21)" but this is ref 22 in the current version.

REVIEWER COMMENTS

In the following response to Reviewer #1, **the original reviewer comments are shown in bold**, our previous response to these comments is in roman, **the new Reviewer comments are highlighted in blue**, our new response to them is highlighted in green and orange indicates the related changes in the manuscript.

Reviewer #1 (Remarks to the Author):

In this manuscript, Fedorova and the co-authors used two approaches to identify non-AUG initiated proteoforms. The first is phyloCSF, in which 60 candidate genes were identified based on genomic data; while the second, Trips-Viz based on Ribo-seq and proteomics data, found 392 candidate genes. However, only 8 genes are common between the two methods. By comparing their dataset with GENCODE v35, less than half annotated non-AUG proteoforms were validated. They further studied the properties of predicted non-AUG initiation codons, including the distribution of start codon types, TIS sequence logo and frequency, mRNA secondary structure, and domains. Finally, they updated 11 genes about the non-AUG initiated proteoforms.

We kindly thank the reviewer for their feedback.

We would like to note that we did not compare our datasets with GENCODE v35. We excluded genes that have already been annotated in GENCODE v35 from RiboSET and PhyloSET. Instead, we tested whether we were able to re-identify genes with already annotated non-AUG extensions in GENCODE v35 from the Ivanov et al 2011 study using the PhyloCSF score if we moved the start to the originally annotated AUG. Indeed, only 11 genes in the original study have a positive PhyloCSF score and we discuss the possible explanations of such an overlap in the Discussion. Briefly, in Ivanov et al only human-mouse paired alignments were used unlike the 120-mammals alignment that we used for PhyloCSF score calculation. Also we calculated the score for 50 upstream codons while conserved extension could be way shorter thus extra upstream codons could add a large negative value to the final score.

We updated annotations for 11 genes, however we expect many more to be annotated in the future, for example SLC25A32 was annotated as such during the revision of this paper.

I was not arguing about their comparison, instead I was talking about the work's significance to the field as only 11 genes were updated.

Current gene annotation guidelines rely heavily on phylogenetic evidence for annotation of 'missing' protein coding sequences. In order to be annotated, proteins are required to be evolutionary 'old', essentially to present evidence as evolving as coding sequence across the mammalian order as a minimum. Ape- or human- specific proteins are not typically annotated without experimental support for protein existence, which in practice generally means proteomics data.

If ORF protein-coding ability is endorsed only by Ribo-seq, it is not subjected to the immediate annotation since no suitable guidelines have been introduced yet.

Also, each individual gene is being studied manually by the GENCODE curators. At the moment of first submission, we had 11 genes annotated, before the first revision another gene has been introduced (*SLC25A32*) and now another 18 (*ADO, JUN, HDGF, POGZ, BRD7, PIM2, PTMS, CARM1, TARBP2, CHTOP, TNKS2, KAT7, FHIP2A, SYNCRIP, KCTD9, PPP4R2, ZNF384, SNRNP25*) have gained non-AUG models in both human and mouse. It is an ongoing effort and only genes with both sufficient phylogenetic evidence and clear Ribo-seq signal are considered. Such a massive translation at non-AUG starts raised an important issue regarding our understanding of protein-coding evolution. Currently we are working on novel annotation policies which will enable using Ribo-seq data as an independent evidence of protein-coding ability. It is a community effort therefore it will take a while before more proteoforms can be annotated.

Although the authors claim they “develop” the approaches for identifying of non-AUG translation initiation, the methods/tools used are not developed in the study. The authors did not generate any high-throughput data, they thus did not produce useful resources yet.

We thank the reviewer for bringing our attention to the fact that we did not sufficiently highlight the new methods developed in our work.

First of all, we described the algorithm which allows for prediction of translated ORFs using aggregated Ribo-seq data. While we used it in this manuscript only for identification of non-AUG N-terminal extensions, its implementation in the Trips-Viz browser can be used for detecting any kind of unannotated translation including upstream, downstream, nested and overlapping ORFs. The detailed description of the algorithm is in Supplementary Methods. We also provided a link to the code at github - https://github.com/skiniry/Trips-Viz/blob/master/orfquery_routes.py.

We apologise for not explicitly stating this in the original version of manuscript and thank the Reviewer for pointing this out. We have now added the following sentence to the corresponding Results:

We have developed a translating ORF predictor, an algorithm for detecting translated ORFs using aggregated Ribo-seq data (Supplementary Methods). The implementation of this algorithm is now available in the Trips-Viz browser.

And also modified the abstract:

Here we analysed a large number of publicly available ribo-seq datasets to identify novel, previously uncharacterised non-AUG proteoforms using Trips-Viz implementation of a novel algorithm for detecting translated ORFs.

In addition to the general purpose of translated ORF detection, specifically for the purpose of this work, we designed and described a pipeline that can detect non-AUG N-terminal extensions using 2 orthogonal approaches. This pipeline includes extracting upstream

regions of canonically annotated transcript models, deriving genomic coordinates of theoretical N-terminal extensions (NTEs), filtering overlaps with known coding regions, stitching together exons of NTEs and calculating PhyloCSF scores. Another branch includes applying the aforementioned ORF-predictor on transcript models. To our knowledge, this pipeline has not been used for detection of non-AUG initiated proteoforms anywhere else.

They did not describe their new approach in the original manuscript, so that I had to point out the problem. In the revised version, they provided some details, but did not show its novelty relative to others. In addition, the reliability and advancement of their method is unknown.

We described the algorithm in Supplementary Methods in the previous revised version. For the clarity now we have added following description of Trips-viz ORF predictor algorithm and its advancements:

Trips-viz is a computational data environment for analysing Ribo-seq data on a transcriptome level^{37,38}. It contains thousands of uniformly processed public Ribo-seq data and provides tools for analysis and visualisation of translation. Our recent addition to the Trips-viz platform is Ribo-seq ORF predictor which outputs a list of ranked translated ORFs. The biggest advantage of the tool is that a large number of processed public Ribo-seq data is already available to users and they can apply ORF predictor and visualise results immediately. The tool is tailored to detect different types of ORFs including uORFs, N-terminal extensions (NTEs), nORFs, CDSs and dORFs. The algorithm for NTEs is based on triplet periodicity present across the entire length of extension and utilises patterns of ORF translation such as consistency of ribosome footprint triplet periodicity within the reading frame, the increase of footprint density at the potential start, non zero coverage and average read density (see **Supplementary Methods**). Of note, it automatically filters out regions that overlap with coding exons. We employed this algorithm to detect non-AUG N-terminally extended proteoforms using aggregated elongating and initiating ribosome profiling data with high triplet periodicity score (**Supplementary Table 8**).

During the round of review, we also noticed a study [PMID: 35841888], published in August 2022 (our BiorXiv paper was available since May) which has a Ribo-seq ORF predictor very much resembling our algorithm. They did not perform any benchmarking against existing tools. This is because the objective benchmarking of the Ribo-seq ORF predictors is impossible due to the lack of a proper gold standard.

In the absence of a gold standard, we could rely only on manual evaluation of the profiles (which we do and we also provide plots of footprint densities in the manuscript as well as in Trips-Viz, so that the readers can visually evaluate the evidence).

Again, I was arguing about the work's significance to the field. They did not provide us method with novelty and did not generate original high-throughput useful data yet even in current revised version.

There are three aspects of the novelty in our approach.

1. New algorithm for the detection of translation.
2. The use of aggregated data.
3. The combination of Ribo-seq evidence with phylogenetic evidence of protein coding evolution (PhyloCSF).

These aspects have led to the annotation of novel coding sequences in numerous protein-coding genes, and - as far as we know - each of these cases represents a novel discovery.

We added the following text:

PhyloCSF is a state-of-the-art method for assessing the evolutionary protein-coding potential of a genomic region based on multiple sequence alignment.

Furthermore we introduced the Trips-viz ORF predictor as a part of the Trips-viz environment. We mentioned its key features and advantages in the comment above and in the text and we also pointed out that a very similar algorithm has been very recently published.

Having a comprehensive and accurate gene annotation is crucially important for a wide research community especially for clinicians who heavily rely on gene annotation with regards to the variant interpretation. Typically variant calling is based on annotated coding sequences thus discovery of novel protein-coding regions is necessary for ensuring that pathogenic variants are not overlooked.

We generated 3 sets of genes with non-AUG N-terminal extensions: PhyloSET, RiboSET and RiboSET_ext and we've been gradually annotating them. These sets provide information about additional protein coding regions located in previously thought to be untranslated regions (5'UTRs) and can be used for inferring variation impact and for experimental validation.

We overlapped ClinVar variants with predicted non-AUG extensions from RiboSET_ext and found 124 genes (201 variants) with either pathogenic, likely pathogenic or conflicting interpretations of pathogenicity. This is likely to be an underestimation of variation simply because only annotated coding regions are generally used for variant calling. Similarly, for primary extensions in PhyloSET we found gene *GDF5OS* carrying 4 variants (pathogenic and likely pathogenic).

We also added these lines to the main text, into Reannotation of non_AUG proteoforms:

Having a comprehensive and accurate gene annotation is crucially important for a wide research community especially for clinicians who heavily rely on gene annotation with regards to the variant interpretation. Since we discovered novel protein-coding regions, we overlapped ClinVar variants with predicted non-AUG extensions from RiboSET_ext and found 124 genes (201 variants) with either pathogenic, likely pathogenic or conflicting interpretations of pathogenicity (**Supplementary Table 9**). This is likely to be an underestimation of variation simply because only annotated coding regions are generally used for variant calling. Similarly, for primary extensions in PhyloSET we found gene *GDF5OS* carrying 4 variants (pathogenic and likely pathogenic, **Supplementary Table 9**). Therefore this set can be used for assessing variation occurring in 5'UTRs.

We also added the following lines to the Methods:

We utilised variants from ClinVar (clinvar_20221015.vcf.gz) and overlapped them with predicted extensions (RiboSET_ext, PhyloSET) using '*bedtools intersect*'.

Although they analyzed 11 datasets, the data is still small relative to tons of existed Ribo-seq data and proteomics data, which cannot provide an impressive landscape of non-AUG initiated proteoforms.

Again, we thank the reviewer for drawing our attention to this matter as the text in the previous version somewhat undermined the scale of our effort.

It is important to point out that for this analysis we started with over 1000 ribosome profiling datasets from 48 published papers which have been processed and are currently available in Trips-Viz. However, for accurate detection of N-extensions it is crucial to detect the correct translated frame. Thus, we analysed all these datasets to determine the strength of triplet periodicity. We chose only 152 datasets from 13 research publications that produced ribo-seq data with high periodicity score (more than 0.5). Perhaps misleadingly we also used the number of research papers used as the number of datasets. This is now corrected, we also added an explanatory statement in the “Detection of translated regions in Trips-Viz” section of Methods explaining this.

We provided periodicity scores for each study in Supplementary table 8.

In relation to the entire landscape of N-terminal extensions in human genes, this would require a generation of ribosome profiling data from virtually all human cells, which is currently not feasible for technical and ethical reasons.

We emphasised that the aim of our study was not to exhaustively detect all existing non-AUG initiated proteoforms, but to develop methodology for this and explore the relationship between the two approaches. Unexpectedly we found that the number of N-extensions detectable with ribosome profiling obtained in a limited number of human cells by far exceeds that detectable with current phylogenetic approaches. This points to the existence of thousands of translated non-AUG initiated proteoforms with no evidence of purifying selection across mammals. We made the following change in the discussion to clarify this:

We did not aim to exhaustively detect all existing non-AUG initiated proteoforms but rather compare two approaches and the relationship between translated N-extensions detected with ribosome profiling and evidence of protein coding evolution in the corresponding genes.

As the authors emphasized their aim of this study is to develop methodology, they need to compare with other methodologies to show the novelty and reliability, but they did not in both versions.

Indeed we have the following aims: (1) develop methodology for detection of non-AUG proteoforms based on translation and evolutionary approaches; (2) detect novel non-AUG proteoforms; (3) compare gene sets resulted from these methods.

The main outcome of the study is stated in the title - this way we wanted to emphasise that we did not intend the reader to interpret the main goal of the paper as only development of the new methodology for detection of the non-AUG proteoforms.

Novelty and reliability of our methodology is provided in the above comments (the

description of Trips-viz ORF caller and why we chose PhyloCSF).

More importantly, an interesting story, such as a genomic/genetical pattern or mechanism, should be proposed, but it seems to be lack. Taking together, this research needs to be improved on the novelty and workload, and does not fulfil the publication standard of Nature Communications at its current stage.

The mechanisms involved in non-AUG initiation in general are reviewed e.g. in <http://www.genesdev.org/cgi/doi/10.1101/gad.305250.117>, <https://doi.org/10.1186/s13059-022-02674-2> and briefly described in our introduction. It is not clear to us what the reviewer meant by genomic/genetical pattern.

Again, I was arguing about their work's significance. One of the reasons is that their work failed to provide any new stories about the genomic or genetical pattern/principles in both previous and current versions.

As we mentioned above, we showed the occurrence of pathogenic variants from ClinVar within predicted non-AUG extensions and it's likely to be an underestimation due to the protein-coding region's bias in variant calling. We also tested hypothesis that alternative N-termini encoded with non-AUG carry alternative compartmentalisation signals, although there was no enrichment found in comparison to control genes with no translation upstream, we still cannot exclude the possibility that at least some of them genuinely contain alternative localisation signals. Such an abundance of translated unconserved in mammals non-AUG proteoforms also raised a need for considering Ribo-seq data as primary evidence of protein-coding ability and we are currently working on that. Novel proteoforms are gradually annotated.

We also emphasise that our novel annotations are found within genes of clear medical relevance, such as *JUN*. It is reasonable to assume that such N-t extensions will hold biological information that will prove to be of profound importance in genomic science and medicine. We accept that we are not in a position to describe the precise functionality of such N-terminal extensions in physiological terms. These aspects will need to be studied with single locus high granularity experiments, as part of dedicated research projects. However, others will now be able to explore these possibilities based on our work, as we are producing reference GENCODE annotation. Without our work here, the type of 'stories' that the reviewer has in mind would not be able to be told by anyone.

We also added following lines into the discussion:

The substantial number of pathogenic and likely pathogenic variants from ClinVar overlapping with predicted nonAUG extensions and the fact that novel annotations are found within genes of clear medical relevance, such as *JUN*, it is reasonable to assume that such extensions will hold biological information that will prove to be of profound importance in genomic science and medicine.

Some concerns:

The first problem is the title "Thousands of human non-AUG extended proteoforms lack evidence of evolutionary selection among mammals". In fact, the main work or goal of this study is the identification of some non-AUG initiated proteoforms, which is far from the content in the title. The authors need to apply as many ways as possible to validate the title's content (especially for the "lack evidence"), but they did

not. The authors seem to be deliberately overstating their research to draw attentions to readers.

The title indeed focuses on an important outcome of our study that has not been emphasised enough in the manuscript.

Describing features of extensions, we focused on only hundreds of detected genes taking a very conservative threshold (Fig.3d) initially and we argue that it is better to use a high-confident set which is more likely devoid of false positives.

The relaxed thresholds lead to detection of thousands of translated non-AUG extensions enriched with those from PhyloSET up to thresholds (6000-7000 rank) see Fig. 3d. Most lack phylogenetic support (considering that PhyloSET has only 60 genes). We have now generated RiboSET_ext - a set of 3451 genes with predicted translated extensions based on a relaxed threshold (rank=5000) which is now provided in Supplementary table 2. We also replicated the analysis of non-AUG initiation features for RiboSET_ext (Supplementary Fig.10), it is very similar to what we observed in RiboSET, so it does not contradict with findings that we discovered using the strict threshold.

We claim that a negative PhyloCSF score based on 120-mammals alignment calculated for extensions in both RiboSET (Fig.3h) and RiboSET_ext (Supplementary Fig.10a) provides us with evidence of a lack of evolutionary selection among mammals. We added following sentences in 'Characterisation of genes with predicted non-AUG initiation' section:

More importantly, the PhyloCSF score of upstream regions of RiboSET_ext is also skewed towards large negative values providing us with evidence of no evolutionary selection among mammals for thousands of translated non-AUG extensions (Supplementary Fig.10a).

Perhaps, the reviewer misinterpreted our claim as “evidence of a lack of selection” in general. This is not the case. We cannot exclude a possibility of evolutionary selection acting on these genes within a narrow phylogenetic group such as primates, hominids or even humans only and we discuss such a possibility in the Discussion.

As the authors emphasized above their aim of this study is to develop methodology, but their ABSTRACT and title does not reflect the aim in both versions. It is hard to accept that the authors did not talk about the aim of this study at all in the ABSTRACT but only described the result in one or two panels in the manuscript.

As mentioned in the above comments, developing the methodology was one of the necessary steps and it should not be read as the only thing that was done in the study.

Abstract outlines the Trips-viz Ribo-seq ORF predictor and analysis of 120 mammal genomes alignment as developed methodology and the comparison between 2 approaches.

If the authors want to draw a conclusion that “Thousands of human non-AUG extended proteoforms lack evidence of evolutionary selection among mammals”, they need to provide more evidence to validate this point. Only one or two panels with evidence are not enough.

The title signifies main parts/conclusions.

1. Translation of novel non-AUG proteoforms is supported with analysing a large number of publicly available Ribo-seq data aggregated in a uniform manner.
2. Lack of protein-coding evolution upstream of annotated AUG codons in mammals - we proved that utilising state-of-the-art PhyloCSF score and 120 mammals alignment.
3. The existence of a small overlap which is stated in the main text and indeed shown only in one panel, **Fig.3f**. However, we do not agree with the reviewer that the strength of evidence should be based on the number of figures presented in a manuscript as it may lead to artificial and unnecessary inflation of manuscripts' sizes rather than strengthening the evidence. One clear analysis is sufficient.

In addition, non-AUG extended proteoforms are evolutionarily un-conserved is already known, because most of them are translational molecular error that has been reported before (Mol Biol Evol. 2020;37:2015–28). Even the authors tried their best to validate this focus, it lacks novelty.

1. We disagree with placing an equal sign between the lack of selection within a specific phylogenetic group to “a molecular error”. First the selection may occur in a narrower phylogenetic group as argued above. Second, most of the evolution occurring in species with a small effective population size such as mammals is neutral. This was established through seminal works by Motoo Kimura in the 70s and 80s. Just as an example, consider olfactory receptors in humans. While most are not selected as evident from the large number of corresponding pseudogenes in our genome, we wouldn't term expression of an active olfactory receptor as a “molecular error”. If we accept this point of view we would need to refer to whole classes of non-conserved biological molecules such as lncRNAs as “molecular errors”. On the other hand, all evolution may be described as a result of “molecular errors” as errors are a major driver of evolution.

I was arguing about the significance of this study not the molecular error itself, as that non-AUG extended proteoforms are evolutionarily un-conserved has been reflected in Mol Biol Evol. 2020;37:2015–28.

“Slightly deleterious or nearly neutral” is better than “neutral” when talking about molecular evolution.

Under the “molecular error hypothesis” in Mol Biol Evol. 2020;37:2015–28, the authors should not refer to all lncRNAs as molecular error, but could refer to most of them as molecular error.

The authors can analogically treat “molecular error” as a kind of mutation. Most mutations are deleterious and un-conserved, but they are one major driver of evolution.

Regretfully we failed to interpret these comments of the Reviewer and to understand their relationship to our manuscript.

2. Furthermore Mol Biol Evol. 2020;37:2015–28, focuses on non-AUG initiation irrespective of its functional consequences, e.g. whether they result in uORFs or N-extensions. Most importantly their study was limited to QTI-seq data. While QTI-seq data are useful and we used it in our own study when trying to predict locations of potential non-AUG codons, these data are very noisy and subject to technical errors.

First QTI-seq has been shown to enrich ribosomes not only at initiating codons, but also at two specific codons during elongation see <http://www.rnajournal.org/cgi/doi/10.1261/rna.049908.115> and Dmitriev SE, Akulich KA, Andreev DE, Terenin IM, Shastky IN. The peculiar mode of translation elongation inhibition by antitumor drug harringtonin. FEBS J. 2013;280:51.

QTI-seq paper was published in 2015, but the cited references here, the latter was published in 2013, although the former was published in 2015 but it does not cite QTI-seq. The authors did not show the analyses based QTI-seq and did not provide any evidence (not the WORDS in the response) about why did not use QTI-seq data in both versions.

The reviewer is correct that Dmitriev et al (2013) preceded the QTI-seq paper (2015). Dmitriev et al deals with an artefact generated by a drug Harringtonine (arrest of ribosomes at AAG/AGG codons during elongation). QTI-seq is based on the drug lactimidomycin that was first introduced in 2012:

Lee S, Liu B, Lee S, Huang SX, Shen B, Qian SB. Global mapping of translation initiation sites in mammalian cells at single-nucleotide resolution. Proc Natl Acad Sci U S A . 2012 Sep 11;109(37):E2424-32.

We have shown that the artefact reported by Dmitriev et al for Harringtonin also occurs in the case of lactimidomycin:

Michel AM, Andreev DE, Baranov PV. Computational approach for calculating the probability of eukaryotic translation initiation from ribo-seq data that takes into account leaky scanning. BMC Bioinformatics. 2014 Nov 21;15(1):380. doi: 10.1186/s12859-014-0380-4. PMID: 25413677; PMCID: PMC4245810.

This publication also precedes the QTI-seq publication. However, QTI-seq is based on the use of lactimidomycin in combination with puromycin. The authors of QTI-seq did not provide any evidence that puromycin treatment removes this artefact in their publication:

Gao X, Wan J, Liu B, Ma M, Shen B, Qian SB. Quantitative profiling of initiating ribosomes in vivo. Nat Methods. 2015 Feb;12(2):147-53. doi: 10.1038/nmeth.3208. Epub 2014 Dec 8. PMID: 25486063; PMCID: PMC4344187.

Subsequently, we see no reason to believe that QTI-seq should be free from this artefact and therefore refer to a prior study in relation to this artefact.

Nonetheless, despite these artefacts we do use the data obtained with inhibitors of translation initiation. We selected the data for the analysis based on the quality of the

datasets (such as the strength of triplet periodicity) irrespective of the specific drugs that were used. The description of studies that we used was available in Supplementary Table 8 for the previous round of review process. We used aggregation of both types of Ribo-seq data, those generated with inhibitors of elongation as well as initiation. Asterisks show initiating ribosome profiling studies. The advantage of aggregating datasets obtained with different ribosome profiling protocols is that the effects of artefacts pertinent to specific protocols is reduced.

Furthermore QTI-seq is very narrow, reads supporting initiation at a particular codon map to a corresponding single location. Assessing translation based on a single translation event is highly unreliable due to sequencing biases, mappability and other issues that we discussed in detail in the first section of our recent review on the topic (doi: 10.1002/wrna.1577).

I did not find the citation of QTI-seq in the first section of the review (doi: 10.1002/wrna.1577); the discussion about QIT-seq is only one small paragraph in the fifth section, but did not talk about the problem you mentioned here either. In addition, if the authors think it is not reliable to base on a single translation event, they can combine the Ribo-seq and QTI-seq, but they did nothing at genome scale in both versions.

See the above comments - we did it originally.

The approach that we used here is by far more reliable and robust because it relies on reads aligned from a large region of mRNA and multiple studies, thus reducing the effect of artefacts and biases of individual locations and studies.

The authors did not provide results using other methods and datasets, how to say "more reliable"? For example, comparing with the combination of QTI-seq and Ribo-seq, which will be better?

The higher reliability of using several data points per transcript than a single one is expected from the central limit theorem. The larger the sample size the better it represents the real distribution.

This is particularly pertinent to ribosome profiling which is significantly affected by sequencing biases, e.g. due to substrate specificity during adaptor ligation. The interpretation of ribosome profiling data based on a single location is therefore dangerous. For example, in this study:

Han Y, Gao X, Liu B, Wan J, Zhang X, Qian SB. Ribosome profiling reveals sequence-independent post-initiation pausing as a signature of translation. *Cell Res.* 2014 Jul;24(7):842-51. doi: 10.1038/cr.2014.74. Epub 2014 Jun 6. PMID: 24903108; PMCID: PMC4085768.

The authors interpreted the peak at the 5th codon in metagene profile (Fig R1) as a genuine reflection of ribosome progression over mRNA and published the above paper suggesting that this is a real ribosomal pause. However, the peak occurs because the 5' end of ribosomal footprints coincides with the location of AUG codon since the ribosome protects

approximately 4 codons 5' of its A-site. Sequences with AUG at the 5' end of footprints were preferentially recognised by ligase enzymes thus leading to preferential sequencing at this location.

[redacted]

Figure R1. A high peak of ribosome footprint density is often observed 5 codons downstream of the initiation site. The panels are taken from Figure 1 of Han et al (2014)

The improvements in the ribosome profiling library generation led to the reduction in sequencing biases. Therefore this peak is not observed in most recently generated data (Fig R2).

[redacted]

Figure R2. The metagene profile for a recently generated data does not contain a peak at the 5th codon. The data are from Gameiro PA, Struhl K. Nutrient Deprivation Elicits a Transcriptional and Translational Inflammatory Response Coupled to Decreased Protein Synthesis. Cell Rep. 2018 Aug 7;24(6):1415-1424. doi: 10.1016/j.celrep.2018.07.021. PMID: 30089253; PMCID: PMC6419098.

While the technology improved substantially, relying on a large sample of data (in our case the number of coordinates per translated region) is a more cautious approach which we chose in our study.

Of note, the MBE study did not rely on phylogenetic approaches to infer non-adaptiveness of extensions encoded by the non-AUG starts; their claims are based mostly on comparisons between translational amount of genes and TIS diversity.

MBE study did, but the authors ignored at reading. Please see it's Fig. 4 about the phastCons scores.

On the Fig.4, PhastCons was used only for assessing Kozak sequence contexts unlike our study where protein-coding ability of the entire extension sequence was assessed with a PhyloCSF score.

They key difference between PhyloCSF and PhastCons is that PhyloCSF tracks represent a signal of constraint specifically for protein-coding function, whereas the signal represented by PhastCons is independent of the cellular function imposing the constraint and demonstrates rather nucleotide-level conservation.

There are some important datasets this study did not collect and use. For translation initiation, the high throughput sequencing method, GTI/QTI-seq (Nature methods, 12(2), 147-153), can directly catch the initiation codon, level and position. Thus, it is more direct and useful than Ribo-seq. I did not see how this study include or analyzes this kind of data.

As we discussed above, while useful, GTI/QTI-seq is very noisy and predictions made based on it are unlikely to be robust since they would rely on very narrow mapping. We did use it however in manual assessments of exact potential locations of non-AUG starts during manual evaluation of the selected genes.

See the above about the QTI-seq. The authors did not show the results based QTI-seq.

We replied to this comment in the comments above.

The authors did comparisons, including between the two methods and between this study and previous study, but the overlap part is always small, which makes people suspect the reliability and rationality of the data or methods used. As the main aim in this study is not on the comparison, the authors should not take much effort on the comparison.

We extensively discussed why overlap between phylogenetic and ribo-seq approaches is smaller. It is one of the most important outcomes of our study.

The small overlap with the previous study is due to various reasons that we thoroughly explained in Results. E.g. in comparison with Ivanov et.al there are multiple reasons including differences in RefSeq and GENCODE; a considerable fraction of originally discovered non-AUG genes have been already annotated in the recent GENCODE version. Also, we tried to rediscover genes that have not yet been annotated.

Back then, in 2011, massive multiple sequence codon alignments such as 100 vertebrates or 120 mammals were not available and the analysis was based solely on human-mouse

paired alignment which is likely to have less information in comparison to 120-mammals. Also, there was only a single Ribo-seq dataset available for humans and we used it purely for illustrative purposes. It was not used for detection of N-extensions.

I was arguing about the reliability and rationality of the data or methods they used. Although they explain the overlap part is small, but lack the discussion about the reliability of their method.

We discussed the issues that may arise while using PhyloCSF score in the Discussion.

We added the following text into the discussion regarding Trips-viz algorithm:

For prediction of translated non-AUG NTEs we utilised an Ribo-seq ORF predictor - a new addition to the Trips-viz environment - platform for analysing and visualising ribosome profiling data. The key advantage of the ORF predictor is its convenience: users have a large collection of Ribo-seq samples already embedded into the platform and can start using the predictor and visualise results straight away. Its output is in the form of a ranked list of translated ORFs that can be treated as both advantage and disadvantage since it provides a flexibility for setting out your own threshold although it might not be always straightforward to select one. In general, the Trips-viz ORF predictor algorithm is adapted individually for all classes of ORFs including uORFs, CDSs, NTEs, nORFs and dORFs. It is based on the triplet periodicity signal in the entire ORF and therefore it performs its best when applied to the aggregation of elongating ribosome data either with or without initiating ribosome profiling.

Reviewer #3 (Remarks to the Author):

The revisions have addressed my concerns with the previous version.

I did notice a few minor points in the current manuscript:

In Fig. 3h, the x- and y-axis labels appear to be switched

We thank the reviewer for pointing that out and we switched labels.

On p. 7, it is written, "To show whether a threshold of 500 top ranking candidates is too conserved..." and I think this should be "conservATIVE"

We corrected that in a manuscript.

On p. 8, there is a citation to "(Ivanov et al 2011, ref 21)" but this is ref 22 in the current version.

We corrected that in a manuscript.

REVIEWERS' COMMENTS

Reviewer #1 (Remarks to the Author):

See the attached file. Basically, this study lacks the novelty when the authors reject to provide more evidence for the title that they claimed.

In the rounds of reviews for this manuscript, I always have two major comments in fact. One is about the QTI-seq data, which has been addressed in the second revision. Another one is the novelty of this study, because its focus/title has been already known from the MBE study. I have suggested the authors twice to provide more new evidence, not only based one or two panels, to improve the novelty, but the authors rejected in the rounds of review.

REVIEWER COMMENTS

In the following response to Reviewer #1, **the original reviewer comments are shown in bold**, our previous response to these comments is in roman, **the new Reviewer comments are highlighted in blue**, our new response to them is highlighted in green and orange indicates the related changes in the manuscript.

Reviewer #1 (Remarks to the Author):

In this manuscript, Fedorova and the co-authors used two approaches to identify non-AUG initiated proteoforms. The first is phyloCSF, in which 60 candidate genes were identified based on genomic data; while the second, Trips-Viz based on Ribo-seq and proteomics data, found 392 candidate genes. However, only 8 genes are common between the two methods. By comparing their dataset with GENCODE v35, less than half annotated non-AUG proteoforms were validated. They further studied the properties of predicted non-AUG initiation codons, including the distribution of start codon types, TIS sequence logo and frequency, mRNA secondary structure, and domains. Finally, they updated 11 genes about the non-AUG initiated proteoforms.

We kindly thank the reviewer for their feedback.

We would like to note that we did not compare our datasets with GENCODE v35. We excluded genes that have already been annotated in GENCODE v35 from RiboSET and PhyloSET. Instead, we tested whether we were able to re-identify genes with already annotated non-AUG extensions in GENCODE v35 from the Ivanov et al 2011 study using the PhyloCSF score if we moved the start to the originally annotated AUG. Indeed, only 11 genes in the original study have a positive PhyloCSF score and we discuss the possible explanations of such an overlap in the Discussion. Briefly, in Ivanov et al only human-mouse paired alignments were used unlike the 120-mammals alignment that we used for PhyloCSF score calculation. Also we calculated the score for 50 upstream codons while conserved extension could be way shorter thus extra upstream codons could add a large negative value to the final score.

We updated annotations for 11 genes, however we expect many more to be annotated in the future, for example SLC25A32 was annotated as such during the revision of this paper.

I was not arguing about their comparison, instead I was talking about the work's significance to the field as only 11 genes were updated.

Current gene annotation guidelines rely heavily on phylogenetic evidence for annotation of 'missing' protein coding sequences. In order to be annotated, proteins are required to be evolutionary 'old', essentially to present evidence as evolving as coding sequence across the mammalian order as a minimum. Ape- or human- specific proteins are not typically annotated without experimental support for protein existence, which in practice generally means proteomics data.

If ORF protein-coding ability is endorsed only by Ribo-seq, it is not subjected to the immediate annotation since no suitable guidelines have been introduced yet.

Also, each individual gene is being studied manually by the GENCODE curators. At the moment of first submission, we had 11 genes annotated, before the first revision another gene has been introduced (*SLC25A32*) and now another 18 (*ADO, JUN, HDGF, POGZ, BRD7, PIM2, PTMS, CARM1, TARBP2, CHTOP, TNKS2, KAT7, FHIP2A, SYNCRIP, KCTD9, PPP4R2, ZNF384, SNRNP25*) have gained non-AUG models in both human and mouse. It is an ongoing effort and only genes with both sufficient phylogenetic evidence and clear Ribo-seq signal are considered. Such a massive translation at non-AUG starts raised an important issue regarding our understanding of protein-coding evolution. Currently we are working on novel annotation policies which will enable using Ribo-seq data as an independent evidence of protein-coding ability. It is a community effort therefore it will take a while before more proteoforms can be annotated.

As most of the non-AUG translation belongs to translation error, I did not see the significance even the authors added another 18 genes with non-AUG initiation.

Although the authors claim they “develop” the approaches for identifying of non-AUG translation initiation, the methods/tools used are not developed in the study. The authors did not generate any high-throughput data, they thus did not produce useful resources yet.

We thank the reviewer for bringing our attention to the fact that we did not sufficiently highlight the new methods developed in our work.

First of all, we described the algorithm which allows for prediction of translated ORFs using aggregated Ribo-seq data. While we used it in this manuscript only for identification of non-AUG N-terminal extensions, its implementation in the Trips-Viz browser can be used for detecting any kind of unannotated translation including upstream, downstream, nested and overlapping ORFs. The detailed description of the algorithm is in Supplementary Methods. We also provided a link to the code at github - https://github.com/skiniry/Trips-Viz/blob/master/orfquery_routes.py.

We apologise for not explicitly stating this in the original version of manuscript and thank the Reviewer for pointing this out. We have now added the following sentence to the corresponding Results:

We have developed a translating ORF predictor, an algorithm for detecting translated ORFs using aggregated Ribo-seq data (Supplementary Methods). The implementation of this algorithm is now available in the Trips-Viz browser.

And also modified the abstract:

Here we analysed a large number of publicly available ribo-seq datasets to identify novel, previously uncharacterised non-AUG proteoforms using Trips-Viz implementation of a novel algorithm for detecting translated ORFs.

In addition to the general purpose of translated ORF detection, specifically for the purpose of this work, we designed and described a pipeline that can detect non-AUG N-terminal extensions using 2 orthogonal approaches. This pipeline includes extracting upstream regions of canonically annotated transcript models, deriving genomic coordinates of theoretical N-terminal extensions (NTEs), filtering overlaps with known coding regions, stitching together exons of NTEs and calculating PhyloCSF scores. Another branch includes applying the aforementioned ORF-predictor on transcript models. To our knowledge, this pipeline has not been used for detection of non-AUG initiated proteoforms anywhere else.

They did not describe their new approach in the original manuscript, so that I had to point out the problem. In the revised version, they provided some details, but did not show its novelty relative to others. In addition, the reliability and advancement of their method is unknown.

We described the algorithm in Supplementary Methods in the previous revised version. For the clarity now we have added following description of Trips-viz ORF predictor algorithm and its advancements:

Trips-viz is a computational data environment for analysing Ribo-seq data on a transcriptome level^{37,38}. It contains thousands of uniformly processed public Ribo-seq data and provides tools for analysis and visualisation of translation. Our recent addition to the Trips-viz platform is Ribo-seq ORF predictor which outputs a list of ranked translated ORFs. The biggest advantage of the tool is that a large number of processed public Ribo-seq data is already available to users and they can apply ORF predictor and visualise results immediately. The tool is tailored to detect different types of ORFs including uORFs, N-terminal extensions (NTEs), nORFs, CDSs and dORFs. The algorithm for NTEs is based on triplet periodicity present across the entire length of extension and utilises patterns of ORF translation such as consistency of ribosome footprint triplet periodicity within the reading frame, the increase of footprint density at the potential start, non zero coverage and average read density (see **Supplementary Methods**). Of note, it automatically filters out regions that overlap with coding exons. We employed this algorithm to detect non-AUG N-terminally extended proteoforms using aggregated elongating and initiating ribosome profiling data with high triplet periodicity score (**Supplementary Table 8**).

During the round of review, we also noticed a study [PMID: 35841888], published in August 2022 (our BiorXiv paper was available since May) which has a Ribo-seq ORF predictor very much resembling our algorithm. They did not perform any benchmarking against existing tools. This is because the objective benchmarking of the Ribo-seq ORF predictors is impossible due to the lack of a proper gold standard.

In the absence of a gold standard, we could rely only on manual evaluation of the profiles (which we do and we also provide plots of footprint densities in the manuscript as well as in Trips-Viz, so that the readers can visually evaluate the evidence).

Again, I was arguing about the work's significance to the field. They did not provide us method with novelty and did not generate original high-throughput useful data yet even in current revised version.

There are three aspects of the novelty in our approach.

1. New algorithm for the detection of translation.
2. The use of aggregated data.
3. The combination of Ribo-seq evidence with phylogenetic evidence of protein coding evolution (PhyloCSF).

These aspects have led to the annotation of novel coding sequences in numerous protein-coding genes, and - as far as we know - each of these cases represents a novel discovery.

We added the following text:

PhyloCSF is a state-of-the-art method for assessing the evolutionary protein-coding potential of a genomic region based on multiple sequence alignment.

Furthermore we introduced the Trips-viz ORF predictor as a part of the Trips-viz environment. We mentioned its key features and advantages in the comment above and in the text and we also pointed out that a very similar algorithm has been very recently published.

Having a comprehensive and accurate gene annotation is crucially important for a wide research community especially for clinicians who heavily rely on gene annotation with regards to the variant interpretation. Typically variant calling is based on annotated coding sequences thus discovery of novel protein-coding regions is necessary for ensuring that pathogenic variants are not overlooked.

We generated 3 sets of genes with non-AUG N-terminal extensions: PhyloSET, RiboSET and RiboSET_ext and we've been gradually annotating them. These sets provide information about additional protein coding regions located in previously thought to be untranslated regions (5'UTRs) and can be used for inferring variation impact and for experimental validation.

We overlapped ClinVar variants with predicted non-AUG extensions from RiboSET_ext and found 124 genes (201 variants) with either pathogenic, likely pathogenic or conflicting interpretations of pathogenicity. This is likely to be an underestimation of variation simply because only annotated coding regions are generally used for variant calling. Similarly, for primary extensions in PhyloSET we found gene *GDF5OS* carrying 4 variants (pathogenic and likely pathogenic).

We also added these lines to the main text, into Reannotation of non_AUG proteoforms:

Having a comprehensive and accurate gene annotation is crucially important for a wide research community especially for clinicians who heavily rely on gene annotation with regards to the variant interpretation. Since we discovered novel protein-coding regions, we overlapped ClinVar variants with predicted non-AUG extensions from RiboSET_ext and

found 124 genes (201 variants) with either pathogenic, likely pathogenic or conflicting interpretations of pathogenicity (**Supplementary Table 9**). This is likely to be an underestimation of variation simply because only annotated coding regions are generally used for variant calling. Similarly, for primary extensions in PhyloSET we found gene *GDF5OS* carrying 4 variants (pathogenic and likely pathogenic, **Supplementary Table 9**). Therefore this set can be used for assessing variation occurring in 5'UTRs.

We also added the following lines to the Methods:

We utilised variants from ClinVar (clinvar_20221015.vcf.gz) and overlapped them with predicted extensions (RiboSET_ext, PhyloSET) using '*bedtools intersect*'.

Go back to original comments, although the authors added more details about their methods/tools, this manuscript, in essence, just used tools (i.e., Trip-Viz) that have been published before to do some analyses. Thus, this part should not be the novelty.

Although they analyzed 11 datasets, the data is still small relative to tons of existed Ribo-seq data and proteomics data, which cannot provide an impressive landscape of non-AUG initiated proteoforms.

Again, we thank the reviewer for drawing our attention to this matter as the text in the previous version somewhat undermined the scale of our effort.

It is important to point out that for this analysis we started with over 1000 ribosome profiling datasets from 48 published papers which have been processed and are currently available in Trips-Viz. However, for accurate detection of N-extensions it is crucial to detect the correct translated frame. Thus, we analysed all these datasets to determine the strength of triplet periodicity. We chose only 152 datasets from 13 research publications that produced ribo-seq data with high periodicity score (more than 0.5). Perhaps misleadingly we also used the number of research papers used as the number of datasets. This is now corrected, we also added an explanatory statement in the "Detection of translated regions in Trips-Viz" section of Methods explaining this.

We provided periodicity scores for each study in Supplementary table 8.

In relation to the entire landscape of N-terminal extensions in human genes, this would require a generation of ribosome profiling data from virtually all human cells, which is currently not feasible for technical and ethical reasons.

We emphasised that the aim of our study was not to exhaustively detect all existing non-AUG initiated proteoforms, but to develop methodology for this and explore the relationship between the two approaches. Unexpectedly we found that the number of N-extensions detectable with ribosome profiling obtained in a limited number of human cells by far exceeds that detectable with current phylogenetic approaches. This points to the existence of thousands of translated non-AUG initiated proteoforms with no evidence of purifying selection across mammals. We made the following change in the discussion to clarify this:

We did not aim to exhaustively detect all existing non-AUG initiated proteoforms but rather compare two approaches and the relationship between translated N-extensions detected with ribosome profiling and evidence of protein coding evolution in the corresponding genes.

As the authors emphasized their aim of this study is to develop methodology, they need to compare with other methodologies to show the novelty and reliability, but they did not in both versions.

Indeed we have the following aims: (1) develop methodology for detection of non-AUG proteoforms based on translation and evolutionary approaches; (2) detect novel non-AUG proteoforms; (3) compare gene sets resulted from these methods.

The main outcome of the study is stated in the title - this way we wanted to emphasise that we did not intend the reader to interpret the main goal of the paper as only development of the new methodology for detection of the non-AUG proteoforms.

Novelty and reliability of our methodology is provided in the above comments (the description of Trips-viz ORF caller and why we chose PhyloCSF).

In order to avoid additional work suggested by the reviewer, the author did not improve their manuscript enough, but chose to frequently change the main aim of their study in the Responses. It is hard to accept.

More importantly, an interesting story, such as a genomic/genetical pattern or mechanism, should be proposed, but it seems to be lack. Taking together, this research needs to be improved on the novelty and workload, and does not fulfil the publication standard of Nature Communications at its current stage.

The mechanisms involved in non-AUG initiation in general are reviewed e.g. in <http://www.genesdev.org/cgi/doi/10.1101/gad.305250.117>, <https://doi.org/10.1186/s13059-022-02674-2> and briefly described in our introduction. It is not clear to us what the reviewer meant by genomic/genetical pattern.

Again, I was arguing about their work's significance. One of the reasons is that their work failed to provide any new stories about the genomic or genetical pattern/principles in both previous and current versions.

As we mentioned above, we showed the occurrence of pathogenic variants from ClinVar within predicted non-AUG extensions and it's likely to be an underestimation due to the protein-coding region's bias in variant calling. We also tested hypothesis that alternative N-termini encoded with non-AUG carry alternative compartmentalisation signals, although there was no enrichment found in comparison to control genes with no translation upstream, we still cannot exclude the possibility that at least some of them genuinely contain alternative localisation signals. Such an abundance of translated unconserved in mammals non-AUG proteoforms also raised a need for considering Ribo-seq data as primary evidence of protein-coding ability and we are currently working on that. Novel proteoforms are gradually annotated.

We also emphasise that our novel annotations are found within genes of clear medical relevance, such as *JUN*. It is reasonable to assume that such N-t extensions will hold biological information that will prove to be of profound importance in genomic science and medicine. We accept that we are not in a position to describe the precise functionality of such N-terminal extensions in physiological terms. These aspects will need to be studied with single locus high granularity experiments, as part of dedicated research projects. However, others will now be able to explore these possibilities based on our work, as we are producing reference GENCODE annotation. Without our work here, the type of 'stories' that the reviewer has in mind would not be able to be told by anyone.

We also added following lines into the discussion:

The substantial number of pathogenic and likely pathogenic variants from ClinVar overlapping with predicted nonAUG extensions and the fact that novel annotations are found within genes of clear medical relevance, such as *JUN*, it is reasonable to assume that such extensions will hold biological information that will prove to be of profound importance in genomic science and medicine.

As most of the non-AUG translation belongs to translation error, the overlap with ClinVar likely means nothing. The authors did not show additional evidence to validate the possible functions, which do not necessarily need wet experiments.

Some concerns:

The first problem is the title “Thousands of human non-AUG extended proteoforms lack evidence of evolutionary selection among mammals”. In fact, the main work or goal of this study is the identification of some non-AUG initiated proteoforms, which is far from the content in the title. The authors need to apply as many ways as possible to validate the title’s content (especially for the “lack evidence”), but they did not. The authors seem to be deliberately overstating their research to draw attentions to readers.

The title indeed focuses on an important outcome of our study that has not been emphasised enough in the manuscript.

Describing features of extensions, we focused on only hundreds of detected genes taking a very conservative threshold (Fig.3d) initially and we argue that it is better to use a high-confident set which is more likely devoid of false positives.

The relaxed thresholds lead to detection of thousands of translated non-AUG extensions enriched with those from PhyloSET up to thresholds (6000-7000 rank) see Fig. 3d. Most lack phylogenetic support (considering that PhyloSET has only 60 genes). We have now generated RiboSET_ext - a set of 3451 genes with predicted translated extensions based on a relaxed threshold (rank=5000) which is now provided in Supplementary table 2. We also replicated the analysis of non-AUG initiation features for RiboSET_ext (Supplementary Fig.10), it is very similar to what we observed in RiboSET, so it does not contradict with findings that we discovered using the strict threshold.

We claim that a negative PhyloCSF score based on 120-mammals alignment calculated for extensions in both RiboSET (Fig.3h) and RiboSET_ext (Supplementary Fig.10a) provides us

with evidence of a lack of evolutionary selection among mammals. We added following sentences in 'Characterisation of genes with predicted non-AUG initiation' section:

More importantly, the PhyloCSF score of upstream regions of RiboSET_ext is also skewed towards large negative values providing us with evidence of no evolutionary selection among mammals for thousands of translated non-AUG extensions (Supplementary Fig.10a).

Perhaps, the reviewer misinterpreted our claim as “evidence of a lack of selection” in general. This is not the case. We cannot exclude a possibility of evolutionary selection acting on these genes within a narrow phylogenetic group such as primates, hominids or even humans only and we discuss such a possibility in the Discussion.

As the authors emphasized above their aim of this study is to develop methodology, but their ABSTRACT and title does not reflect the aim in both versions. It is hard to accept that the authors did not talk about the aim of this study at all in the ABSTRACT but only described the result in one or two panels in the manuscript.

As mentioned in the above comments, developing the methodology was one of the necessary steps and it should not be read as the only thing that was done in the study.

Abstract outlines the Trips-viz Ribo-seq ORF predictor and analysis of 120 mammal genomes alignment as developed methodology and the comparison between 2 approaches.

If the authors want to draw a conclusion that “Thousands of human non-AUG extended proteoforms lack evidence of evolutionary selection among mammals”, they need to provide more evidence to validate this point. Only one or two panels with evidence are not enough.

The title signifies main parts/conclusions.

1. Translation of novel non-AUG proteoforms is supported with analysing a large number of publicly available Ribo-seq data aggregated in a uniform manner.
2. Lack of protein-coding evolution upstream of annotated AUG codons in mammals - we proved that utilising state-of-the-art PhyloCSF score and 120 mammals alignment.
3. The existence of a small overlap which is stated in the main text and indeed shown only in one panel, **Fig.3f**. However, we do not agree with the reviewer that the strength of evidence should be based on the number of figures presented in a manuscript as it may lead to artificial and unnecessary inflation of manuscripts' sizes rather than strengthening the evidence. One clear analysis is sufficient.

See the below my original comment: “In addition, non-AUG extended proteoforms are evolutionarily un-conserved...”, this work lacks novelty. One already known pattern do not need the authors to claim it again only based on one or two panels. I was asking the authors to provide more new evidence to validate their title so that the novelty could be improved, but the authors almost did nothing for this in the revisions.

In addition, non-AUG extended proteoforms are evolutionarily un-conserved is already known, because most of them are translational molecular error that has been reported before (Mol Biol Evol. 2020;37:2015–28). Even the authors tried their best to validate this focus, it lacks novelty.

1. We disagree with placing an equal sign between the lack of selection within a specific phylogenetic group to “a molecular error”. First the selection may occur in a narrower phylogenetic group as argued above. Second, most of the evolution occurring in species with a small effective population size such as mammals is neutral. This was established through seminal works by Motoo Kimura in the 70s and 80s. Just as an example, consider olfactory receptors in humans. While most are not selected as evident from the large number of corresponding pseudogenes in our genome, we wouldn’t term expression of an active olfactory receptor as a “molecular error”. If we accept this point of view we would need to refer to whole classes of non-conserved biological molecules such as lncRNAs as “molecular errors”. On the other hand, all evolution may be described as a result of “molecular errors” as errors are a major driver of evolution.

I was arguing about the significance of this study not the molecular error itself, as that non-AUG extended proteoforms are evolutionarily un-conserved has been reflected in Mol Biol Evol. 2020;37:2015–28.

“Slightly deleterious or nearly neutral” is better than “neutral” when talking about molecular evolution.

Under the “molecular error hypothesis” in Mol Biol Evol. 2020;37:2015–28, the authors should not refer to all lncRNAs as molecular error, but could refer to most of them as molecular error.

The authors can analogically treat “molecular error” as a kind of mutation. Most mutations are deleterious and un-conserved, but they are one major driver of evolution.

Regretfully we failed to interpret these comments of the Reviewer and to understand their relationship to our manuscript.

Go back to original comment: “In addition, non-AUG extended proteoforms are evolutionarily un-conserved is already known, because most of them are translational molecular error that has been reported before (Mol Biol Evol. 2020;37:2015–28). Even the authors tried their best to validate this focus, it lacks novelty.” I was talking about the novelty. The authors did not reply with a reasonable explanation and even shown some inaccurate understanding on evolution in the first Response.

2. Furthermore Mol Biol Evol. 2020;37:2015–28, focuses on non-AUG initiation irrespective of its functional consequences, e.g. whether they result in uORFs or N-extensions. Most importantly their study was limited to QTI-seq data. While QTI-seq data are useful and we used it in our own study when trying to predict locations of potential non-AUG codons, these data are very noisy and subject to technical errors.

First QTI-seq has been shown to enrich ribosomes not only at initiating codons, but also at two specific codons during elongation see <http://www.rnajournal.org/cgi/doi/10.1261/rna.049908.115> and Dmitriev SE, Akulich KA, Andreev DE, Terenin IM, Shastky IN. The peculiar mode of translation elongation inhibition by antitumor drug harringtonin. FEBS J. 2013;280:51.

QTI-seq paper was published in 2015, but the cited references here, the latter was published in 2013, although the former was published in 2015 but it does not cite QTI-seq. The authors did not show the analyses based QTI-seq and did not provide any evidence (not the WORDS in the response) about why did not use QTI-seq data in both versions.

The reviewer is correct that Dmitriev et al (2013) preceded the QTI-seq paper (2015). Dmitriev et al deals with an artefact generated by a drug Harringtonine (arrest of ribosomes at AAG/AGG codons during elongation). QTI-seq is based on the drug lactimidomycin that was first introduced in 2012:

Lee S, Liu B, Lee S, Huang SX, Shen B, Qian SB. Global mapping of translation initiation sites in mammalian cells at single-nucleotide resolution. Proc Natl Acad Sci U S A . 2012 Sep 11;109(37):E2424-32.

We have shown that the artefact reported by Dmitriev et al for Harringtonin also occurs in the case of lactimidomycin:

Michel AM, Andreev DE, Baranov PV. Computational approach for calculating the probability of eukaryotic translation initiation from ribo-seq data that takes into account leaky scanning. BMC Bioinformatics. 2014 Nov 21;15(1):380. doi: 10.1186/s12859-014-0380-4. PMID: 25413677; PMCID: PMC4245810.

This publication also precedes the QTI-seq publication. However, QTI-seq is based on the use of lactimidomycin in combination with puromycin. The authors of QTI-seq did not provide any evidence that puromycin treatment removes this artefact in their publication:

Gao X, Wan J, Liu B, Ma M, Shen B, Qian SB. Quantitative profiling of initiating ribosomes in vivo. Nat Methods. 2015 Feb;12(2):147-53. doi: 10.1038/nmeth.3208. Epub 2014 Dec 8. PMID: 25486063; PMCID: PMC4344187.

Subsequently, we see no reason to believe that QTI-seq should be free from this artefact and therefore refer to a prior study in relation to this artefact.

Nonetheless, despite these artefacts we do use the data obtained with inhibitors of translation initiation. We selected the data for the analysis based on the quality of the datasets (such as the strength of triplet periodicity) irrespective of the specific drugs that were used. The description of studies that we used was available in Supplementary Table 8 for the previous round of review process. We used aggregation of both types of Ribo-seq data, those generated with inhibitors of elongation as well as initiation. Asterisks show initiating ribosome profiling studies. The advantage of aggregating datasets obtained with

different ribosome profiling protocols is that the effects of artefacts pertinent to specific protocols is reduced.

Happy to see the authors addressed this concern in current response, which is much better than previous response (e.g., mis-citations). The authors should explain this in the main text as well, otherwise the readers will be puzzled because it is hard to notice it when only one word mentions QTI-seq.

Furthermore QTI-seq is very narrow, reads supporting initiation at a particular codon map to a corresponding single location. Assessing translation based on a single translation event is highly unreliable due to sequencing biases, mappability and other issues that we discussed in detail in the first section of our recent review on the topic (doi: 10.1002/wrna.1577).

I did not find the citation of QTI-seq in the first section of the review (doi: 10.1002/wrna.1577); the discussion about QIT-seq is only one small paragraph in the fifth section, but did not talk about the problem you mentioned here either. In addition, if the authors think it is not reliable to base on a single translation event, they can combine the Ribo-seq and QTI-seq, but they did nothing at genome scale in both versions.

See the above comments - we did it originally.

The approach that we used here is by far more reliable and robust because it relies on reads aligned from a large region of mRNA and multiple studies, thus reducing the effect of artefacts and biases of individual locations and studies.

The authors did not provide results using other methods and datasets, how to say "more reliable"? For example, comparing with the combination of QTI-seq and Ribo-seq, which will be better?

The higher reliability of using several data points per transcript than a single one is expected from the central limit theorem. The larger the sample size the better it represents the real distribution.

This is particularly pertinent to ribosome profiling which is significantly affected by sequencing biases, e.g. due to substrate specificity during adaptor ligation. The interpretation of ribosome profiling data based on a single location is therefore dangerous. For example, in this study:

Han Y, Gao X, Liu B, Wan J, Zhang X, Qian SB. Ribosome profiling reveals sequence-independent post-initiation pausing as a signature of translation. Cell Res. 2014 Jul;24(7):842-51. doi: 10.1038/cr.2014.74. Epub 2014 Jun 6. PMID: 24903108; PMCID: PMC4085768.

The authors interpreted the peak at the 5th codon in metagene profile (Fig R1) as a genuine reflection of ribosome progression over mRNA and published the above paper suggesting that this is a real ribosomal pause. However, the peak occurs because the 5' end of

ribosomal footprints coincides with the location of AUG codon since the ribosome protects approximately 4 codons 5' of its A-site. Sequences with AUG at the 5' end of footprints were preferentially recognised by ligase enzymes thus leading to preferential sequencing at this location.

[redacted]

Figure R1. A high peak of ribosome footprint density is often observed 5 codons downstream of the initiation site. The panels are taken from Figure 1 of Han et al (2014)

The improvements in the ribosome profiling library generation led to the reduction in sequencing biases. Therefore this peak is not observed in most recently generated data (Fig R2).

[redacted]

Figure R2. The metagene profile for a recently generated data does not contain a peak at the 5th codon. The data are from Gameiro PA, Struhl K. Nutrient Deprivation Elicits a Transcriptional and Translational Inflammatory Response Coupled to Decreased Protein Synthesis. Cell Rep. 2018 Aug 7;24(6):1415-1424. doi: 10.1016/j.celrep.2018.07.021. PMID: 30089253; PMCID: PMC6419098.

While the technology improved substantially, relying on a large sample of data (in our case the number of coordinates per translated region) is a more cautious approach which we chose in our study.

Of note, the MBE study did not rely on phylogenetic approaches to infer non-adaptiveness of extensions encoded by the non-AUG starts; their claims are based mostly on comparisons between translational amount of genes and TIS diversity.

MBE study did, but the authors ignored at reading. Please see it's Fig. 4 about the phastCons scores.

On the Fig.4, PhastCons was used only for assessing Kozak sequence contexts unlike our study where protein-coding ability of the entire extension sequence was assessed with a PhyloCSF score.

They key difference between PhyloCSF and PhastCons is that PhyloCSF tracks represent a signal of constraint specifically for protein-coding function, whereas the signal represented by PhastCons is independent of the cellular function imposing the constraint and demonstrates rather nucleotide-level conservation.

When the initiation is nonadaptive, it is reasonable to speculate the extension is also nonadaptive. The MBE study does not necessarily use the same approaches as the authors did, but they had already studied the non-adaptiveness of non-AUG starts from phylogenetic ways.

There are some important datasets this study did not collect and use. For translation initiation, the high throughput sequencing method, GTI/QTI-seq (Nature methods, 12(2), 147-153), can directly catch the initiation codon, level and position. Thus, it is more direct and useful than Ribo-seq. I did not see how this study include or analyzes this kind of data.

As we discussed above, while useful, GTI/QTI-seq is very noisy and predictions made based on it are unlikely to be robust since they would rely on very narrow mapping. We did use it however in manual assessments of exact potential locations of non-AUG starts during manual evaluation of the selected genes.

See the above about the QTI-seq. The authors did not show the results based QTI-seq.

We replied to this comment in the comments above.

The authors should explain this in the main text as well, otherwise the readers will be puzzled because it is hard to notice it when only one word mentions QTI-seq.

The authors did comparisons, including between the two methods and between this study and previous study, but the overlap part is always small, which makes people suspect the reliability and rationality of the data or methods used. As the main aim in this study is not on the comparison, the authors should not take much effort on the comparison.

We extensively discussed why overlap between phylogenetic and ribo-seq approaches is smaller. It is one of the most important outcomes of our study.

The small overlap with the previous study is due to various reasons that we thoroughly explained in Results. E.g. in comparison with Ivanov et.al there are multiple reasons including differences in RefSeq and GENCODE; a considerable fraction of originally discovered non-AUG genes have been already annotated in the recent GENCODE version. Also, we tried to rediscover genes that have not yet been annotated.

Back then, in 2011, massive multiple sequence codon alignments such as 100 vertebrates or 120 mammals were not available and the analysis was based solely on human-mouse paired alignment which is likely to have less information in comparison to 120-mammals. Also, there was only a single Ribo-seq dataset available for humans and we used it purely for illustrative purposes. It was not used for detection of N-extensions.

I was arguing about the reliability and rationality of the data or methods they used. Although they explain the overlap part is small, but lack the discussion about the reliability of their method.

We discussed the issues that may arise while using PhyloCSF score in the Discussion.

We added the following text into the discussion regarding Trips-viz algorithm:

For prediction of translated non-AUG NTEs we utilised an Ribo-seq ORF predictor - a new addition to the Trips-viz environment - platform for analysing and visualising ribosome profiling data. The key advantage of the ORF predictor is its convenience: users have a large collection of Ribo-seq samples already embedded into the platform and can start using the predictor and visualise results straight away. Its output is in the form of a ranked list of translated ORFs that can be treated as both advantage and disadvantage since it provides a flexibility for setting out your own threshold although it might not be always straightforward to select one. In general, the Trips-viz ORF predictor algorithm is adapted individually for all classes of ORFs including uORFs, CDSs, NTEs, nORFs and dORFs. It is based on the triplet periodicity signal in the entire ORF and therefore it performs its best when applied to the aggregation of elongating ribosome data either with or without initiating ribosome profiling.

Reviewer #3 (Remarks to the Author):

The revisions have addressed my concerns with the previous version.

I did notice a few minor points in the current manuscript:

In Fig. 3h, the x- and y-axis labels appear to be switched

We thank the reviewer for pointing that out and we switched labels.

On p. 7, it is written, "To show whether a threshold of 500 top ranking candidates is too conserved..." and I think this should be "conservATIVE"

We corrected that in a manuscript.

On p. 8, there is a citation to "(Ivanov et al 2011, ref 21)" but this is ref 22 in the current version.

We corrected that in a manuscript.

REVIEWER COMMENTS

In the following response to Reviewer #1, **the original reviewer comments are shown in bold**, our previous response to these comments is in roman, **the new Reviewer comments are highlighted in blue**, our new response to them is highlighted in green and orange indicates the related changes in the manuscript. The newest Reviewer's comments are in **orange**, our response in **green**.

Reviewer #1 (Remarks to the Author):

In this manuscript, Fedorova and the co-authors used two approaches to identify non AUG initiated proteoforms. The first is phyloCSF, in which 60 candidate genes were identified based on genomic data; while the second, Trips-Viz based on Ribo-seq and proteomics data, found 392 candidate genes. However, only 8 genes are common between the two methods. By comparing their dataset with GENCODE v35, less than half annotated non-AUG proteoforms were validated. They further studied the properties of predicted non-AUG initiation codons, including the distribution of start codon types, TIS sequence logo and frequency, mRNA secondary structure, and domains. Finally, they updated 11 genes about the non-AUG initiated proteoforms.

We kindly thank the reviewer for their feedback.

We would like to note that we did not compare our datasets with GENCODE v35. We excluded genes that have already been annotated in GENCODE v35 from RiboSET and PhyloSET. Instead, we tested whether we were able to re-identify genes with already annotated non-AUG extensions in GENCODE v35 from the Ivanov et al 2011 study using the PhyloCSF score if we moved the start to the originally annotated AUG. Indeed, only 11 genes in the original study have a positive PhyloCSF score and we discuss the possible explanations of such an overlap in the Discussion. Briefly, in Ivanov et al only human-mouse paired alignments were used unlike the 120-mammals alignment that we used for PhyloCSF score calculation. Also we calculated the score for 50 upstream codons while conserved extension could be way shorter thus extra upstream codons could add a large negative value to the final score.

We updated annotations for 11 genes, however we expect many more to be annotated in the future, for example SLC25A32 was annotated as such during the revision of this paper.

I was not arguing about their comparison, instead I was talking about the work's significance to the field as only 11 genes were updated.

Current gene annotation guidelines rely heavily on phylogenetic evidence for annotation of 'missing' protein coding sequences. In order to be annotated, proteins are required to be evolutionary 'old', essentially to present evidence as evolving as coding sequence across the mammalian order as a minimum. Ape- or human- specific proteins are not typically annotated without experimental support for protein existence, which in practice generally means proteomics data.

If ORF protein-coding ability is endorsed only by Ribo-seq, it is not subjected to the immediate annotation since no suitable guidelines have been introduced yet.

Also, each individual gene is being studied manually by the GENCODE curators. At the moment of first submission, we had 11 genes annotated, before the first revision another gene has been introduced (*SLC25A32*) and now another 18 (*ADO, JUN, HDGF, POGZ, BRD7, PIM2, PTMS, CARM1, TARBP2, CHTOP, TNKS2, KAT7, FHIP2A, SYNCRIP, KCTD9, PPP4R2, ZNF384, SNRNP25*) have gained non-AUG models in both human and mouse. It is an ongoing effort and only genes with both sufficient phylogenetic evidence and clear Ribo-seq signal are considered. Such a massive translation at non-AUG starts raised an important issue regarding our understanding of protein-coding evolution. Currently we are working on novel annotation policies which will enable using Ribo-seq data as an independent evidence of protein-coding ability. It is a community effort therefore it will take a while before more proteoforms can be annotated.

As most of the non-AUG translation belongs to translation error, I did not see the significance even the authors added another 18 genes with non-AUG initiation.

We disagree with the reviewer on the putting equal sign between absence of conservation in mammals and the translation error. Novel nonAUG proteoforms have been gradually annotated.

Although the authors claim they “develop” the approaches for identifying of non-AUG translation initiation, the methods/tools used are not developed in the study. The authors did not generate any high-throughput data, they thus did not produce useful resources yet.

We thank the reviewer for bringing our attention to the fact that we did not sufficiently highlight the new methods developed in our work.

First of all, we described the algorithm which allows for prediction of translated ORFs using aggregated Ribo-seq data. While we used it in this manuscript only for identification of non AUG N-terminal extensions, its implementation in the Trips-Viz browser can be used for detecting any kind of unannotated translation including upstream, downstream, nested and overlapping ORFs. The detailed description of the algorithm is in Supplementary Methods. We also provided a link to the code at github - https://github.com/skiniry/Trips_Viz/blob/master/orfquery_routes.py.

We apologise for not explicitly stating this in the original version of manuscript and thank the Reviewer for pointing this out. We have now added the following sentence to the corresponding Results:

We have developed a translating ORF predictor, an algorithm for detecting translated ORFs using aggregated Ribo-seq data (Supplementary Methods). The implementation of this algorithm is now available in the Trips-Viz browser.

And also modified the abstract:

Here we analysed a large number of publicly available ribo-seq datasets to identify novel,

previously uncharacterised non-AUG proteoforms using Trips-Viz implementation of a novel algorithm for detecting translated ORFs.

In addition to the general purpose of translated ORF detection, specifically for the purpose of this work, we designed and described a pipeline that can detect non-AUG N-terminal extensions using 2 orthogonal approaches. This pipeline includes extracting upstream regions of canonically annotated transcript models, deriving genomic coordinates of theoretical N-terminal extensions (NTEs), filtering overlaps with known coding regions, stitching together exons of NTEs and calculating PhyloCSF scores. Another branch includes applying the aforementioned ORF-predictor on transcript models. To our knowledge, this pipeline has not been used for detection of non-AUG initiated proteoforms anywhere else.

They did not describe their new approach in the original manuscript, so that I had to point out the problem. In the revised version, they provided some details, but did not show its novelty relative to others. In addition, the reliability and advancement of their method is unknown.

We described the algorithm in Supplementary Methods in the previous revised version. For the clarity now we have added following description of Trips-viz ORF predictor algorithm and its advancements:

Trips-viz is a computational data environment for analysing Ribo-seq data on a transcriptome level^{37,38}. It contains thousands of uniformly processed public Ribo-seq data and provides tools for analysis and visualisation of translation. Our recent addition to the Trips-viz platform is Ribo-seq ORF predictor which outputs a list of ranked translated ORFs. The biggest advantage of the tool is that a large number of processed public Ribo-seq data is already available to users and they can apply ORF predictor and visualise results immediately. The tool is tailored to detect different types of ORFs including uORFs, N-terminal extensions (NTEs), nORFs, CDSs and dORFs. The algorithm for NTEs is based on triplet periodicity present across the entire length of extension and utilises patterns of ORF translation such as consistency of ribosome footprint triplet periodicity within the reading frame, the increase of footprint density at the potential start, non zero coverage and average read density (see **Supplementary Methods**). Of note, it automatically filters out regions that overlap with coding exons. We employed this algorithm to detect non-AUG N-terminally extended proteoforms using aggregated elongating and initiating ribosome profiling data with high triplet periodicity score (**Supplementary Table 8**).

During the round of review, we also noticed a study [PMID: 35841888], published in August 2022 (our BiorXiv paper was available since May) which has a Ribo-seq ORF predictor very much resembling our algorithm. They did not perform any benchmarking against existing tools. This is because the objective benchmarking of the Ribo-seq ORF predictors is impossible due to the lack of a proper gold standard.

In the absence of a gold standard, we could rely only on manual evaluation of the profiles (which we do and we also provide plots of footprint densities in the manuscript as well as in Trips-Viz, so that the readers can visually evaluate the evidence).

Again, I was arguing about the work's significance to the field. They did not provide us method with novelty and did not generate original high-throughput useful data yet even in current revised version.

There are three aspects of the novelty in our approach.

1. New algorithm for the detection of translation.
2. The use of aggregated data.
3. The combination of Ribo-seq evidence with phylogenetic evidence of protein coding evolution (PhyloCSF).

These aspects have led to the annotation of novel coding sequences in numerous protein coding genes, and - as far as we know - each of these cases represents a novel discovery.

We added the following text:

PhyloCSF is a state-of-the-art method for assessing the evolutionary protein-coding potential of a genomic region based on multiple sequence alignment.

Furthermore we introduced the Trips-viz ORF predictor as a part of the Trips-viz environment. We mentioned its key features and advantages in the comment above and in the text and we also pointed out that a very similar algorithm has been very recently published.

Having a comprehensive and accurate gene annotation is crucially important for a wide research community especially for clinicians who heavily rely on gene annotation with regards to the variant interpretation. Typically variant calling is based on annotated coding sequences thus discovery of novel protein-coding regions is necessary for ensuring that pathogenic variants are not overlooked.

We generated 3 sets of genes with non-AUG N-terminal extensions: PhyloSET, RiboSET and RiboSET_ext and we've been gradually annotating them. These sets provide information about additional protein coding regions located in previously thought to be untranslated regions (5'UTRs) and can be used for inferring variation impact and for experimental validation.

We overlapped ClinVar variants with predicted non-AUG extensions from RiboSET_ext and found 124 genes (201 variants) with either pathogenic, likely pathogenic or conflicting interpretations of pathogenicity. This is likely to be an underestimation of variation simply because only annotated coding regions are generally used for variant calling. Similarly, for primary extensions in PhyloSET we found gene *GDF5OS* carrying 4 variants (pathogenic and likely pathogenic).

We also added these lines to the main text, into Reannotation of non_AUG proteoforms:

Having a comprehensive and accurate gene annotation is crucially important for a wide research community especially for clinicians who heavily rely on gene annotation with regards to the variant interpretation. Since we discovered novel protein-coding regions, we overlapped ClinVar variants with predicted non-AUG extensions from RiboSET_ext and found 124 genes (201 variants) with either pathogenic, likely pathogenic or conflicting interpretations of pathogenicity (**Supplementary Table 9**). This is likely to be an underestimation of variation simply because only annotated coding regions are generally used for variant calling. Similarly, for primary extensions in PhyloSET we found gene *GDF5OS* carrying 4 variants (pathogenic and likely pathogenic, **Supplementary Table 9**). Therefore this set can be used for assessing variation occurring in 5'UTRs.

We also added the following lines to the Methods:

We utilised variants from ClinVar (clinvar_20221015.vcf.gz) and overlapped them with predicted extensions (RiboSET_ext, PhyloSET) using *'bedtools intersect'*.

Go back to original comments, although the authors added more details about their methods/tools, this manuscript, in essence, just used tools (i.e., Trip-Viz) that have been published before to do some analyses. Thus, this part should not be the novelty.

We disagree with the reviewer here since the ORF predictor embedded in Trips-viz has not been published before and PhyloCSF score for detection of nonAUG proteoforms has not been published anywhere.

Although they analyzed 11 datasets, the data is still small relative to tons of existed Ribo-seq data and proteomics data, which cannot provide an impressive landscape of non-AUG initiated proteoforms.

Again, we thank the reviewer for drawing our attention to this matter as the text in the previous version somewhat undermined the scale of our effort.

It is important to point out that for this analysis we started with over 1000 ribosome profiling datasets from 48 published papers which have been processed and are currently available in Trips-Viz. However, for accurate detection of N-extensions it is crucial to detect the correct translated frame. Thus, we analysed all these datasets to determine the strength of triplet periodicity. We chose only 152 datasets from 13 research publications that produced ribo seq data with high periodicity score (more than 0.5). Perhaps misleadingly we also used the number of research papers used as the number of datasets. This is now corrected, we also added an explanatory statement in the "Detection of translated regions in Trips-Viz" section of Methods explaining this.

We provided periodicity scores for each study in Supplementary table 8.

In relation to the entire landscape of N-terminal extensions in human genes, this would require a generation of ribosome profiling data from virtually all human cells, which is currently not feasible for technical and ethical reasons.

We emphasised that the aim of our study was not to exhaustively detect all existing non AUG initiated proteoforms, but to develop methodology for this and explore the relationship between the two approaches. Unexpectedly we found that the number of N-extensions detectable with ribosome profiling obtained in a limited number of human cells by far exceeds that detectable with current phylogenetic approaches. This points to the existence of thousands of translated non-AUG initiated proteoforms with no evidence of purifying selection across mammals. We made the following change in the discussion to clarify this:

We did not aim to exhaustively detect all existing non-AUG initiated proteoforms but rather

compare two approaches and the relationship between translated N-extensions detected with ribosome profiling and evidence of protein coding evolution in the corresponding genes.

As the authors emphasized their aim of this study is to develop methodology, they need to compare with other methodologies to show the novelty and reliability, but they did not in both versions.

Indeed we have the following aims: (1) develop methodology for detection of non-AUG proteoforms based on translation and evolutionary approaches; (2) detect novel non-AUG proteoforms; (3) compare gene sets resulted from these methods.

The main outcome of the study is stated in the title - this way we wanted to emphasise that we did not intend the reader to interpret the main goal of the paper as only development of the new methodology for detection of the non-AUG proteoforms.

Novelty and reliability of our methodology is provided in the above comments (the description of Trips-viz ORF caller and why we chose PhyloCSF).

In order to avoid additional work suggested by the reviewer, the author did not improve their manuscript enough, but chose to frequently change the main aim of their study in the Responses. It is hard to accept.

We disagree with the reviewer on that matter - we organised and re-formulated better our original goals of the study in order to improve readability, but we did not change them to avoid any additional work suggested by the reviewer.

More importantly, an interesting story, such as a genomic/genetical pattern or mechanism, should be proposed, but it seems to be lack. Taking together, this research needs to be improved on the novelty and workload, and does not fulfil the publication standard of Nature Communications at its current stage.

The mechanisms involved in non-AUG initiation in general are reviewed e.g. in <http://www.genesdev.org/cgi/doi/10.1101/gad.305250.117>, <https://doi.org/10.1186/s13059-022-02674-2> and briefly described in our introduction. It is not clear to us what the reviewer meant by genomic/genetical pattern.

Again, I was arguing about their work's significance. One of the reasons is that their work failed to provide any new stories about the genomic or genetical pattern/principles in both previous and current versions.

As we mentioned above, we showed the occurrence of pathogenic variants from ClinVar within predicted non-AUG extensions and it's likely to be an underestimation due to the protein-coding region's bias in variant calling. We also tested hypothesis that alternative N termini encoded with non-AUG carry alternative compartmentalisation signals, although there was no enrichment found in comparison to control genes with no translation upstream, we still cannot exclude the possibility that at least some of them genuinely contain alternative localisation signals. Such an abundance of translated unconserved in mammals

non-AUG proteoforms also raised a need for considering Ribo-seq data as primary evidence of protein-coding ability and we are currently working on that. Novel proteoforms are gradually annotated.

We also emphasise that our novel annotations are found within genes of clear medical relevance, such as *JUN*. It is reasonable to assume that such N-t extensions will hold biological information that will prove to be of profound importance in genomic science and medicine. We accept that we are not in a position to describe the precise functionality of such N-terminal extensions in physiological terms. These aspects will need to be studied with single locus high granularity experiments, as part of dedicated research projects. However, others will now be able to explore these possibilities based on our work, as we are producing reference GENCODE annotation. Without our work here, the type of 'stories' that the reviewer has in mind would not be able to be told by anyone.

We also added following lines into the discussion:

The substantial number of pathogenic and likely pathogenic variants from ClinVar overlapping with predicted nonAUG extensions and the fact that novel annotations are found within genes of clear medical relevance, such as *JUN*, it is reasonable to assume that such extensions will hold biological information that will prove to be of profound importance in genomic science and medicine.

As most of the non-AUG translation belongs to translation error, the overlap with ClinVar likely means nothing. The authors did not show additional evidence to validate the possible functions, which do not necessarily need wet experiments.

As we answered in the first comment, translational error is not the ultimate explanation of translated nonAUG extensions thus ClinVar variation within extensions can still be useful for gaining insights in genomic research and medicine.

Some concerns:

The first problem is the title "Thousands of human non-AUG extended proteoforms lack evidence of evolutionary selection among mammals". In fact, the main work or goal of this study is the identification of some non-AUG initiated proteoforms, which is far from the content in the title. The authors need to apply as many ways as possible to validate the title's content (especially for the "lack evidence"), but they did not. The authors seem to be deliberately overstating their research to draw attentions to readers.

The title indeed focuses on an important outcome of our study that has not been emphasised enough in the manuscript.

Describing features of extensions, we focused on only hundreds of detected genes taking a very conservative threshold (Fig.3d) initially and we argue that it is better to use a high confident set which is more likely devoid of false positives.

The relaxed thresholds lead to detection of thousands of translated non-AUG extensions enriched with those from PhyloSET up to thresholds (6000-7000 rank) see Fig. 3d. Most

lack phylogenetic support (considering that PhyloSET has only 60 genes). We have now generated RiboSET_ext - a set of 3451 genes with predicted translated extensions based on a relaxed threshold (rank=5000) which is now provided in Supplementary table 2. We also replicated the analysis of non-AUG initiation features for RiboSET_ext (Supplementary Fig.10), it is very similar to what we observed in RiboSET, so it does not contradict with findings that we discovered using the strict threshold.

We claim that a negative PhyloCSF score based on 120-mammals alignment calculated for extensions in both RiboSET (Fig.3h) and RiboSET_ext (Supplementary Fig.10a) provides us with evidence of a lack of evolutionary selection among mammals. We added following sentences in 'Characterisation of genes with predicted non-AUG initiation' section:

More importantly, the PhyloCSF score of upstream regions of RiboSET_ext is also skewed towards large negative values providing us with evidence of no evolutionary selection among mammals for thousands of translated non-AUG extensions (Supplementary Fig.10a).

Perhaps, the reviewer misinterpreted our claim as "evidence of a lack of selection" in general. This is not the case. We cannot exclude a possibility of evolutionary selection acting on these genes within a narrow phylogenetic group such as primates, hominids or even humans only and we discuss such a possibility in the Discussion.

As the authors emphasized above their aim of this study is to develop methodology, but their ABSTRACT and title does not reflect the aim in both versions. It is hard to accept that the authors did not talk about the aim of this study at all in the ABSTRACT but only described the result in one or two panels in the manuscript.

As mentioned in the above comments, developing the methodology was one of the necessary steps and it should not be read as the only thing that was done in the study.

Abstract outlines the Trips-viz Ribo-seq ORF predictor and analysis of 120 mammal genomes alignment as developed methodology and the comparison between 2 approaches.

If the authors want to draw a conclusion that "Thousands of human non-AUG extended proteoforms lack evidence of evolutionary selection among mammals", they need to provide more evidence to validate this point. Only one or two panels with evidence are not enough.

The title signifies main parts/conclusions.

1. Translation of novel non-AUG proteoforms is supported with analysing a large number of publicly available Ribo-seq data aggregated in a uniform manner.
2. Lack of protein-coding evolution upstream of annotated AUG codons in mammals - we proved that utilising state-of-the-art PhyloCSF score and 120 mammals alignment.
3. The existence of a small overlap which is stated in the main text and indeed shown only in one panel, **Fig.3f**. However, we do not agree with the reviewer that the

strength of evidence should be based on the number of figures presented in a manuscript as it may lead to artificial and unnecessary inflation of manuscripts' sizes rather than strengthening the evidence. One clear analysis is sufficient.

See the below my original comment: “In addition, non-AUG extended proteoforms are evolutionarily un-conserved...”, this work lacks novelty. One already known pattern do not need the authors to claim it again only based on one or two panels. I was asking the authors to provide more new evidence to validate their title so that the novelty could be improved, but the authors almost did nothing for this in the revisions.

We want to emphasise again that translated nonAUG extensions detectable with Ribo-seq are non-conserved in mammals (and not just molecular errors) and thus raise awareness of their high frequency which can help to re-evaluate current gene annotation guidelines. We strongly believe that lengthening of the manuscript by additional panels for the sake of just having more panels is not necessary.

In addition, non-AUG extended proteoforms are evolutionarily un-conserved is already known, because most of them are translational molecular error that has been reported before (Mol Biol Evol. 2020;37:2015–28). Even the authors tried their best to validate this focus, it lacks novelty.

1. We disagree with placing an equal sign between the lack of selection within a specific phylogenetic group to “a molecular error”. First the selection may occur in a narrower phylogenetic group as argued above. Second, most of the evolution occurring in species with a small effective population size such as mammals is neutral. This was established through seminal works by Motoo Kimura in the 70s and 80s. Just as an example, consider olfactory receptors in humans. While most are not selected as evident from the large number of corresponding pseudogenes in our genome, we wouldn't term expression of an active olfactory receptor as a “molecular error”. If we accept this point of view we would need to refer to whole classes of non-conserved biological molecules such as lncRNAs as “molecular errors”. On the other hand, all evolution may be described as a result of “molecular errors” as errors are a major driver of evolution.

I was arguing about the significance of this study not the molecular error itself, as that non-AUG extended proteoforms are evolutionarily un-conserved has been reflected in Mol Biol Evol. 2020;37:2015–28.

“Slightly deleterious or nearly neutral” is better than “neutral” when talking about molecular evolution.

Under the “molecular error hypothesis” in Mol Biol Evol. 2020;37:2015–28, the authors should not refer to all lncRNAs as molecular error, but could refer to most of them as molecular error.

The authors can analogically treat “molecular error” as a kind of mutation. Most mutations are deleterious and un-conserved, but they are one major driver of evolution.

Regretfully we failed to interpret these comments of the Reviewer and to understand their relationship to our manuscript.

Go back to original comment: “In addition, non-AUG extended proteoforms are evolutionarily un-conserved is already known, because most of them are translational molecular error that has been reported before (Mol Biol Evol. 2020;37:2015–28). Even the authors tried their best to validate this focus, it lacks novelty.” I was talking about the novelty. The authors did not reply with a reasonable explanation and even shown some inaccurate understanding on evolution in the first Response.

We already addressed the ‘novelty’ issue in the comments above.

2. Furthermore Mol Biol Evol. 2020;37:2015–28, focuses on non-AUG initiation irrespective of its functional consequences, e.g. whether they result in uORFs or N-extensions. Most importantly their study was limited to QTI-seq data. While QTI-seq data are useful and we used it in our own study when trying to predict locations of potential non-AUG codons, these data are very noisy and subject to technical errors.

First QTI-seq has been shown to enrich ribosomes not only at initiating codons, but also at two specific codons during elongation see

<http://www.rnajournal.org/cgi/doi/10.1261/rna.049908.115> and Dmitriev SE, Akulich KA, Andreev DE, Terenin IM, Shastky IN. The peculiar mode of translation elongation inhibition by antitumor drug harringtonin. FEBS J. 2013;280:51.

QTI-seq paper was published in 2015, but the cited references here, the latter was published in 2013, although the former was published in 2015 but it does not cite QTI seq. The authors did not show the analyses based QTI-seq and did not provide any evidence (not the WORDS in the response) about why did not use QTI-seq data in both versions.

The reviewer is correct that Dmitriev et al (2013) preceded the QTI-seq paper (2015). Dmitriev et al deals with an artefact generated by a drug Harringtonine (arrest of ribosomes at AAG/AGG codons during elongation). QTI-seq is based on the drug lactimidomycin that was first introduced in 2012:

Lee S, Liu B, Lee S, Huang SX, Shen B, Qian SB. Global mapping of translation initiation sites in mammalian cells at single-nucleotide resolution. Proc Natl Acad Sci U S A . 2012 Sep 11;109(37):E2424-32.

We have shown that the artefact reported by Dmitriev et al for Harringtonin also occurs in the case of lactimidomycin:

Michel AM, Andreev DE, Baranov PV. Computational approach for calculating the probability of eukaryotic translation initiation from ribo-seq data that takes into account leaky scanning. BMC Bioinformatics. 2014 Nov 21;15(1):380. doi: 10.1186/s12859-014-

0380-4. PMID: 25413677; PMCID: PMC4245810.

This publication also precedes the QTI-seq publication. However, QTI-seq is based on the use of lactimidomycin in combination with puromycin. The authors of QTI-seq did not provide any evidence that puromycin treatment removes this artefact in their publication:

Gao X, Wan J, Liu B, Ma M, Shen B, Qian SB. Quantitative profiling of initiating ribosomes in vivo. Nat Methods. 2015 Feb;12(2):147-53. doi: 10.1038/nmeth.3208. Epub 2014 Dec 8. PMID: 25486063; PMCID: PMC4344187.

Subsequently, we see no reason to believe that QTI-seq should be free from this artefact and therefore refer to a prior study in relation to this artefact.

Nonetheless, despite these artefacts we do use the data obtained with inhibitors of translation initiation. We selected the data for the analysis based on the quality of the datasets (such as the strength of triplet periodicity) irrespective of the specific drugs that were used. The description of studies that we used was available in Supplementary Table 8 for the previous round of review process. We used aggregation of both types of Ribo-seq data, those generated with inhibitors of elongation as well as initiation. Asterisks show initiating ribosome profiling studies. The advantage of aggregating datasets obtained with different ribosome profiling protocols is that the effects of artefacts pertinent to specific protocols is reduced.

Happy to see the authors addressed this concern in current response, which is much better than previous response (e.g., mis-citations). The authors should explain this in the main text as well, otherwise the readers will be puzzled because it is hard to notice it when only one word mentions QTI-seq.

Indeed, we agree here with the reviewer's suggestions about discussing QTI-seq limitations in the manuscript and we added the following into discussion:

...[Trips-viz ORF algorithm]... It can use ribosome profiling data obtained with inhibitors of both, elongation and initiation. The latter is achieved with inhibitors such as lactimidomycin that preferentially arrest ribosomes at the sites of initiation⁵⁸. Variations of this approach are known under different names such as Quantitative Initiation Sequencing QTI-seq³⁶ and are very useful in identification of translation initiation sites. However, the data obtained with this approach should be used with caution because it is intrinsically noisy as reads come from a single location per initiation site. Translation initiation inhibitors also generate artefacts such as arrest of ribosomes at AAG/AGG codons during elongation and may distort translation initiation⁵⁹⁻⁶¹. Thus, we believe that the combination of data obtained with different approaches is more reliable.

And we also mentioned usage of combination of initiating and elongating ribo-seq in the Results:

Another set of candidates (RiboSET) was selected solely based on ranked translated extensions predicted using ribosome profiling data (both elongating and initiating) with Trips-Viz...

It is also mentioned in Methods:

We used the translating ORF predictor in Trips-Viz using Ribo-seq data (elongating and initiating ribosome profiling) ...

Furthermore QTI-seq is very narrow, reads supporting initiation at a particular codon map to a corresponding single location. Assessing translation based on a single translation event is highly unreliable due to sequencing biases, mappability and other issues that we discussed in detail in the first section of our recent review on the topic (doi: 10.1002/wrna.1577).

I did not find the citation of QTI-seq in the first section of the review (doi: 10.1002/wrna.1577); the discussion about QIT-seq is only one small paragraph in the fifth section, but did not talk about the problem you mentioned here either. In addition, if the authors think it is not reliable to base on a single translation event, they can combine the Ribo-seq and QTI-seq, but they did nothing at genome scale in both versions.

See the above comments - we did it originally.

The approach that we used here is by far more reliable and robust because it relies on reads aligned from a large region of mRNA and multiple studies, thus reducing the effect of artefacts and biases of individual locations and studies.

The authors did not provide results using other methods and datasets, how to say "more reliable"? For example, comparing with the combination of QTI-seq and Ribo seq, which will be better?

The higher reliability of using several data points per transcript than a single one is expected from the central limit theorem. The larger the sample size the better it represents the real distribution.

This is particularly pertinent to ribosome profiling which is significantly affected by sequencing biases, e.g. due to substrate specificity during adaptor ligation. The interpretation of ribosome profiling data based on a single location is therefore dangerous. For example, in this study:

Han Y, Gao X, Liu B, Wan J, Zhang X, Qian SB. Ribosome profiling reveals sequence independent post-initiation pausing as a signature of translation. *Cell Res.* 2014 Jul;24(7):842-51. doi: 10.1038/cr.2014.74. Epub 2014 Jun 6. PMID: 24903108; PMCID: PMC4085768.

The authors interpreted the peak at the 5th codon in metagene profile (Fig R1) as a genuine reflection of ribosome progression over mRNA and published the above paper suggesting that this is a real ribosomal pause. However, the peak occurs because the 5' end of ribosomal footprints coincides with the location of AUG codon since the ribosome protects approximately 4 codons 5' of its A-site. Sequences with AUG at the 5' end of footprints were preferentially recognised by ligase enzymes thus leading to preferential sequencing at this location.

[redacted]

Figure

R1. A high peak of ribosome footprint density is often observed 5 codons downstream of the initiation site. The panels are taken from Figure 1 of Han et al (2014)

The improvements in the ribosome profiling library generation led to the reduction in sequencing biases. Therefore this peak is not observed in most recently generated data (Fig R2).

[redacted]

Figure R2. The metagene profile for a recently generated data does not contain a peak at the 5th codon. The data are from Gameiro PA, Struhl K. Nutrient Deprivation Elicits a Transcriptional and Translational Inflammatory Response Coupled to Decreased Protein

Synthesis. Cell Rep. 2018 Aug 7;24(6):1415-1424. doi: 10.1016/j.celrep.2018.07.021. PMID: 30089253; PMCID: PMC6419098.

While the technology improved substantially, relying on a large sample of data (in our case the number of coordinates per translated region) is a more cautious approach which we chose in our study.

Of note, the MBE study did not rely on phylogenetic approaches to infer non-adaptiveness of extensions encoded by the non-AUG starts; their claims are based mostly on comparisons between translational amount of genes and TIS diversity.

MBE study did, but the authors ignored at reading. Please see it's Fig. 4 about the phastCons scores.

On the Fig.4, PhastCons was used only for assessing Kozak sequence contexts unlike our study where protein-coding ability of the entire extension sequence was assessed with a PhyloCSF score.

The key difference between PhyloCSF and PhastCons is that PhyloCSF tracks represent a signal of constraint specifically for protein-coding function, whereas the signal represented by PhastCons is independent of the cellular function imposing the constraint and demonstrates rather nucleotide-level conservation.

When the initiation is nonadaptive, it is reasonable to speculate the extension is also nonadaptive. The MBE study does not necessarily use the same approaches as the authors did, but they had already studied the non-adaptiveness of non-AUG starts from phylogenetic ways.

We cited the MBE study in Discussion:

The lack of phylogenetic support for the majority of non-AUG initiation events have been reported earlier⁶². However, that study did not discriminate between initiation resulting in extended proteoforms or translation of short ORFs. One may argue that unlike N-terminally extended proteoforms, products of short ORF translation are less likely to have phenotypic effects.

There are some important datasets this study did not collect and use. For translation initiation, the high throughput sequencing method, GTI/QTI-seq (Nature methods, 12(2), 147-153), can directly catch the initiation codon, level and position. Thus, it is

more direct and useful than Ribo-seq. I did not see how this study include or analyzes this kind of data.

As we discussed above, while useful, GTI/QTI-seq is very noisy and predictions made based on it are unlikely to be robust since they would rely on very narrow mapping. We did use it however in manual assessments of exact potential locations of non-AUG starts during manual evaluation of the selected genes.

See the above about the QTI-seq. The authors did not show the results based QTI-seq. We replied to this comment in the comments above.

The authors should explain this in the main text as well, otherwise the readers will be puzzled because it is hard to notice it when only one word mentions QTI-seq.

We added the explanation about QTI-seq into the discussion, and mentioned it in the results and methods.

The authors did comparisons, including between the two methods and between this study and previous study, but the overlap part is always small, which makes people suspect the reliability and rationality of the data or methods used. As the main aim in this study is not on the comparison, the authors should not take much effort on the comparison.

We extensively discussed why overlap between phylogenetic and ribo-seq approaches is smaller. It is one of the most important outcomes of our study.

The small overlap with the previous study is due to various reasons that we thoroughly explained in Results. E.g. in comparison with Ivanov et.al there are multiple reasons including differences in RefSeq and GENCODE; a considerable fraction of originally discovered non-AUG genes have been already annotated in the recent GENCODE version. Also, we tried to rediscover genes that have not yet been annotated.

Back then, in 2011, massive multiple sequence codon alignments such as 100 vertebrates or 120 mammals were not available and the analysis was based solely on human-mouse paired alignment which is likely to have less information in comparison to 120-mammals. Also, there was only a single Ribo-seq dataset available for humans and we used it purely for illustrative purposes. It was not used for detection of N-extensions.

I was arguing about the reliability and rationality of the data or methods they used. Although they explain the overlap part is small, but lack the discussion about the reliability of their method.

We discussed the issues that may arise while using PhyloCSF score in the

Discussion. We added the following text into the discussion regarding Trips-viz

algorithm:

For prediction of translated non-AUG NTEs we utilised an Ribo-seq ORF predictor - a new addition to the Trips-viz environment - platform for analysing and visualising ribosome profiling data. The key advantage of the ORF predictor is its convenience: users have a large collection of Ribo-seq samples already embedded into the platform and can start using the predictor and visualise results straight away. Its output is in the form of a ranked list of translated ORFs that can be treated as both advantage and disadvantage since it provides a flexibility for setting out your own threshold although it might not be always straightforward to select one. In general, the Trips-viz ORF predictor algorithm is adapted individually for all classes of ORFs including uORFs, CDSs, NTEs, nORFs and dORFs. It is based on the triplet periodicity signal in the entire ORF and therefore it performs its best when applied to the aggregation of elongating ribosome data either with or without initiating ribosome profiling.

Reviewer #3 (Remarks to the Author):

The revisions have addressed my concerns with the previous version.

I did notice a few minor points in the current manuscript:

In Fig. 3h, the x- and y-axis labels appear to be switched

We thank the reviewer for pointing that out and we switched labels.

On p. 7, it is written, "To show whether a threshold of 500 top ranking candidates is too conserved..." and I think this should be "conservATIVE"

We corrected that in a manuscript.

On p. 8, there is a citation to "(Ivanov et al 2011, ref 21)" but this is ref 22 in the current version.

We corrected that in a manuscript.